# PROVABLE UNCERTAINTY DECOMPOSITION VIA HIGHER-ORDER CALIBRATION

**Gustaf Ahdritz**
Harvard University*

**Aravind Gollakota**
Apple

**Parikshit Gopalan**
Apple

**Charlotte Peale**
Stanford University*

**Udi Wieder**
Apple

## ABSTRACT

We give a principled method for decomposing the predictive uncertainty of a model into aleatoric and epistemic components with explicit semantics relating them to the real-world data distribution. While many works in the literature have proposed such decompositions, they lack the type of formal guarantees we provide. Our method is based on the new notion of higher-order calibration, which generalizes ordinary calibration to the setting of higher-order predictors that predict *mixtures* over label distributions at every point. We show how to measure as well as achieve higher-order calibration using access to $k$-snapshots, namely examples where each point has $k$ independent conditional labels. Under higher-order calibration, the estimated aleatoric uncertainty at a point is guaranteed to match the real-world aleatoric uncertainty averaged over all points where the prediction is made. To our knowledge, this is the first formal guarantee of this type that places no assumptions whatsoever on the real-world data distribution. Importantly, higher-order calibration is also applicable to existing higher-order predictors such as Bayesian and ensemble models and provides a natural evaluation metric for such models. We demonstrate through experiments that our method produces meaningful uncertainty decompositions for image classification.

## 1 INTRODUCTION

Decomposing predictive uncertainty into aleatoric and epistemic components is of fundamental importance in machine learning and statistical prediction (Hüllermeier & Waegeman, 2021; Abdar et al., 2021). Aleatoric (or data) uncertainty is the inherent uncertainty in the prediction task at hand, arising from randomness present in nature's data generating process, while epistemic (or model) uncertainty arises from a model's lack of perfect knowledge about nature. Accurate estimates of these quantities are of great use to practitioners, as they allow them to understand whether the difficulty of their prediction task arises primarily from the data or their model.

As a running example, consider the task of predicting based on an X-ray scan whether a bone will require a cast. Suppose our model issues a 50% prediction for a scan. This could occur because the scan is genuinely ambiguous. A very different scenario is one where the model is unable to decide between two diagnoses: one where a cast is unambiguously necessary and one where it is not. In both scenarios, the total predictive uncertainty is the same. However, the nature of this uncertainty differs; in the first scenario, it is purely aleatoric, while in the second, it is entirely epistemic.

For any such explanation of a model's uncertainty to be trusted, it must be accompanied by some meaningful semantics telling us what it means about the real world. The canonical example of such a notion in the prediction literature is *calibration*, which requires that whenever our model predicts a certain distribution of outcomes, we actually observe that distribution of outcomes on average. In our X-ray example, a 50% prediction would be calibrated if exactly half of all scans with the same prediction did ultimately require casts. But this would be true regardless of which of the two

---

*Work done while interning at Apple. Authors in alphabetical order.

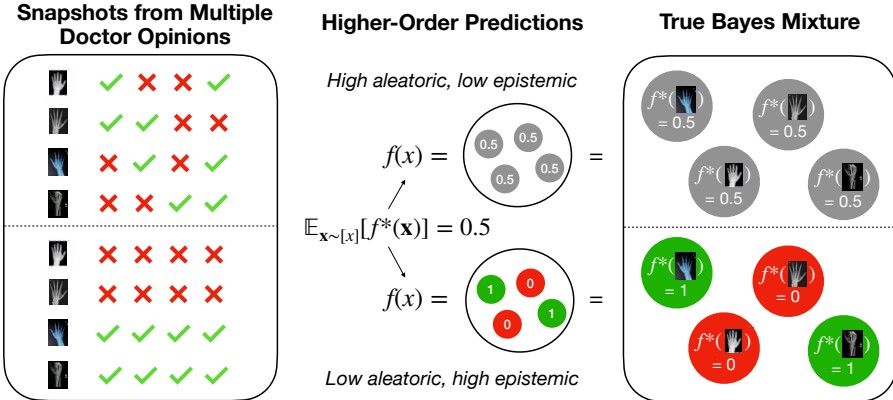

Figure 1: **An illustration of higher-order calibration using the X-ray classification example.** We depict scenarios 1 and 2 on the top and bottom respectively. On the left, we have instances grouped together into one level set $[x]$ by the predictor. By learning from snapshots drawn from the level set in either case, we are able to predict mixtures that match the true Bayes mixture $f^*([x])$.

scenarios above we were in. Standard calibration is purely a guarantee regarding total predictive uncertainty, and does not account for the source of uncertainty. Accordingly, we ask:

> *Can we build a theory of calibration that lets us decompose uncertainty into epistemic and aleatoric components and give explicit semantics for these components?*

In this work we answer this question by proposing a theory of *higher-order calibration*, a generalization of ordinary (first-order) calibration. This theory pertains to *higher-order predictors*, which predict *mixtures* of distributions over outcomes rather than just single distributions. Thus in the first scenario of the X-ray example, we might predict a mixture concentrated on the 50% distribution, while in the second we might predict an equal mixture of 100% and 0% (see Figure 1). As we shall discuss, higher-order predictors arise in many settings, including in Bayesian and ensemble methods. In a nutshell, a higher-order predictor is higher-order calibrated if whenever it predicts a certain mixture of distributions, then we do observe that mixture of distributions over outcomes in aggregate. The main contributions of our work are as follows:

1. We propose the notion of higher-order calibration, motivated by the question of rigorously decomposing predictive uncertainty into aleatoric and epistemic components. We consider standard uncertainty decompositions from the Bayesian literature (Houlsby et al., 2011; Hofman et al., 2024; Kotelevskii & Panov, 2024) and prove that under higher-order calibration, a predictor's estimate of its aleatoric uncertainty matches the real-world aleatoric uncertainty averaged over all points with the same prediction. Similarly, the epistemic uncertainty matches the true "variance" in real-world label distributions over such points.
2. We propose a tractable relaxation termed $k^{\text{th}}$-order calibration, which approaches higher-order calibration for large $k$. Our theory builds on and generalizes recent work by Johnson et al. (2024), which studied the $k = 2$ case. We show that for certain entropy functions, even small values of $k$ (like 2) yield provable uncertainty decomposition.
3. We give practical methods for achieving higher-order calibration using access to $k$-snapshots, namely examples with $k$ independent conditional labels per instance, by leveraging connections to the problem of mixture learning (Li et al., 2015).
4. We verify that our methods yield useful decompositions in real image classification tasks.

**Main result: provable uncertainty decomposition.** Suppose nature generates random labeled data $(\boldsymbol{x}, \boldsymbol{y}) \in \mathcal{X} \times \mathcal{Y}$, where $\mathcal{X}$ is the instance space and $\mathcal{Y}$ the label space, which in this paper we assume is discrete. Let $\Delta\mathcal{Y}$ be the space of probability distributions over $\mathcal{Y}$. Let $f^* : \mathcal{X} \to \Delta\mathcal{Y}$ map each instance $x$ to the conditional distribution of $\boldsymbol{y} \mid (\boldsymbol{x} = x)$. We make no assumptions about the marginal distribution or the form of $f^*$. Thus each $x$ has true aleatoric uncertainty $\mathrm{AU}^*(x) = H(f^*(x))$, where $H$ is a measure of variability or entropy. Epistemic uncertainty, by contrast, will emerge as a property of our predictor. We denote random variables using boldface throughout. See Appendix A for an extended discussion of notation.

**Definition 1.1** (Higher-order predictors and calibration). A *higher-order predictor* $f : \mathcal{X} \to \Delta\Delta\mathcal{Y}$ is a function that maps each instance to a distribution over $\Delta\mathcal{Y}$ (the space of such mixtures is denoted $\Delta\Delta\mathcal{Y}$). Let $[x] = \{x' \in \mathcal{X} : f(x') = f(x)\}$ denote the level set of $f$ that $x$ lies in. Let the *Bayes mixture* $f^*([x]) \in \Delta\Delta\mathcal{Y}$ be the distribution of $f^*(\boldsymbol{x}) \mid (\boldsymbol{x} \sim [x])$, where $\boldsymbol{x} \sim [x]$ denotes a draw from the marginal distribution $\mathcal{D}_{\mathcal{X}}$ restricted to $[x]$. We say $f$ is *higher-order calibrated* if for every $x$, $f(x) = f^*([x])$.

Higher-order calibration guarantees that our prediction $f(x)$ is a proxy for the true "Bayes mixture" of ground truth distributions $f^*(\boldsymbol{x})$ as we range over $\boldsymbol{x} \sim [x]$. This implies that any quantity defined in terms of the predicted mixture has a "real-world" interpretation as the corresponding quantity defined in terms of the Bayes mixture. This turns out to be a powerful guarantee.

We now consider the well-studied "mutual information decomposition" of predictive uncertainty (see *e.g.* Houlsby et al. (2011); Gal (2016)), and see what semantics it has under higher-order calibration. We state it in terms of the Shannon entropy $H : \Delta\mathcal{Y} \to \mathbb{R}$, given by $H(p) = -\sum_{y \in \mathcal{Y}} p_y \log p_y$. Let $\pi = f(x) \in \Delta\Delta\mathcal{Y}$ be the predicted mixture at an instance $x$. The induced marginal distribution of $\boldsymbol{y}$ when we draw $\boldsymbol{p} \sim \pi$ at random and then draw $\boldsymbol{y} \sim \boldsymbol{p}$ is given by $\overline{p} = \mathbb{E}[\boldsymbol{p}]$, the centroid of $\pi$. The *mutual information decomposition* is as follows[1]:

$$\underbrace{H(\overline{p})}_{\text{Predictive uncertainty } \mathrm{PU}(f:x)} = \underbrace{\mathbb{E}_{\boldsymbol{p}\sim\pi}[H(\boldsymbol{p})]}_{\text{Aleatoric uncertainty estimate } \mathrm{AU}(f:x)} + \underbrace{H(\overline{p}) - \mathbb{E}_{\boldsymbol{p}\sim\pi}[H(\boldsymbol{p})]}_{\text{Epistemic uncertainty } \mathrm{EU}(f:x)} \qquad (1.1)$$

**Theorem 1.2** (Uncertainty decomposition under higher-order calibration). *Suppose $f$ is higher-order calibrated. Let $\pi^* = f^*([x])$ be the Bayes mixture over the level set $[x]$, and let $\overline{p}^* = \mathbb{E}_{\boldsymbol{p}^*\sim\pi^*}[\boldsymbol{p}^*]$. Then*

- *The aleatoric uncertainty estimate is accurate on average:*

$$\mathrm{AU}(f : x) = \mathbb{E}_{\boldsymbol{p}\sim\pi}[H(\boldsymbol{p})] = \mathbb{E}_{\boldsymbol{p}^*\sim\pi^*}[H(\boldsymbol{p}^*)] = \mathbb{E}_{\boldsymbol{x}\sim[x]}[\mathrm{AU}^*(\boldsymbol{x})].$$

- *The epistemic uncertainty is exactly the average divergence of $\boldsymbol{p}^* \sim \pi^*$ to the mean $\overline{p}_*$:*

$$\mathrm{EU}(f : x) = \mathbb{E}_{\boldsymbol{p}\sim\pi}[D_{\mathrm{KL}}(\boldsymbol{p} \parallel \overline{p})] = \mathbb{E}_{\boldsymbol{p}^*\sim\pi^*}[D_{\mathrm{KL}}(\boldsymbol{p}^* \parallel \overline{p}^*)].$$

## 1.1 Learning from snapshots and $k^{\text{TH}}$-order calibration

Theorem 1.2 highlights the power of higher-order calibration. But how do we achieve it or even check if a predictor satisfies it? The key difficulty is that for any $x$, we never see $f^*(x)$ explicitly, but only get samples drawn from it.

It turns out that checking high-order calibration requires going beyond the standard learning model. To see why, returning to our X-ray example, let $x$ be the given scan, and let $[x]$ be all scans that also get a similar 50% prediction. Assume that $[x]$ is large so that we never see the same image twice. In scenario 1, the set $[x]$ consists entirely of genuinely ambiguous scans. In scenario 2, $[x]$ consists of an equal balance of unambiguously positive and negative cases. Then if all we receive are ordinary labeled examples with a single label per image, we see a 50-50 distribution of labels in both scenarios 1 and 2 and have no hope of telling them apart.[2]

Suppose instead that for each image $x$, we receive 2 labels, each of them drawn independently from $f^*(x)$ (*e.g.* from consulting two independent doctors). A little thought reveals that we can now tell the two scenarios apart quite easily. Scenario 1 results in pairs of labels distributed uniformly over the set $\{00, 01, 10, 11\}$, whereas scenario 2 only produces either $00$ or $11$, but never $01$ or $10$.

We formalize the above intuition by augmenting the classic learning setting: we assume that we have access to *k-snapshots*, namely independent conditional samples $\boldsymbol{y}_1, \ldots, \boldsymbol{y}_k \sim f^*(\boldsymbol{x})$ for each sample $\boldsymbol{x}$. There are many learning settings where one can expect to have access to $k$-snapshots. In crowd-sourcing settings each $\boldsymbol{x}$ can be shown to multiple experts, each of whom answers with their own label $\boldsymbol{y}_i$. We can model these $\boldsymbol{y}_i$s as independent draws from the true conditional distribution

---

[1]To avoid confusion, note that we always view $H$ as taking in an object of type $\Delta\mathcal{Y}$. When $\boldsymbol{p}$ is itself a random variable, i.e. a random distribution, $H(\boldsymbol{p})$ is also a random variable.

[2]This can be formalized using the outcome indistinguishability framework (Dwork et al., 2021).

$f^*(\boldsymbol{x})$. Here $k$ is decided by the number of experts we can call upon. A different setting is one where we want to distill a large teacher model into a smaller student model. In this case we have oracle access to the teacher and can ask for $k$-snapshots for large $k$. Of course not every setting admits such access, for instance health outcomes which by definition occur only once per patient.[3]

A $k$-snapshot of labels from $\mathcal{Y}$ can be seen as a coarse representation of a distribution in $\Delta\mathcal{Y}$ by considering its normalized histogram. A distribution over $k$-snapshots is thus a coarse representation of a mixture in $\Delta\Delta\mathcal{Y}$. By framing our goal as that of learning a $k$-snapshot predictor from $k$-snapshot examples, we obtain an effective way of learning an (approximate) higher-order predictor.

This naturally defines a hierarchy of calibration notions, which we term $k^{\text{th}}$-order calibration, that ranges from ordinary first-order calibration when $k = 1$ to (full) higher-order calibration as $k \to \infty$. Our work builds on and generalizes the elegant recent work of Johnson et al. (2024), who consider the case of $k = 2$; we discuss this further in related work. Moreover, $k^{\text{th}}$-order calibration essentially reduces to ordinary first-order calibration over the extended label space of $k$-snapshots; the only subtlety is in the choice of metric over this space. This leads to a simple and general recipe: we can leverage any general-purpose procedure guaranteeing first-order calibration over this extended label space to achieve $k^{\text{th}}$-order calibration. We prove formally that $k^{\text{th}}$-order calibration converges to higher-order calibration at a $1/\sqrt{k}$ rate (see Theorem 2.4).

We also provide a different way of achieving $k^{\text{th}}$-order calibration using a purely *post-hoc* routine, analogous to first-order calibration procedures such as Platt scaling. Suppose we start from an ordinary first-order calibrated model. We can consider its level set partition $[\cdot]$ (or more generally any desired partition $[\cdot]$ of the space, with $[x]$ denoting the equivalence class of $x$). Now, higher-order calibration with respect to $[\cdot]$ entails exactly the following: for any given equivalence class $[x]$, we need to learn the true Bayes mixture $f^*([x])$. Thus we have reduced our problem to a collection of pure *mixture learning* problems. This problem has been studied before in the theoretical computer science literature (Li et al., 2015). We leverage these ideas to provide a simple post-hoc $k^{\text{th}}$-order calibration procedure whose sample complexity scales as $|\mathcal{Y}|^k$ (see Theorem 2.6).

Importantly, for common choices of entropy functions (like Brier entropy), we show in Theorem 3.3 that $k^{\text{th}}$-order calibration for small $k$ ($k = 2$ for Brier) suffices to approximate the uncertainty decomposition guarantees from Theorem 1.2. This relies on a key property of $k^{\text{th}}$-order calibration that we prove: it gives good estimates of the first $k$ moments of the Bayes mixture over the level sets (Theorem 2.5). $k^{\text{th}}$-order calibration is thus a natural goal in itself for uncertainty decomposition.

## 1.2 RELATED WORK

**Bayesian methods and mixture-based uncertainty decompositions.** The predominant modern approach to uncertainty decomposition has been the Bayesian one (see e.g. Gal (2016); Mena et al. (2021)), wherein one uses Bayesian inference to obtain for any given instance a full posterior distribution over a family of models, or equivalently over corresponding distributions of outcomes. In this way every Bayesian model is a higher-order predictor. A similar logic holds for ensemble models (Gal & Ghahramani, 2016; Lakshminarayanan et al., 2017).

A number of works in the Bayesian literature have studied uncertainty decompositions based on predicted mixtures (sometimes termed second-order or higher-order distributions) (Houlsby et al., 2011; Depeweg et al., 2018; Hüllermeier & Waegeman, 2021; Malinin & Gales, 2021; Wimmer et al., 2023; Schweighofer et al., 2023; Sale et al., 2023c;a; Kotelevskii & Panov, 2024; Hofman et al., 2024). Bayesian neural networks (Lampinen & Vehtari, 2001) and variants (Malinin & Gales, 2018; Sensoy et al., 2018; Osband et al., 2023) can also be seen as explicit constructions of certain types of parameterized higher-order predictors. We build on these works, and in particular on Hofman et al. (2024); Kotelevskii & Panov (2024), for our analysis of uncertainty decompositions.

While the idea of predicting mixtures is already present in these works, they do not consider any formal semantics such as calibration. The real-world semantics of Bayesian methods typically arise from the following type of asymptotic consistency guarantee (see e.g. Doob's theorem (Miller, 2018)): if the model class is well-specified (i.e. the unknown $f^*$ is realizable by the class), and Bayesian inference is computationally feasible (at least in some approximate sense), then in the

---

[3]While this is a limitation of our method, it is inherent to the task — as just noted, if a predictor is unable to separate instances in a set $[x]$, it cannot distinguish between scenarios like those in the X-ray example.

limit as the sample size grows large, the posterior becomes tightly concentrated around the true $f^*$. In practice, however, it is often very difficult (or impossible) to test that the model class is indeed well-specified, as well as to perform true Bayesian inference over expressive classes. In such cases, it is not clear what semantics (if any) hold for the resulting uncertainty decomposition. This is the key gap we address with higher-order calibration. We note that higher-order calibration can treated as an evaluation metric for any higher-order predictor, even if it is misspecified or poorly fit.

**Pair prediction and second-order calibration.** The $k = 2$ case of $k^{\text{th}}$-order calibration was introduced and studied by Johnson et al. (2024). Our work builds on theirs and greatly extends their notion of second-order calibration by defining the limiting notion of higher-order calibration that is implicit in their work. This allows us to view $k^{\text{th}}$-order calibration as defining a natural hierarchy between first-order and higher-order calibration. We believe this considerably clarifies the conceptual principles at play and also provides a bridge to the Bayesian literature. We draw on recent work on calibration (Błasiok et al., 2023; Gopalan et al., 2024b;a) to provide a full-fledged theory of higher-order calibration error and more. The idea of joint prediction has separately been studied by Wen et al. (2021); Osband et al. (2023); Durasov et al. (2024); Lee et al. (2024).

Please see Appendix B for a full discussion of additional related work.

## 2 Higher-order calibration

In this section we define higher-order and $k^{\text{th}}$-order calibration in full generality, and describe methods for achieving them.Recall that a *partition* of the space $\mathcal{X}$ is a collection of disjoint subsets, or equivalence classes, whose union is the entire space. For any $x \in \mathcal{X}$, we use $[x] \subseteq \mathcal{X}$ to refer to its equivalence class, and we use $[\cdot]$ as shorthand for the entire partition. The *Bayes mixture* over an equivalence class $[x]$, denoted $f^*([x])$, is the mixture obtained by drawing $\boldsymbol{x} \sim [x]$ (according to the marginal distribution $D_{\mathcal{X}}$ restricted to $[x]$) and considering $f^*(\boldsymbol{x})$. We define approximate higher-order calibration to allow for some error between the predicted distribution and the Bayes mixture, where we measure closeness between mixtures in $\Delta\Delta\mathcal{Y}$ using the Wasserstein distance with respect to the $\ell_1$ (or statistical) distance between distributions in $\Delta\mathcal{Y}$. Recall that for us $\mathcal{Y}$ is discrete.

**Definition 2.1** (Approximate higher-order calibration). We say a higher-order predictor $f : \mathcal{X} \to \Delta\Delta\mathcal{Y}$ is $\epsilon$-higher-order calibrated to $f^*$ wrt a partition $[\cdot]$ if for all $x \in \mathcal{X}$,[4]

$$W_1(f(x), f^*([x])) = \inf_{\mu \in \Gamma(f(x), f^*([x]))} \mathbb{E}_{(\boldsymbol{p}, \boldsymbol{p}^*) \sim \mu}[\|\boldsymbol{p} - \boldsymbol{p}^*\|_1] \le \epsilon,$$

where we use $\Gamma(\pi, \pi')$ to denote the set of all couplings of $\pi$ and $\pi'$.

Even with this relaxation, it is unclear how to measure higher-order calibration error, since we never get access to $f^*$ itself. As a tractable path towards this goal, we will introduce the model of learning from $k$-snapshots and $k^{\text{th}}$-order calibration.

**Definition 2.2** ($k$-snapshots and $k^{\text{th}}$-order projections). A $k$-snapshot for $x \in \mathcal{X}$ is a tuple $\boldsymbol{s} = (\boldsymbol{y}_1, \ldots, \boldsymbol{y}_k) \in \mathcal{Y}^k$ where each label $\boldsymbol{y}_i$ is drawn independently from $f^*(x)$. For a mixture $\pi \in \Delta\Delta\mathcal{Y}$, its $k^{\text{th}}$-order projection is the $k$-snapshot distribution $\text{proj}_k(\pi) \in \Delta\mathcal{Y}^{(k)}$ defined as follows:

- We draw $\boldsymbol{p} \sim \pi$ and a $k$-snapshot $\boldsymbol{s} = (\boldsymbol{y}_1, ..., \boldsymbol{y}_k)$ by sampling $k$ times i.i.d. from $\boldsymbol{p}$.
- We output $\text{Unif}(\boldsymbol{y}_1, \ldots, \boldsymbol{y}_k)$.

Here $\mathcal{Y}^{(k)}$ denotes the space of such uniform distributions over snapshots (aka normalized histograms). Thus $\mathcal{Y}^{(k)} \subseteq \Delta\mathcal{Y}$ is a coarsened version (with granularity $1/k$) of the space $\Delta\mathcal{Y}$.

In the model of learning with $k$-snapshots, we receive pairs $(\boldsymbol{x}, \boldsymbol{s})$ where $\boldsymbol{x} \sim D_{\mathcal{X}}$ and $\boldsymbol{s} \in \mathcal{Y}^k$ is a $k$-snapshot for $\boldsymbol{x}$, and the goal is to learn a "$k$-snapshot predictor" that predicts a distribution over possible $k$-snapshots drawn from a given $x \in \mathcal{X}$. While this distribution could be represented directly over $\mathcal{Y}^k$, we can simplify our approach by leveraging a key property of $k$-snapshots: the $\boldsymbol{y}_i$s are each drawn iid. Consequently, the order of the tuple does not matter. Thus, instead of viewing a $k$-snapshot as a tuple $(y_1, ..., y_k) \in \mathcal{Y}^k$, we will view it as the distribution $\text{Unif}(y_1, ..., y_k) \in \mathcal{Y}^{(k)} \subseteq$

---

[4]Throughout this paper, for simplicity we require closeness for all equivalence classes $[x]$. A more relaxed notion would only require it to hold with high probability over equivalence classes.

$\Delta \mathcal{Y}$. The goal of $k$-snapshot prediction is to learn a predictor $g : \mathcal{X} \rightarrow \Delta \mathcal{Y}^{(k)}$ such that for each $x \in \mathcal{X}$, $g(x)$ is close to $\text{proj}_k(f^*(x))$, the true distribution of $k$-snapshots for $x$.

Viewing the space $\mathcal{Y}^{(k)}$ as a subset of $\Delta \mathcal{Y}$, and hence the space $\Delta \mathcal{Y}^{(k)}$ as a subset of $\Delta \Delta \mathcal{Y}$, is an important conceptual simplification. Firstly, because $g(x) \in \Delta \mathcal{Y}^{(k)} \subseteq \Delta \Delta \mathcal{Y}$, a $k$-snapshot predictor can be directly viewed as a higher-order predictor with no extra effort. Secondly, it suggests a non-obvious error metric for $k$-snapshot prediction which turns out to be the right one: we simply use the Wasserstein distance on $\Delta \Delta \mathcal{Y}$ as our distance measure between two distributions over snapshots. (This is a better measure than obvious choices like viewing $\mathcal{Y}^{(k)}$ as a discrete space and using statistical distance. See Appendix A for some more discussion of alternative representations.) We define calibration for $k$-snapshot prediction analogously to the higher-order setting:

**Definition 2.3** (Approximate $k^{\text{th}}$-order calibration). We say a $k$-snapshot predictor $g : \mathcal{X} \rightarrow \Delta \mathcal{Y}^{(k)}$ is $\epsilon$-$k^{\text{th}}$-order calibrated wrt $f^*$ and a partition $[\cdot]$ if for all $x \in \mathcal{X}$, $W_1(g(x), \text{proj}_k f^*([x])) \leq \epsilon$.

### 2.1 KEY PROPERTIES OF $k^{\text{TH}}$-ORDER CALIBRATION

We highlight three advantages of $k^{\text{th}}$-order calibration as a relaxation of higher-order calibration.

**Tractability.** Verifying higher-order calibration for a predictor requires knowledge of $f^*$, which is generally unknown. In contrast, $k^{\text{th}}$-order calibration is defined relative to the $k^{\text{th}}$-order projection of $f^*$ on each partition. This projection corresponds exactly to the distribution we observe when collecting $k$-snapshots as data. Thus, $k^{\text{th}}$-order calibration can be easily verified by comparing the empirical distribution of $k$-snapshot data against the predictor's output distribution. This makes $k^{\text{th}}$-order calibration a practical metric for assessing predictor performance. Beyond verification, $k^{\text{th}}$-order calibration also comes with simple methods for achieving it, which we detail in Section 2.2.

**Convergence to higher-order calibration.** As one might expect, $k^{\text{th}}$-order calibration for large $k$ is closely related to higher-order calibration. In particular, we can precisely characterize the rate at which $k^{\text{th}}$-order calibration approaches higher-order calibration as $k \rightarrow \infty$. The proof can be found in Appendix D.3, and follows via a concentration argument:

**Theorem 2.4** ($k^{\text{th}}$-order calibration implies higher-order calibration.[5]). *If $g : \mathcal{X} \rightarrow \Delta \mathcal{Y}^{(k)}$ is $\epsilon$-$k^{\text{th}}$-order calibrated with respect to a partition $[\cdot]$, then it is also $(\epsilon + |\mathcal{Y}|/(2\sqrt{k}))$-higher-order calibrated with respect to the same partition $[\cdot]$.*

This shows the convergence of approximate $k^{\text{th}}$-order calibration to approximate higher-order calibration in a way that preserves the approximation error.

**Moment recovery via $k^{\text{th}}$-order calibration.** At small values of $k$, $k^{\text{th}}$-order calibration can still guarantee valuable information about the mixture distribution over each partition, if not about the higher-order calibration of a predictor. In particular, a $k^{\text{th}}$-order calibrated predictor gives good estimates of the first $k$ moments of $f^*([x])$. This should be contrasted with higher-order calibration, which tells us the full distribution of $f^*([x])$. In settings where we only care about the first $k$ moments of $f^*([x])$, this tells us that $k^{\text{th}}$-order calibration can substitute for higher-order calibration. For simplicity, we restrict to the binary case, where $\Delta \Delta \mathcal{Y}$ can be identified with $\Delta[0, 1]$.

**Theorem 2.5** (Moment estimates from $k^{\text{th}}$-order calibration). *Let $\mathcal{Y} = \{0, 1\}$, and let $g : \mathcal{X} \rightarrow \Delta \mathcal{Y}^{(k)}$ be $\epsilon$-$k^{\text{th}}$-order calibrated with respect to a partition $[\cdot]$. Then for each $x \in \mathcal{X}$, we can use $g(x)$ to generate a vector of moment estimates $(m_1, \ldots, m_k) \in \mathbb{R}^n$ such that for each $i \in [k]$,*

$$\left| m_i - \mathop{\mathbb{E}}_{\boldsymbol{x} \sim [x]}[f^*(\boldsymbol{x})^i] \right| \leq i\epsilon/2.$$

This is proved in Appendix D.4 by showing that the first $k$ moments of a mixture can be exactly recovered from the mixture's $k^{\text{th}}$-order projection. This allows us to get a calibrated vector of predictions for the first $k$ moments from a $k^{\text{th}}$-order calibrated predictor. It implies that we can make calibrated predictions for the expectation of any low-degree polynomial with bounded coefficients evaluated on $f^*(x)$. This property will be useful for estimating aleatoric uncertainty using $k^{th}$-order calibration (Theorems 3.3, E.8) and giving prediction sets with coverage guarantees (Appendix F).

---

[5]We present this theorem in a simplified manner to emphasize the connections between $k^{\text{th}}$- and higher-order calibration error. See Appendix D.2 for additional guarantees that both lower- and upper- bound the higher-order calibration error within a partition in terms of the $k^{\text{th}}$-order error.

## 2.2 ACHIEVING $k^{\text{TH}}$-ORDER CALIBRATION

We present two simple methods for $k^{\text{th}}$-order calibration based on common approaches for first-order calibration: minimizing a proper loss, or post-processing the level sets of a learned predictor.

**Learning directly from snapshots.** Our definition of $k^{\text{th}}$-order calibration is immediately accompanied by a natural method for achieving it: view the learning problem as a multiclass classification problem over the extended label space $\mathcal{Y}^{(k)}$ of $k$-snapshots, and use any off-the-shelf procedure that tries to achieve ordinary first-order calibration over this space when we draw true $k$-snapshots from nature. Perfect first-order calibration over $\mathcal{Y}^{(k)}$ means exactly that $g(x) = \text{proj}_k f^*([x])$ for all $x$, which is equivalent to $k^{\text{th}}$-order calibration. A simple concrete method is to minimize a proper multiclass classification loss over the label space $\mathcal{Y}^{(k)}$ (such as cross entropy) over an expressive function class. This is the method followed in Johnson et al. (2024) for achieving second-order calibration.

**Post-hoc calibration using snapshots.** Here we describe a simple alternative way of achieving $k^{\text{th}}$-order calibration using a pure post-processing routine which allows us to turn any ordinary first-order predictor into a higher-order calibrated predictor, thereby "eliciting" its epistemic uncertainty. This method requires only a calibration set of $k$-snapshots rather than a full training set.

Let us briefly recall the standard post-processing procedure for achieving first-order calibration.[6] For a partition $[\cdot]$, first-order calibration requires that for every $x$, $f(x) = \mathbb{E}_{\boldsymbol{x} \sim [x]}[f^*(\boldsymbol{x})]$. If this is not the case, then we "patch" the prediction at $x$ to be the centroid of the Bayes mixture within its partition, $\mathbb{E}_{\boldsymbol{x} \sim [x]}[f^*(\boldsymbol{x})]$. That is, we have reduced the problem to that of *learning the mean label distribution* over a given set $[x]$. We implement this by drawing a sufficiently large sample from each set $[x]$ in the partition and use the empirical distribution as an estimate of the true mean.

Our procedure for post-hoc $k^{\text{th}}$-order calibration is a very natural extension of this algorithm. Let a partition $[\cdot]$ be given (perhaps arising from an initial first-order predictor). For each equivalence class $[x]$ in the partition, instead of simply trying to estimate the centroid of the Bayes mixture, we will now try to learn the entire Bayes mixture (or technically its $k^{\text{th}}$-order projection $\text{proj}_k f^*([x])$). That is, we reduce to *mixture learning* instead of mere distribution learning. Mixture learning has been previously explored in the literature; e.g. Li et al. (2015) give algorithms for learning mixtures given access to snapshots. While their methods could be applied for post-processing for higher-order calibration, their focus on learning complete mixtures rather than their $k^{\text{th}}$-order projections (as is sufficient for $k^{\text{th}}$-order calibration) leads to different guarantees and snapshot size requirements.

We present a simple post-processing algorithm for achieving $k^{\text{th}}$-order calibration using a calibration set of $k$-snapshots from each $[x]$. Our algorithm is as follows: given $N$ $k$-snapshots — viewed as distributions $\boldsymbol{p}_1, ..., \boldsymbol{p}_N \in \mathcal{Y}^{(k)} \subseteq \Delta\mathcal{Y}$ — drawn from $\text{proj}_k f^*([x])$, output the empirical mixture, i.e. $\text{Unif}(\boldsymbol{p}_1, ..., \boldsymbol{p}_N)$. We can use standard concentration arguments to show that for sufficiently large $N$, this will give a good estimate of $\text{proj}_k f^*([x])$ and hence a $k^{\text{th}}$-order calibrated predictor.

**Theorem 2.6** (Empirical estimate of $k^{\text{th}}$-order projection guarantee). *Consider any $x \in \mathcal{X}$, and a sample $\boldsymbol{p}_1, ..., \boldsymbol{p}_N \in \mathcal{Y}^{(k)}$ where each $\boldsymbol{p}_i$ is drawn i.i.d. from $\text{proj}_k f^*([x])$. Given $\epsilon > 0$ and $0 \leq \delta \leq 1$, if $N \geq (2(|\mathcal{Y}^{(k)}| \log(2) + \log(1/\delta)))/\epsilon^2$, then we can guarantee that with probability at least $1 - \delta$ over the randomness of the sample we will have*

$$W_1(\text{proj}_k f^*([x]), \text{Unif}(\boldsymbol{p}_1, ..., \boldsymbol{p}_N)) \leq \epsilon.$$

The proof of this theorem can be found in Appendix D.5. We note that $|\mathcal{Y}^{(k)}|$ is exactly the number of multisets of size $k$ that can be chosen from $\ell$ items, with repeats. Thus, $|\mathcal{Y}^{(k)}| = \binom{k+\ell-1}{\ell-1} \leq \ell^k$ and so replacing $|\mathcal{Y}^{(k)}|$ with $\ell^k$ also gives the desired guarantee. In the special case of binary prediction, $|\mathcal{Y}^{(k)}|$ simplifies to just $k + 1$, resulting in a bound that is linear in the size of the snapshot.

## 3 UNCERTAINTY DECOMPOSITIONS

Having defined higher-order and $k^{\text{th}}$-order calibration, we show how these notions can be leveraged to provide meaningful semantics for decomposing predictive uncertainty. Let any concave general-

---

[6]Here we mean distribution calibration with respect to distributions in $\Delta\mathcal{Y}$ for any discrete space $\mathcal{Y}$, but the reader can consider the binary case for simplicity. There, distribution learning reduces to mean estimation.

ized entropy function $G : \Delta\mathcal{Y} \to \mathbb{R}$ be given (*e.g.*, the Shannon entropy; see Appendix C.1 for more details). Fix a point $x \in \mathcal{X}$, and let the predicted mixture at $x$ be $f(x) \in \Delta\Delta\mathcal{Y}$. Consider a random distribution $\boldsymbol{p} \sim f(x)$ drawn from this mixture, viewed as a random vector on the simplex in $\mathbb{R}^\ell$. The induced marginal distribution of $\boldsymbol{y}$ when we draw $\boldsymbol{p} \sim f(x)$ and then draw $\boldsymbol{y} \sim \boldsymbol{p}$ is given by $\bar{p} = \mathbb{E}[\boldsymbol{p}]$, the centroid of $f(x)$.[7] Perhaps the most studied decomposition is the "mutual information decomposition" (Houlsby et al., 2011; Gal, 2016), which we restate here. Our formulation in terms of generalized entropy follows that of Hofman et al. (2024); Kotelevskii & Panov (2024).[8]

$$\underbrace{G(\bar{p})}_{\text{Predictive uncertainty}} = \underbrace{\mathbb{E}[G(\boldsymbol{p})]}_{\text{Aleatoric uncertainty estimate}} + \underbrace{G(\bar{p}) - \mathbb{E}[G(\boldsymbol{p})]}_{\text{Epistemic uncertainty}} \qquad (3.1)$$

The predictive uncertainty is the total entropy in a random outcome $\boldsymbol{y}$ drawn marginally from $\pi$. The aleatoric uncertainty estimate is the conditional entropy in $\boldsymbol{y}$ given the mixture component $\boldsymbol{p}$. The epistemic uncertainty is precisely the mutual information between $\boldsymbol{y}$ and $\boldsymbol{p}$; it is non-negative by Jensen's inequality. It can be equivalently written as a variance-like quantity, $\mathbb{E}_{\boldsymbol{p} \sim \pi}[D(\boldsymbol{p} \,\|\, \bar{p})]$, where $D$ is the divergence associated with $G$ (e.g., the KL divergence for the Shannon entropy). Accordingly we define the following estimates for uncertainty.

**Definition 3.1.** Let $G : \Delta\mathcal{Y} \to \mathbb{R}$ be a concave generalized entropy function. For a higher-order predictor $f : \mathcal{X} \to \Delta\Delta\mathcal{Y}$, we define the predictive, aleatoric, and epistemic uncertainties of $f$ at $x$ with respect to $G$ as

$$\mathrm{PU}_G(f : x) = G\big(\underset{\boldsymbol{p} \sim f(x)}{\mathbb{E}}[\boldsymbol{p}]\big)$$

$$\mathrm{AU}_G(f : x) = \underset{\boldsymbol{p} \sim f(x)}{\mathbb{E}}[G(\boldsymbol{p})]$$

$$\mathrm{EU}_G(f : x) = \mathrm{PU}_G(f : x) - \mathrm{AU}_G(f : x).$$

These quantities are meaningful mainly for higher-order predictors. Nature itself only associates a pure distribution $f^*(x)$ with every $x$, and has only true aleatoric uncertainty: $\mathrm{AU}_G^*(x) = G(f^*(x))$.

The mutual information decomposition (Eq. (3.1)) is purely a function of our prediction and independent of nature. Under what conditions does it tell us something meaningful about true aleatoric uncertainty? It turns out that higher-order calibration is a sufficient condition: it guarantees that our estimate of aleatoric uncertainty at a point $x$ equals the average *true* aleatoric uncertainty over $[x]$.

**Lemma 3.2.** *If $f : \mathcal{X} \to \Delta\Delta\mathcal{Y}$ is perfectly higher-order calibrated, then for all $x \in \mathcal{X}$,*

$$\mathrm{AU}_G(f : x) = \underset{\boldsymbol{x} \sim [x]}{\mathbb{E}}[\mathrm{AU}_G^*(\boldsymbol{x})].$$

This is proved in Appendix E and establishes the first part of Theorem 1.2. At first, higher-order calibration may seem like a very strong condition that just happens to provide calibrated uncertainty estimates. In fact, Theorem E.2 will show that higher-order calibration is a *necessary* condition for a predictor to produce calibrated estimates of aleatoric uncertainty with respect to all concave entropy functions. In this way we see that higher-order calibration identifies a predictor's epistemic uncertainty in bucketing multiple potentially different points into a single equivalence class $[x]$. Our estimate of aleatoric uncertainty captures the average true aleatoric uncertainty in $f^*(\boldsymbol{x})$ as $\boldsymbol{x} \sim [x]$, and our epistemic uncertainty can be thought of as the true "variance"[9] in $f^*(\boldsymbol{x})$ as $\boldsymbol{x} \sim [x]$.

**Uncertainty decomposition from $k^{\text{th}}$-order calibration.** While Lemma 3.2 draws an important connection between higher-order calibration and uncertainty estimation, it does not allow for calibration error. Even if it did, achieving sufficiently small higher-order calibration error might require $k^{\text{th}}$-order calibration for large $k$ by Theorem 2.4. What if we can only guarantee $k^{\text{th}}$-order calibration for reasonably small $k$? Is this of any use towards the goal of rigorous uncertainty decomposition?

---

[7]In the context of Bayesian models, $f(x)$ is often called the posterior predictive distribution, and $\bar{p}$ is often called the Bayesian model average (BMA).

[8]In general there are multiple approaches to uncertainty decomposition given a predicted mixture. Here we focus on the mutual information decomposition. In Appendix C we discuss other natural mixture-based decompositions and show how they can also be endowed with meaningful semantics via higher-order calibration.

[9]In particular, it is the divergence of $f^*(\boldsymbol{x})$ for $\boldsymbol{x} \sim [x]$ from the centroid $\mathbb{E}_{\boldsymbol{x} \sim [x]}[f^*(\boldsymbol{x})]$, as formalized in Theorem 1.2 and more generally in Lemma C.1.

We show that for the two most commonly used entropy functions, the binary Brier and Shannon entropies, the answer is yes. In fact, the former only requires $k = 2$. We prove a general result which applies to any uniformly continuous entropy function. The key insight is that $k^{\text{th}}$-order calibration provides us with good estimates of the first $k$ moments of the Bayes mixture (Theorem 2.5). The Brier entropy is a degree 2 polynomial, whereas we show that the Shannon entropy has good approximations by low-degree polynomials. A similar guarantee applies to all uniformly continuous functions via Jackson's theorem. Taken together, these results show that $k^{\text{th}}$-order calibration is a natural and tractable goal in itself for uncertainty decomposition. We state the following theorem informally; formal statements and proofs (with sample complexity) may be found in Appendix E.3.

**Theorem 3.3** (Estimating entropy functions; informal)**.** *We can obtain estimates* $\widehat{\text{AU}}_G$ *satisfying*

$$\left| \widehat{\text{AU}}_G - \mathbb{E}_{\boldsymbol{x} \sim [x]}[\text{AU}_G^*(\boldsymbol{x})] \right| \leq \epsilon$$

*using only $k^{th}$-order calibration for the following common entropy functions G:*

- *For Brier entropy $G_{Brier}(p) = 4p(1-p)$ using $(\epsilon/8)$-second-order calibration;*

- *For Shannon entropy $G_{Shannon}(p) = -p \log p - (1-p) \log(1-p)$ using $\epsilon'$-$k^{th}$-order calibration, where $\epsilon' \leq \epsilon / \exp(\Theta(k))$ and $k \geq \Theta((1/\epsilon)^{\ln 4})$;*

- *For any uniformly continuous G with modulus of continuity $\omega_G$ (see Definition E.3) using $\epsilon'$-$k^{th}$-order calibration where $\epsilon'$ satisfies $\omega_G(\epsilon' + |\mathcal{Y}|/(2\sqrt{k})) \leq \epsilon$.*

## 4 EXPERIMENTS

In this section, we describe experiments on training higher-order calibrated models using $k$-snapshots. We focus on the task of classifying ambiguous images using CIFAR-10H (Peterson et al., 2019), a relabeling of the test set of CIFAR-10 (Krizhevsky, 2009). For additional experiments, including comparisons with other uncertainty decomposition methods, see Appendix G.

CIFAR-10H is an image recognition dataset of 10,000 $32 \times 32$ color images with 10 possible classes (*e.g.* truck, deer) and with at least 50 independent human annotations per image. Thus $\mathcal{Y}$ is the space of 10 possible entities, and for each image $x \in \mathcal{X}$, we treat the normalized histogram of its $\geq 50$ independent ground-truth annotations as a distribution $f^*(x)$ over $\mathcal{Y}$. In other words, $f^*(x) \in \Delta \mathcal{Y}$ is the uniform distribution over the independent annotations of $x$. All entropy computations are done with respect to this distribution. There is disagreement among annotators: the mean true aleatoric uncertainty $\mathbb{E}[\text{AU}_H^*(\boldsymbol{x})] = \mathbb{E}[H(f^*(\boldsymbol{x}))]$ is 0.17 (STD 1.2), where $H$ is the Shannon entropy.

We first train a regular 1-snapshot ResNet (Zagoruyko & Komodakis, 2016) on 45,000 images from the CIFAR training set, setting aside the remaining 5,000 for validation. We then apply our post-hoc calibration algorithm (see Section 2.2) to CIFAR-10H, using half as a calibration set and the other half as a test set. For more details, see Appendix G. In this way we obtain a $k^{\text{th}}$-order predictor $g : \mathcal{X} \to \Delta \mathcal{Y}^{(k)}$. The induced marginal first-order predictor $\overline{g} : \mathcal{X} \to \Delta \mathcal{Y}$ is $\overline{g}(x) = \mathbb{E}_{\boldsymbol{p} \sim g(x)}[\boldsymbol{p}]$.

In this high-dimensional setting, it is infeasible to compute the true level set partition of $g$. We instead use a coarse-grained partition defined in terms of $\overline{g}$. For each possible class (e.g. truck), we bin together all images $x$ where $\overline{g}(x)$ takes its maximum value at that class and where that value lies in one of 10 slices of $[0, 1]$ (e.g. $[0.8, 0.9]$). Thus we obtain a partition with 100 bins in total.

In Appendix G we report higher-order calibration error as measured by Wasserstein error $\epsilon$ in Definition 2.3. As a more easily interpretable measure, we also compute the aleatoric estimation error:

$$E_{\text{AU}}(x) = \left| \text{AU}_H(g : x) - \mathbb{E}_{\boldsymbol{x} \sim [x]}[\text{AU}_H^*(\boldsymbol{x})] \right| \tag{4.1}$$

In Figure 2, we report the mean aleatoric estimation error over all images, $\mathbb{E}_{\boldsymbol{x}}[E_{\text{AU}}(\boldsymbol{x})]$. As expected, predictions of aleatoric uncertainty grow more accurate as the size of the snapshots used to calibrate the model is increased, and there is a significant benefit to going beyond $k = 2$.

To illustrate the advantages of accurate estimates of aleatoric uncertainty, we include in Figure 3 examples of images skewed most towards pure "epistemic" or "aleatoric" uncertainty. The former

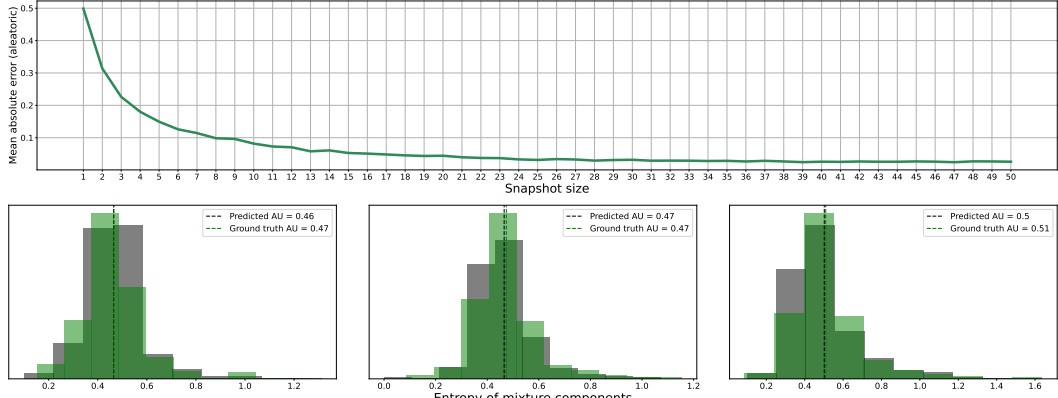

Figure 2: **Calibrating models with $k$-snapshots yields accurate uncertainty decompositions.** *Top:* Average aleatoric uncertainty estimation error (Eq. (4.1)) of calibrated CIFAR-10 models. *Bottom:* For three of the highest-entropy equivalence classes, we depict the distribution of entropies ranging over the predicted mixture (gray) and the Bayes mixture (green). We see that the distributions and in particular the means are similar.

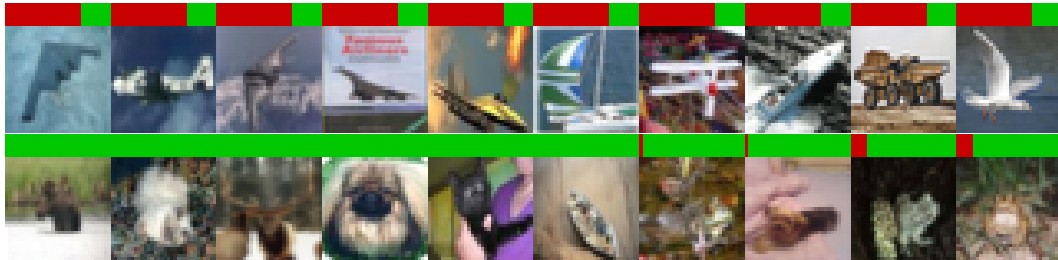

Figure 3: **Qualitatively, accurate estimates of aleatoric uncertainty help separate unusual, poorly learned images (mostly epistemic) from genuinely ambiguous ones (mostly aleatoric).** *Top:* CIFAR-10H images with the highest ratio of epistemic uncertainty to aleatoric uncertainty (depicted by colored bars), as estimated by a well-higher-order-calibrated model. *Bottom:* The most aleatoric images according to the same model.

category mainly consists of images that are unusual in some way, e.g. images of objects at unusual angles, or closely cropped images. Nevertheless, all are relatively easy for humans to identify, and thus point to cases where the model could be improved. The "aleatoric" images, on the other hand, include mostly low-quality or otherwise ambiguous pictures on which the annotators do in fact disagree. Interestingly, this section also includes a few images (e.g. the fifth image, of a `cat` being held by a human) that do not seem ambiguous per se but did in fact divide annotators. This indicates that the mixture model has learned something about aleatoric uncertainty in the label distribution not obvious to the naked eye. Such insights can help practitioners fine-tune their data collection efforts.

## 5 CONCLUSION

This work introduces higher-order calibration as a rigorous framework for decomposing predictive uncertainty with clear semantics, guaranteeing that our predictor's aleatoric uncertainty estimate matches the true average aleatoric uncertainty over points where the prediction is made. We also propose $k^{\text{th}}$-order calibration as a tractable relaxation, showing that even small values of $k$ can yield meaningful uncertainty decompositions. We demonstrate both training-time and post-hoc methods for achieving $k^{\text{th}}$-order calibration using $k$-snapshots. Our method offers significant advantages over existing methods in that it is distribution-free, tractable, and comes with strong provable guarantees.

The main limitation of our work is that achieving $k^{\text{th}}$-order calibration for any $k > 1$ fundamentally requires access multiple independent labels per data instance. While this requirement is satisfied in many scenarios, such as crowd-sourcing or model distillation, it is not always satisfiable. Additionally, our sample complexity scales with $|\mathcal{Y}|^k$, which could become prohibitive for large label spaces or larger $k$. Open questions for future research include: (a) techniques for reducing our dependence on label size and $k$, (b) applications to active learning and (c) out-of-distribution detection, and (d) formally studying the tradeoff between collecting more conditional labels versus more labeled data.

ACKNOWLEDGMENTS

We would like to thank Kunal Talwar and Moises Goldszmidt for helpful discussions. Charlotte Peale is grateful to be supported by the Apple Scholars in AI/ML PhD fellowship.

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

# Appendix

## Table of Contents

## A    Review of notation and preliminaries

**Notation.** We denote our instance space, or domain, by $\mathcal{X}$, and our outcome space by $\mathcal{Y}$. We will focus on discrete or categorical spaces $\mathcal{Y} = \{0, \ldots, \ell - 1\}$. We denote the space of probability distributions on $\mathcal{Y}$ by $\Delta\mathcal{Y}$, and the space of (higher-order) distributions on $\Delta\mathcal{Y}$, or mixtures, by $\Delta\Delta\mathcal{Y}$. The space $\Delta\mathcal{Y}$ can be identified with the $(\ell - 1)$-dimensional simplex, denoted $\Delta_\ell$ and viewed as a subset of $\mathbb{R}^\ell$. We will often think of an element of $\Delta\mathcal{Y}$ as a vector in $\Delta_\ell \subset \mathbb{R}^\ell$ (namely with $\ell$ nonnegative entries summing to one). It will often be illustrative to consider the binary case $\mathcal{Y} = \{0, 1\}$, where we can identify $\Delta\mathcal{Y}$ with a single parameter in $[0, 1]$, namely the bias, or the probability placed on 1. A distribution, i.e. an element of $\Delta\mathcal{Y}$, will typically be denoted by $p$. A mixture, i.e. an element of $\Delta\Delta\mathcal{Y}$, will typically be denoted by $\pi$.

We follow the convention of denoting random variables using boldface. Thus given a distribution $p \in \Delta\mathcal{Y}$, a random outcome from this distribution will be denoted by $\boldsymbol{y} \sim p$. Similarly, given a mixture $\pi \in \Delta\Delta\mathcal{Y}$, a *random distribution* drawn from the mixture (i.e. a random mixture component drawn according to the mixture weights) will be denoted by $\boldsymbol{p} \sim \pi$.

We refer to a $k$-tuple of labels $(\boldsymbol{y}_1, \ldots, \boldsymbol{y}_k)$ drawn iid from a distribution $p \in \Delta\mathcal{Y}$ as a $k$-*snapshot*. Since the $\boldsymbol{y}_i$ are drawn iid, the order of the tuple does not matter, and we generally symmetrize the snapshot and identify it with the distribution $\mathrm{Unif}(\boldsymbol{y}_1, \ldots, \boldsymbol{y}_k)$. We denote the space of all such distributions or symmetrized $k$-snapshots by $\mathcal{Y}^{(k)}$.[10] Note that $\mathcal{Y}^{(k)} \subseteq \Delta\mathcal{Y}$. The space of distributions over symmetrized $k$-snapshots will be denoted $\Delta\mathcal{Y}^{(k)}$.

**Learning setup.** We model nature's data generating process as a marginal distribution $D_{\mathcal{X}}$ on $\mathcal{X}$ paired with a conditional distribution function $f^* : \mathcal{X} \to \Delta\mathcal{Y}$, where $f^*(x)$ describes the distribution of $\boldsymbol{y} \mid \boldsymbol{x} = x$. We make no assumptions about $D_{\mathcal{X}}$ or the form of $f^*$. Recall that we define a higher-order predictor as a mapping $f : \mathcal{X} \to \Delta\Delta\mathcal{Y}$. Even though nature is a pointwise *pure* distribution $f^* : \mathcal{X} \to \Delta\mathcal{Y}$, we model $f$ as predicting mixtures to allow it to express epistemic uncertainty.

A *partition* of the space $\mathcal{X}$ is a collection of disjoint subsets, or equivalence classes, whose union is the entire space. For any $x \in \mathcal{X}$, we use $[x] \subset \mathcal{X}$ to refer to its equivalence class, and we use $[\cdot]$ as shorthand for the entire partition. The *Bayes mixture* over an equivalence class $[x]$, denoted $f^*([x]) \in \Delta\Delta\mathcal{Y}$ by a slight abuse of notation, is the mixture obtained by drawing $\boldsymbol{x} \sim [x]$ according to the marginal distribution $D_{\mathcal{X}}$ restricted to $[x]$ and considering $f^*(\boldsymbol{x})$. That is, every $x' \in [x]$ contributes a mixture component $f^*(x')$ to $f^*([x])$, with mixture weight given by the weight of $x'$ under $D_{\mathcal{X}}$ restricted to $[x]$.

**A note on the role of partitions.** Partitions play an essential role in any formulation of calibration for supervised learning. Consider an ordinary first-order predictor $f : \mathcal{X} \to \Delta\mathcal{Y}$ learned from the data. We view this predictor as a complete summary of all the information the algorithm has learnt. In particular, all instances $x$ that receive the same value of $f(x)$ (i.e., lie in the same level set) are de facto not being distinguished from one another by $f$. This may be because they are in fact highly similar, or because $f$ simply has not learnt enough to distinguish between them. In fact, in this view a predictor $f$ is exactly the same as specifying a level set partition (and values for each level set) — i.e. a predictor and a partition are in exact one-to-one correspondence. First-order calibration states precisely that over all instances that receive the same prediction, the predicted outcome distribution matches the true marginal outcome distribution (over all such instances).

In practice, we naturally relax the level set partition to merely be an approximate level set partition, or in general an even coarser partition $[\cdot]$ of our choice (for reasons of tractability). Nevertheless, a partition should always be thought of as expressing that points in an equivalence class are "similar" to each other from the point of view of the predictor. Any formulation of calibration for supervised learning thus has the following form: in aggregate over all points that are treated similarly by the predictor, the prediction is faithful to nature. Our definition of higher-order calibration takes this form as well.

**Alternative $k^{\text{th}}$-order representations of mixtures.** In this paper we have chosen to represent the $k^{\text{th}}$-order projection of a mixture as a distribution over $k$-snapshots, treating each snapshot as a uniform distribution over its elements. More generally one can work with other representations. The following are three natural ways to define the $k^{\text{th}}$-order projection of a mixture that will all turn out to be equivalent for us:

1. As a $k$-snapshot distribution, an object in $\Delta\mathcal{Y}^k$: The $k$-*snapshot projection* of a mixture $\pi$ is the $k$-tuple distribution arising from drawing a random $\boldsymbol{p} \sim \pi$, and drawing $\boldsymbol{y}_1, \ldots, \boldsymbol{y}_k \sim \boldsymbol{p}$ iid.

2. As a $k^{\text{th}}$-moment tensor, an object in $(\mathbb{R}^\ell)^{\otimes k}$: The $k^{\text{th}}$-*moment tensor* of a mixture $\pi$ is exactly the object $\mathbb{E}[\boldsymbol{p}^{\otimes k}]$, where $\boldsymbol{p} \sim \pi$.[11]

3. As a symmetrized $k$-snapshot distribution, an object in $\Delta\mathcal{Y}^{(k)}$: The *symmetrized $k$-snapshot projection* of a mixture $\pi$ is the distribution arising from drawing $\boldsymbol{p} \sim \pi$, then drawing $(\boldsymbol{y}_1, \ldots, \boldsymbol{y}_k) \sim \boldsymbol{p}$, and considering $\mathrm{Unif}(\boldsymbol{y}_1, \ldots, \boldsymbol{y}_k)$. This is how we originally defined $\mathrm{proj}_k \pi$ (see Definition 2.2).

---

[10]It is equivalent to the space $\mathcal{Y}^k$ modulo permutations.

[11]To avoid confusion, note that here we view $\boldsymbol{p}$ as a random vector on the simplex, and thus these are moments of probability masses, not of the labels in $\mathcal{Y}$ themselves. The latter is in any case not well-defined for us since $\mathcal{Y}$ is a categorical space.

Observe that the pmf of the $k$-snapshot projection of $\pi$ is given exactly by the $k^{\text{th}}$-moment tensor of $\pi$: that is, for any tuple $(y_1, \ldots, y_k) \in \mathcal{Y}^k$, the probability mass placed on it by the $k$-snapshot projection is

$$\mathbb{E}[\boldsymbol{p}_{y_1} \cdots \boldsymbol{p}_{y_k}] = \mathbb{E}[\boldsymbol{p}^{\otimes k}]_{(y_1, \ldots, y_k)}.$$

Moreover, this quantity is clearly invariant to permutations of the tuple. Thus it is clear that the symmetrized $k$-snapshot projection of $\pi$ contains exactly the same information as the (unsymmetrized) $k$-snapshot projection, just more compactly represented. To be explicit, we can pass from a $k$-snapshot distribution to a symmetrized $k$-snapshot distribution by summing the probability mass over all permutations of a given tuple, and we can similarly pass from the latter to the former by distributing the weight of a given symmetrized snapshot evenly over all tuples that are equivalent to it up to permutation.

In this sense the three $k^{\text{th}}$-order representations above of a true higher-order mixture $\pi$ are all formally equivalent, and we can in principle refer simply to *the $k^{\text{th}}$-order projection* $\text{proj}_k \pi$ of $\pi$ without specifying a particular representation. Notice that from $\text{proj}_k \pi$ we can also form $\text{proj}_j \pi$ for all $j \leq k$, simply by appropriate marginalization (e.g., take the marginal distribution over the first $j$ elements of each tuple).

We could formally define $k^{\text{th}}$-order calibration in terms of any of these representations. But for the purposes of this paper the symmetrized representation is by far the most convenient since it allows us to view $\mathcal{Y}^{(k)}$ as a subset of $\Delta \mathcal{Y}$.

Note that in general an arbitrary distribution over $\mathcal{Y}^k$ need not be the $k$-snapshot projection of any mixture, and similarly an arbitrary tensor in $(\mathbb{R}^\ell)^{\otimes k}$ need not be the $k^{\text{th}}$-moment tensor of any mixture. However when these objects do arise from snapshots of some mixture $\pi$, then they are equivalent to each other.

**Worked X-ray example.** For illustrative purposes, we walk through our running X-ray example using fully elaborated notation. Our instance space $\mathcal{X}$ is the space of X-ray scans. The outcome space $\mathcal{Y}$ is binary: possible values are "does not require cast" (0) and "requires cast" (1). Note that in general $\mathcal{Y}$ may be any finite label space, but binary outcome spaces offer some convenient simplifications. For example, in this case, a distribution over $\mathcal{Y}$ can be represented by a single value in $[0, 1]$, namely the probability of "requires cast". We denote the population distribution over scans by $D_{\mathcal{X}}$. We assume there exists an idealized conditional distribution function $f^* : \mathcal{X} \to \Delta \mathcal{Y}$, with $f^*(x)$ representing the idealized probability, taken over all doctors' opinions, of whether $x$ requires a cast.

A first-order predictor outputs a single distribution over $\mathcal{Y}$ for every $x$, i.e. a single prediction for the distribution of doctors' opinions for this $x$. As such, it expresses a composite kind of predictive uncertainty. Even if it were perfectly first-order calibrated, this would only tell us that the prediction at $x$ matches the average opinion ranging over the level set $[x]$ of points with the same prediction as at $x$. There is no way to tell whether an uncertain prediction corresponds to a scenario where the predictor is perfect and there is genuine disagreement in doctors' opinions (its uncertainty is mostly aleatoric), or one where the predictor itself is uncertain even if doctors are not (its uncertainty is mostly epistemic). We illustrate this problem in Figure 1. Therefore, our goal is to learn $f : \mathcal{X} \to \Delta \Delta \mathcal{Y}$, a *higher-order* predictor which outputs a mixture of distributions in $\Delta \mathcal{Y}$ at every $x$. That is, in our example it outputs a distribution over the interval $[0, 1]$ rather than a single value in $[0, 1]$.

To obtain a calibrated higher-order predictor, we can apply one of the algorithms in Section 2.2. For concreteness we opt for the second one, which calibrates a first-order predictor *post hoc*. First, we train a first-order predictor $\overline{f}$ in the usual fashion on a training set of tuples $\{(x_i, y_i)\}_{i \in [N]}$, where for each $i$, $x_i$ is sampled independently from $D_{\mathcal{X}}$ and $y_i$ is sampled from $f^*(x_i)$ (simulating an individual doctor's opinion). We use the trained predictor to compute a partition $[\cdot]$ over the input space. The canonical "level set partition" places any given X-ray $x$ in the equivalence class $[x]$ of all other inputs that $\overline{f}$ assigns approximately the same prediction (at some desired coarseness). (For example, $[x]$ might correspond to all images for which the probability assigned by $\overline{f}$ lies in $[0.6, 0.65]$.) Next, for every such class $[x]$ we sample a calibration set $\{(x_j, y_j^1, \ldots, y_j^k)\}_{j \in [C]}$ of $k$-snapshots. Within a snapshot, each of the $k$ labels $y_j^\ell$ ($\ell \in [k]$) is sampled independently of the others, corresponding to asking $k$ different doctors for their opinion on the scan. Each $k$-snapshot can be turned into a distribution in $\Delta \mathcal{Y}$: we count the occurrences of each discrete label, turn them into a histogram, and normalize it. The larger the value of $k$, the more closely this

distribution will resemble the ground-truth distribution $f^*(x_j)$. Finally, we calibrate the model by replacing the model's shared prediction for each equivalence class (namely $\overline{f}(x)$) with the uniform mixture of the $k$-snapshot distributions for each of the calibration set inputs that fall into that class. The resulting predictor is our higher-order predictor $f$. (Technically, for finite $k$, $f$ outputs discrete distributions in $\Delta \mathcal{Y}^{(k)}$, not $\Delta\Delta\mathcal{Y}$, but this approximation also improves as $k$ grows.)

Our theory shows that, for large enough $N$, $C$, and $k$, this predictor will be $k$-th order calibrated, meaning that its estimates of epistemic and aleatoric uncertainty are guaranteed to be good approximations of their actual values. For a given input $x$, aleatoric uncertainty is estimated by measuring the average entropy of all of the $k$-snapshot distributions in the prediction for $[x]$, and approaches the true aleatoric uncertainty: the value $\mathbb{E}[G(f^*(z))|z \sim [x]]$, or the expected entropy (measured by $G$) of the ground-truth label distributions for all inputs in the same equivalence class as $x$. Epistemic uncertainty, by contrast, is the remainder of the predictive uncertainty (see Eq. (3.1)). It turns out to equal a measure of the "variance" (more precisely the average divergence to the mean) of $f^*(z)$ ranging over $z \sim [x]$ (see Lemma C.1).

## B  FURTHER RELATED WORK

**Related notions of calibration.** Our notion of first-order calibration over a general discrete label space $\mathcal{Y}$ is the same as the notion of canonical or distribution calibration due to Kull & Flach (2015); Widmann et al. (2019) (see also Silva Filho et al. (2023) for a broad survey). This was extended to the regression setting ($\mathcal{Y} = \mathbb{R}$) by Song et al. (2019); Jung et al. (2021). While the latter works are related to ours (especially in the binary case where $\Delta\mathcal{Y}$ can be identified with $[0, 1]$), the key difference is that we only ever observe labels in the discrete space $\mathcal{Y}$ and never in the continuous space $\Delta\mathcal{Y}$. In the limit as snapshot size $k \to \infty$ (so that we effectively observe $f^*(x)$ with each example), higher-order calibration over $\mathcal{Y}$ can be thought of as reducing to distribution calibration for (high-dimensional) regression over the space $\Delta\mathcal{Y}$. However, no provable guarantees are known for the latter problem, whereas we actually provide provable algorithms using only finite-length snapshots.

**Distribution-free uncertainty quantification and conformal prediction.** Our techniques share a common spirit with the literature on distribution-free uncertainty quantification and especially conformal prediction (Vovk et al., 2005; Angelopoulos & Bates, 2021; Cella & Martin, 2022). This literature also makes minimal assumptions about the data-generating process and provides formal frequentist guarantees, typically prioritizing prediction intervals (or sets) for the labels. Our work provides "higher-order prediction sets" for entire probability vectors conditional on an equivalence class (see Appendix F). See Jung et al. (2021); Gibbs et al. (2023) for related work on conditional coverage guarantees. A key common theme in all these works as well as in ours is the use of *post-hoc calibration* as a way of imbuing ordinary predictors with formal coverage-like guarantees; see Roth (2022) for lecture notes on the topic and (Johnson et al., 2024, Appendix C) for an extended discussion of the relationship between second-order calibration and conformal prediction.

**Other notions of epistemic uncertainty.** In recent years one popular alternative approach to defining epistemic uncertainty is to view it as a measure of whether an instance is "out of distribution" (see e.g. Liu et al. (2020); Van Amersfoort et al. (2020)). While sometimes reasonable, we believe this is not always justifiable. We instead define our epistemic uncertainty measure based on first principles, and allow for epistemic uncertainty to occur even in-distribution.

Another formulation of epistemic uncertainty studied in Lahlou et al. (2021) is as a kind of excess risk measure, assuming an oracle estimator for aleatoric uncertainty. See also Kull & Flach (2015) for similar decompositions.

**Pitfalls of uncertainty quantification via mixtures and credal sets.** Mixtures (also termed second-order or higher-order probability distributions) are common tools for modeling uncertainty (e.g. in Bayesian methods). However, they have been critiqued (see e.g. Bengs et al. (2022); Pandey & Yu (2023); Jürgens et al. (2024)) on the grounds that it is unsound to learn a mixture (a "Level-2" object in $\Delta\Delta\mathcal{Y}$) using ordinary labeled examples ("Level-0" objects in $\mathcal{Y}$). We agree with this critique, and it is why we instead learn a Level-2 predictor from $k$-snapshots, i.e. multilabeled examples (which are approximations of Level-1 objects in $\Delta\mathcal{Y}$). Without $k$-snapshots, existing approaches to using higher-order distributions fall short in various ways. Our work arguably identifies exactly how these

approaches fall short, and proposes (a) learning from $k$-snapshots as a principled way of learning mixtures, and (b) higher-order calibration as a principled way of ensuring that these mixtures have clear frequentist semantics.

Such concerns have also spurred research into alternative approaches that avoid mixtures, such as credal sets (see e.g. Sale et al. (2023b); Chau et al. (2024)). Credal sets are incomparable to mixtures, although for many purposes a nonempty credal set of distributions functions similarly to the uniform mixture over that set, and insofar as they do, they can be miscalibrated in the sense that we propose. In concurrent work, Caprio et al. (2024) introduces a notion of calibration for credal sets similar in flavor to our own (see Appendix F in particular). Ultimately, these and other alternative formalisms make different tradeoffs in capturing a subtle real-world concept, and we work with mixtures as they are natural, expressive, and, importantly, allow proving formal semantics of the type we show in this work.

**Further references.** The area of uncertainty quantification and decomposition is vast, and here we have only mentioned works most related to ours (to the best of our knowledge). We refer the reader to the surveys by Hüllermeier & Waegeman (2021); Abdar et al. (2021); Gruber et al. (2023) as well as the related work sections of Lahlou et al. (2021); Johnson et al. (2024) for much more.

## C    ALTERNATIVE UNCERTAINTY DECOMPOSITIONS AND SEMANTICS

Many uncertainty decompositions have been considered in the literature (Houlsby et al., 2011; Kull & Flach, 2015; Malinin & Gales, 2021; Wimmer et al., 2023; Schweighofer et al., 2023; Sale et al., 2023a;c; Kotelevskii & Panov, 2024; Hofman et al., 2024), each making different tradeoffs (see e.g. Wimmer et al. (2023) for an explicit axiomatic formulation of desiderata for such decompositions). Using the unified language of generalized entropy functions and proper losses, we provide a discussion of some of these alternative uncertainty decompositions. We also discuss further interpretations of the components of the mutual information decomposition.

We first begin with an overview of proper losses and their associated generalized entropy functions.

### C.1    PROPER LOSSES AND GENERALIZED ENTROPY FUNCTIONS

Generalized entropy functions are closely associated with proper loss functions $L : \mathcal{Y} \times \Delta \mathcal{Y} \to \mathbb{R}$, where $L(y \parallel p)$ is to be interpreted as the loss incurred when we predict the distribution $p$ and observe the outcome $y$. We can extend these to functions $L : \Delta \mathcal{Y} \times \Delta \mathcal{Y} \to \mathbb{R}$ expressing the overall expected loss of $p$ with respect to a true outcome distribution $p^*$ by letting $L(p^* \parallel p) = \mathbb{E}_{\boldsymbol{y} \sim p^*}[L(\boldsymbol{y} \parallel p)]$.[12] We call the loss (strictly) proper if for all $p^*$, it is (strictly) minimized when $p = p^*$, and strictly proper if this is the unique minimizer. Define the generalized entropy function and the generalized (Bregman) divergence respectively as

$$G(p^*) = L(p^* \parallel p^*), \;\; D(p^* \parallel p) = L(p^* \parallel p) - G(p^*)$$

Note that in all of these expressions, the expectation is always taken wrt the first argument. The first argument is usually (although not always) interpreted as the truth, and the second as the estimate or prediction.

A basic and important characterization of proper losses (under mild regularity assumptions; boundedness suffices) is the following (see (Gneiting & Raftery, 2007, Theorems 1-2 and Figure 1)): the loss $L$ is (strictly) proper iff its associated generalized entropy function $G$ is (strictly) concave, and for all $p$, $L(\cdot \parallel p)$ is the gradient of $G$ at $p$.[13] In effect, this means that to specify a proper loss it is sufficient to specify a concave generalized entropy function $G$; the loss and divergence can then be obtained in terms of its gradient.

The most important examples of entropy functions are:

---

[12]The extension is natural and consistent if we view $y \in \mathcal{Y}$ as a point distribution $\mathbb{1}[y] \in \Delta \mathcal{Y}$.

[13]Here $L(\cdot \parallel p)$ denotes the vector $(L(0 \parallel p), \dots, L(\ell - 1 \parallel p)) \in \mathbb{R}^\ell$. Moreover, technically the concave analog of a subgradient (a "supergradient", or equivalently an actual subgradient of $-G$) suffices. In any case we mainly work with strictly concave functions, where this distinction does not arise.

1. Shannon entropy, $G_{\text{Shannon}}(p) = \sum_{y \in \mathcal{Y}} p_y \log \frac{1}{p_y}$. The associated loss $L_{\text{Shannon}}(p \parallel p') = \sum_{y \in \mathcal{Y}} p_y \log \frac{1}{p'_y}$ is the cross-entropy loss, and the associated divergence $D_{\text{Shannon}}(p \parallel p') = \sum_{y \in \mathcal{Y}} p_y \log \frac{p_y}{p'_y}$ is precisely the KL divergence. The associated pointwise loss function is the log loss, $L_{\text{Shannon}}(y \parallel p) = \log \frac{1}{p_y}$. In the binary case, $G_{\text{Shannon}} : [0, 1] \to [0, 1]$ is just the binary entropy function $p \mapsto p \log \frac{1}{p} + (1 - p) \log \frac{1}{1-p}$. Note that our $\log$ always has base 2 by convention.

2. Brier entropy $G_{\text{Brier}}(p) = 1 - \|p\|^2$ (also known by many other names, including Gini impurity). The associated loss turns out to be $L_{\text{Brier}}(p \parallel p') = \|p - p'\|^2 + 1 - \|p\|^2$, and the associated divergence $D_{\text{Brier}}(p \parallel p') = \|p - p'\|^2$ is precisely the squared Euclidean distance. The associated pointwise loss function is (the negation of) the Brier score, $L_{\text{Brier}}(y \parallel p) = \|p - \mathbb{1}[y]\|^2$, where $\mathbb{1}[y]$ denotes the indicator vector of $y$. In the binary case, $G_{\text{Brier}} : [0, 1] \to [0, 1]$ is given by $p \mapsto 2p(1 - p)$.[14]

Bregman divergences satisfy the following *cosine law*, a generalization of the cosine law for Euclidean distances:

$$D(p \parallel p'') = D(p \parallel p') + D(p' \parallel p'') + \langle p - p', \nabla G(p'') - \nabla G(p') \rangle, \qquad \text{(C.1)}$$

where $\nabla G(p)$ is the gradient of $G$ at $p$, viewed as a vector in $\mathbb{R}^\ell$.[15]

**A note on Shannon vs Brier entropy.** It is worth noting that in the binary case, the Shannon entropy $G_{\text{Shannon}}(p) = p \log \frac{1}{p} + (1 - p) \log \frac{1}{1-p}$ and the Brier entropy $G_{\text{Brier}}(p) = 4p(1 - p)$ are very similar as functions (where for better comparison we scale $G_{\text{Brier}}$ by a factor of 2 so that $G_{\text{Brier}}(\frac{1}{2}) = G_{\text{Shannon}}(\frac{1}{2}) = 1$). They match at $p = 0, \frac{1}{2}$ and 1, follow a very similar shape, and differ by no more than $0.15$ at any point (see Fig. 4). For most practical purposes, we recommend that practitioners consider using Brier instead of Shannon, as the benefits afforded by its simplicity largely outweigh the minor differences. In particular, the simple exact quadratic formula for $G_{\text{Brier}}$ means that second-order calibration already suffices to approximate true Brier entropy and with a much better sample complexity (and simpler estimator) than for Shannon (compare Corollaries E.9 and E.10). Unless one specifically needs to approximate Shannon entropy with very small approximation error $\epsilon$ and can afford large snapshot size $k$, Brier will typically be the more pragmatic choice.

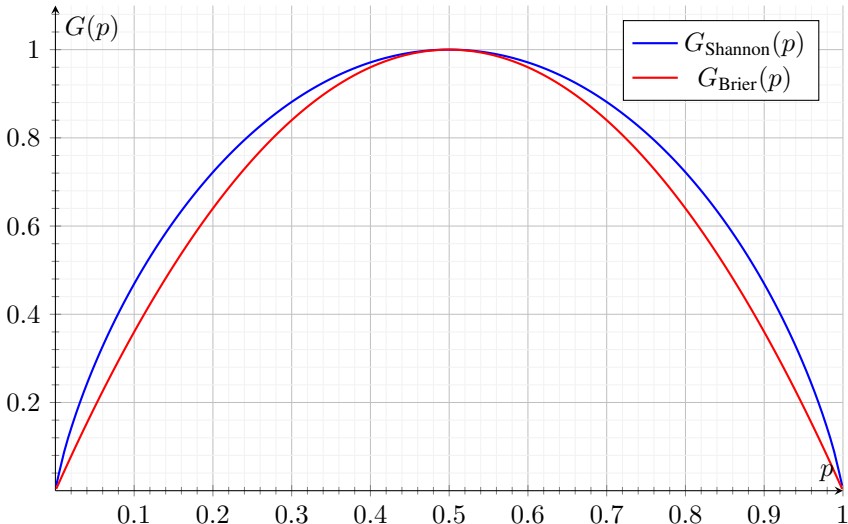

Figure 4: Comparison of Shannon and Brier binary entropy functions. Here we use the scaled version of Brier entropy, namely $G_{\text{Brier}}(p) = 4p(1 - p)$, for a better comparison.

---

[14]In the binary case, a further scaling factor of 2 can be used so that $G_{\text{Brier}}(\frac{1}{2})$ equals $G_{\text{Shannon}}(\frac{1}{2}) = 1$, but this is not an essential part of the definition.

[15]For the Brier score, $\nabla G(p) = -2p$, and this equation is exactly the result of expanding $\|p - p''\|^2$ as $\|(p - p') + (p' - p'')\|^2$.

## C.2 Loss-based semantics of the mutual uncertainty decomposition

Throughout this section, we fix a proper loss $L : \Delta\mathcal{Y} \times \Delta\mathcal{Y} \to \mathbb{R}$ and its associated generalized entropy function $G : \Delta\mathcal{Y} \to \mathbb{R}$ and generalized Bregman divergence $D : \Delta\mathcal{Y} \times \Delta\mathcal{Y} \to \mathbb{R}$. As a reminder, for $p, p^* \in \Delta\mathcal{Y}$, these are defined as follows:

$$G(p^*) = L(p^* \parallel p^*), \quad D(p^* \parallel p) = L(p^* \parallel p) - G(p^*)$$

**Lemma C.1.** *Given a generalized concave entropy function $G : \Delta\mathcal{Y} \to \mathbb{R}$ and its associated proper loss $L : \Delta\mathcal{Y} \times \Delta\mathcal{Y} \to \mathbb{R}$ and divergence $D : \Delta\mathcal{Y} \times \Delta\mathcal{Y} \to \mathbb{R}$, the following equalities hold true for any higher-order predictor $f : \mathcal{X} \to \Delta\Delta\mathcal{Y}$, and moreover the equalities marked with $\overset{HOC}{=}$ hold whenever $f$ is perfectly higher-order calibrated wrt the partition $[\cdot]$. Let $\overline{p} = \mathbb{E}_{p \sim f(x)}[p]$, $\overline{p}^* = \mathbb{E}_{p^* \sim f^*([x])}[p^*]$. Then,*

$$\mathrm{PU}_G(f : x) = \underset{p \sim f(x)}{\mathbb{E}}[L(p \parallel \overline{p})] \overset{HOC}{=} \underset{p^* \sim f^*([x])}{\mathbb{E}}[L(p^* \parallel \overline{p}^*)]$$

$$\mathrm{EU}_G(f : x) = \underset{p \sim f(x)}{\mathbb{E}}[D(p \parallel \overline{p})] \overset{HOC}{=} \underset{p^* \sim f^*([x])}{\mathbb{E}}[D(p^* \parallel \overline{p}^*)].$$

Note that the characterization of EU implies the second statement of Theorem 1.2 when $G$ is the Shannon entropy function and thus its associated divergence is the KL-divergence. We give some intution on these quantities before providing the proof. The restatement of predictive uncertainty tells us that it can be equivalently thought of as the average loss of the centroid wrt a random mixture component, $\mathbb{E}_{p \sim f(x)}[L(p \parallel \overline{p})]$, while epistemic uncertainty is the loss in entropy of $y$ upon learning the mixture component $p$, and is precisely the (generalized) *mutual information* between $y$ and $p$. Thus, $\mathrm{EU}_G(f : x)$ can be directly interpreted as a (one-sided) variance-like quantity, reflecting the level of disagreement or dispersion among the various mixture components.

*Proof.* We first prove the predictive uncertainty portion:

$$\begin{aligned}
\mathrm{PU}_G(f : x) &= G(\overline{p}) \\
&= L(\overline{p} \parallel \overline{p}) \\
&= \underset{y \sim \overline{p}}{\mathbb{E}}[L(y \parallel \overline{p})] \\
&= \underset{p \sim f(x)}{\mathbb{E}} \underset{y \sim p}{\mathbb{E}}[L(y \parallel \overline{p})] \qquad \text{(since } p \sim \pi, y \sim p \text{ is the same as } y \sim \overline{p}) \\
&= \underset{p \sim f(x)}{\mathbb{E}}[L(p \parallel \overline{p})]. \qquad\qquad\qquad\qquad\qquad\qquad (\mathrm{C.2})
\end{aligned}$$

Under higher-order calibration, we are guaranteed that $f(x)$ is identical to $f^*([x])$, which also implies $\overline{p} = \overline{p}^*$. Thus, when $f$ is perfectly higher-order calibrated, we additionally have

$$\underset{p \sim f(x)}{\mathbb{E}}[L(p \parallel \overline{p})] = \underset{p^* \sim f^*([x])}{\mathbb{E}}[L(p^* \parallel \overline{p}^*)].$$

We now move on to epistemic uncertainty. In this case, we have

$$\begin{aligned}
\mathrm{EU}_G(f : x) &= G(\overline{p}) - \underset{p \sim f(x)}{\mathbb{E}}[G(p)] \\
&= \underset{p \sim f(x)}{\mathbb{E}}[L(p \parallel \overline{p})] - \underset{p \sim f(x)}{\mathbb{E}}[G(p)] \qquad \text{(by Eq. (C.2))} \\
&= \underset{p \sim f(x)}{\mathbb{E}}[D(p \parallel \overline{p})].
\end{aligned}$$

Once again, if $f$ is perfectly higher-order calibrated, it immediately follows that $f(x) = f^*([x])$ and $\overline{p} = \overline{p}^*$, implying

$$\underset{p \sim f(x)}{\mathbb{E}}[D(p \parallel \overline{p})] = \underset{p^* \sim f^*([x])}{\mathbb{E}}[D(p^* \parallel \overline{p}^*)].$$

$\square$

## C.3 THE TOTAL MUTUAL INFORMATION DECOMPOSITION

An alternative natural expression for EU, studied by Malinin & Gales (2021); Schweighofer et al. (2023), is obtained by replacing the divergence between a random mixture component and the centroid with the divergence between two random mixture components:

$$\mathrm{EU}_G^{\mathrm{TMI}}(f:x) := \mathop{\mathbb{E}}_{\boldsymbol{p},\boldsymbol{p}'\sim f(x)}[D(\boldsymbol{p} \parallel \boldsymbol{p}')]$$

We refer to this as the "total mutual information" formulation of EU, as it turns out to be the sum of the mutual information between $\boldsymbol{y}$ and $\boldsymbol{p}$ (i.e. $\mathrm{EU}(f:x)$) and the "reverse mutual information", a notion studied by Malinin & Gales (2021) where the roles of $\boldsymbol{p}$ and $\overline{p}$ are reversed compared to the usual mutual information form of EU (recall Lemma C.1):

$$\mathrm{EU}_G^{\mathrm{RMI}}(f:x) := \mathop{\mathbb{E}}_{\boldsymbol{p}\sim f(x)}[D(\overline{p} \parallel \boldsymbol{p})].$$

The fact that $\mathrm{EU}_{\mathrm{TMI}} = \mathrm{EU} + \mathrm{EU}_{\mathrm{RMI}}$ follows from an application of the cosine law, Eq. (C.1):

$$\mathrm{EU}_G(f:x) + \mathrm{EU}_G^{\mathrm{RMI}}(f:x) = \mathop{\mathbb{E}}_{\boldsymbol{p}\sim f(x)}[D(\boldsymbol{p} \parallel \overline{p})] + \mathop{\mathbb{E}}_{\boldsymbol{p}\sim f(x)}[D(\overline{p} \parallel \boldsymbol{p})]$$
$$= \mathop{\mathbb{E}}_{\boldsymbol{p},\boldsymbol{p}'\sim\pi}[D(\boldsymbol{p} \parallel \boldsymbol{p}')]$$
$$= \mathrm{EU}_G^{\mathrm{TMI}}(f:x)$$

One can use the new notion of $\mathrm{EU}^{\mathrm{TMI}}$ in place of our original notion of epistemic uncertainty in the mutual information decomposition to get a new notion of predictive uncertainty and a new "total mutual information decomposition":

$$\underbrace{\mathop{\mathbb{E}}_{\boldsymbol{p},\boldsymbol{p}'\sim f(x)}[L(\boldsymbol{p} \parallel \boldsymbol{p}')]}_{\text{Predictive uncertainty } \mathrm{PU}_G^{\mathrm{TMI}}(f:x)} = \underbrace{\mathop{\mathbb{E}}_{\boldsymbol{p}\sim f(x)}[G(\boldsymbol{p})]}_{\text{Aleatoric uncertainty estimate } \mathrm{AU}_G(f:x)} + \underbrace{\mathop{\mathbb{E}}_{\boldsymbol{p},\boldsymbol{p}'\sim f(x)}[D(\boldsymbol{p} \parallel \boldsymbol{p}')]}_{\text{Epistemic uncertainty (TMI) } \mathrm{EU}_G^{\mathrm{TMI}}(f:x)}$$

(C.3)

The new $\mathrm{PU}_G^{\mathrm{TMI}}(f:x)$ is the average pairwise loss of the mixture components wrt each other (by comparison, recall that by Eq. (C.2), our original $\mathrm{PU}_G(f:x)$, was the average loss of the centroid wrt a mixture component). The $\mathrm{AU}_G(f:x)$ term remains the same.

## C.4 LOSS DECOMPOSITIONS AND HIGHER-ORDER CALIBRATION

Higher-order calibration also provides information about the *loss* incurred by a particular prediction. We consider two potential ways of measuring the loss of a higher-order prediction $f(x)$ with respect to $f^*$. The first is to predict the centroid of a higher-order prediction, i.e. $\overline{p} = \mathbb{E}_{\boldsymbol{p}\sim f(x)}[\boldsymbol{p}]$. The expected loss over a partition $[x]$ incurred when predicting the centroid $\overline{p}$ is exactly

$$\mathop{\mathbb{E}}_{\boldsymbol{p}^*\sim f^*([x])}[L(\boldsymbol{p}^* \parallel \overline{p})] = \underbrace{\mathop{\mathbb{E}}_{\boldsymbol{p}_*\sim f^*([x])}[G(\boldsymbol{p}_*)]}_{\text{avg AU}} + \underbrace{\mathop{\mathbb{E}}_{\boldsymbol{p}_*\sim f^*([x])}[D(\boldsymbol{p}_* \parallel \overline{p})]}_{\text{avg bias}}. \qquad (C.4)$$

When $f$ is higher-order calibrated, this loss decomposition becomes exactly the mutual-information decomposition as by Lemma C.1, we have

$$\mathop{\mathbb{E}}_{\boldsymbol{p}^*\sim f^*([x])}[L(\boldsymbol{p}^* \parallel \overline{p})] = \mathrm{PU}_G(f:x),$$

$$\mathop{\mathbb{E}}_{\boldsymbol{p}^*}[G(\boldsymbol{p}^*)] = \mathrm{AU}_G(f:x),$$

and

$$\mathop{\mathbb{E}}_{\boldsymbol{p}^*}[D(\boldsymbol{p}^* \parallel \overline{p})] = \mathrm{EU}_G(f:x).$$

Thus, higher-order calibration not only provides meaningful semantics for predictive, epistemic, and aleatoric uncertainty in the mutual information decomposition, but also ties them to the expected loss incurred when making predictions using the centroid of $f(x)$.

A second, slightly less natural, choice of loss measurement is the loss incurred when we predict at random according to $\boldsymbol{p} \sim f(x)$:

$$\mathop{\mathbb{E}}_{\boldsymbol{p},\boldsymbol{p}^*}[L(\boldsymbol{p}_* \parallel \boldsymbol{p})] = \mathop{\mathbb{E}}_{\boldsymbol{p}^*}[G(\boldsymbol{p}^*)] + \mathop{\mathbb{E}}_{\boldsymbol{p},\boldsymbol{p}^*}[D(\boldsymbol{p}^* \parallel \boldsymbol{p})]$$

When $f$ is perfectly higher-order calibrated, these terms are exactly recovered by $\mathrm{PU}_G^{\mathrm{TMI}}(f : x), \mathrm{AU}_G(f : x), \mathrm{EU}_G^{\mathrm{TMI}}(f : x)$, respectively, in the total mutual information decomposition.

We can also break down Eq. (C.4) even further by making use of the notion of the "grouping loss" from Kull & Flach (2015). Let $\overline{p}^* = \mathbb{E}_{\boldsymbol{p}^* \sim f^*([x])}[\boldsymbol{p}^*]$. Then our "average bias" term (which they term "epistemic loss") can be decomposed further as follows:

$$\underbrace{\mathop{\mathbb{E}}_{\boldsymbol{p}^* \sim f^*([x])}[D(\boldsymbol{p}^* \parallel \overline{p})]}_{\text{avg bias}} = \underbrace{\mathop{\mathbb{E}}_{\boldsymbol{p}_* \sim f^*([x])}[D(\boldsymbol{p}^* \parallel \overline{p}^*)]}_{\text{grouping loss}} + \underbrace{D(\overline{p}^* \parallel \overline{p})}_{\text{FOC error}}.$$

The FOC error measures how far our mean prediction $\overline{p}$ over the set $[x]$ is from the Bayes mean $\overline{p}^*$, and the grouping loss measures the inherent spread in $f^*(\boldsymbol{x})$ over $[x]$. If we have first-order calibration, then the FOC error is zero. If we also have higher-order calibration, then the grouping loss is exactly $\mathrm{EU}_G(f : x)$.

## D    PROOFS FROM SECTION 2

### D.1    USEFUL LEMMAS ABOUT WASSERSTEIN DISTANCE

We present two lemmas relating Wasserstein distance and total variation distance that will be useful in the proofs for this section.

**Lemma D.1** (Gibbs & Su (2002), Theorem 4)**.** *Let $p_1, p_2 \in \Delta\mathcal{Y}$ be any two distributions in $\Delta\mathcal{Y}$. Then, we have*

$$W_1(p_1, p_2) \leq \mathsf{diam}(\Delta\mathcal{Y}) d_{TV}(p_1, p_2) = 2 d_{TV}(\boldsymbol{p}_1, \boldsymbol{p}_2)$$

*where $d_{TV}(p_1, p_2) = \frac{1}{2}\|p_1 - p_2\|_1$ is the total variation distance.*

The last equality of the lemma follows as a special case due to the standard fact that the maximum $\ell_1$ distance between any two points in the simplex $\Delta\mathcal{Y} = \Delta_\ell$ is at most two.

**Lemma D.2** (Gibbs & Su (2002), remarks in definition of total variation distance)**.** *Let $p_1, p_2 \in \Delta\mathcal{Y}$ be any two distributions and $\Gamma(p_1, p_2)$ be the space of couplings between these two distributions. Then,*

$$d_{TV}(p_1, p_2) = \inf_{\gamma \in \Gamma(p_1, p_2)} \Pr_{(\boldsymbol{y}_1, \boldsymbol{y}_2) \sim \gamma}[\boldsymbol{y}_1 \neq \boldsymbol{y}_2].$$

### D.2    SANDWICH BOUNDS FOR $k$-SNAPSHOT AND HIGHER-ORDER PREDICTORS

The following lemma will lay the foundation for proving sandwich bounds relating $k^{\text{th}}$-order and higher-order calibration error:

**Lemma D.3.** *Given any mixture $\pi \in \Delta\Delta\mathcal{Y}$ (and in particular for $\pi = f^*([x])$) and $k > 0$, we have $W_1(\pi, \mathrm{proj}_k \pi) \leq |\mathcal{Y}|/(2\sqrt{k})$.*

*Proof.* We consider the coupling $\mu \in \Gamma(\pi, \mathrm{proj}_k \pi)$ defined by first drawing $\boldsymbol{p} \sim \pi$, then drawing a $k$-snapshot from $\boldsymbol{p}$ by taking $k$ i.i.d. draws $\boldsymbol{y}_1, ..., \boldsymbol{y}_k \sim \boldsymbol{p}$, and outputting $(\boldsymbol{p}, \mathrm{Unif}(\boldsymbol{y}_1, ..., \boldsymbol{y}_k))$.

By definition of the Wasserstein distance as the infinum over all possible couplings, we have

$$W_1(\pi, \mathrm{proj}_k \pi) \leq \mathop{\mathbb{E}}_{(\boldsymbol{p}, \mathrm{Unif}(\boldsymbol{y}_1, ..., \boldsymbol{y}_k)) \sim \mu}[\|\boldsymbol{p} - \mathrm{Unif}(\boldsymbol{y}_1, ..., \boldsymbol{y}_k)\|_1].$$

Expanding out our definition of $\mu$, we have that

$$
\begin{aligned}
W_1(\pi, \mathrm{proj}_k \pi) &\leq \mathop{\mathbb{E}}_{\boldsymbol{p} \sim \pi} \left[ \mathop{\mathbb{E}}_{\boldsymbol{y}_1, \ldots, \boldsymbol{y}_k \sim \boldsymbol{p}, i.i.d.} \left[ \| \boldsymbol{p} - \mathrm{Unif}(\boldsymbol{y}_1, \ldots, \boldsymbol{y}_k) \|_1 \right] \right] \\
&= \mathop{\mathbb{E}}_{\boldsymbol{p} \sim \pi} \left[ \mathop{\mathbb{E}}_{\boldsymbol{y}_1, \ldots, \boldsymbol{y}_k \sim \boldsymbol{p}, i.i.d.} \left[ \sum_{j=1}^{|\mathcal{Y}|} \left| \boldsymbol{p}_j - \frac{1}{k} \sum_{i=1}^{k} \mathbf{1}[\boldsymbol{y}_i = j] \right| \right] \right] \\
&= \mathop{\mathbb{E}}_{\boldsymbol{p} \sim \pi} \left[ \sum_{j=1}^{|\mathcal{Y}|} \mathop{\mathbb{E}}_{\boldsymbol{y}_1, \ldots, \boldsymbol{y}_k \sim \boldsymbol{p}, i.i.d.} \left[ \left| \boldsymbol{p}_j - \frac{1}{k} \sum_{i=1}^{k} \mathbf{1}[\boldsymbol{y}_i = j] \right| \right] \right] \\
&= \mathop{\mathbb{E}}_{\boldsymbol{p} \sim \pi} \left[ \frac{1}{k} \sum_{j=1}^{|\mathcal{Y}|} \mathop{\mathbb{E}}_{\boldsymbol{y}_1, \ldots, \boldsymbol{y}_k \sim \boldsymbol{p}, i.i.d.} \left[ \left| k\boldsymbol{p}_j - \sum_{i=1}^{k} \mathbf{1}[\boldsymbol{y}_i = j] \right| \right] \right] \\
&\leq \mathop{\mathbb{E}}_{\boldsymbol{p} \sim \pi} \left[ \frac{1}{k} \sum_{j=1}^{|\mathcal{Y}|} \sqrt{ \mathop{\mathbb{E}}_{\boldsymbol{y}_1, \ldots, \boldsymbol{y}_k \sim \boldsymbol{p}, i.i.d.} \left[ \left( k\boldsymbol{p}_j - \sum_{i=1}^{k} \mathbf{1}[\boldsymbol{y}_i = j] \right)^2 \right] } \right]
\end{aligned}
$$

$$\text{(Jensen's Inequality)}$$

Note that for any $j$, $\sum_{i=1}^{k} \mathbf{1}[\boldsymbol{y}_i = j]$ is distributed as a Binomial random variable outputting the number of 1s out of $k$ draws from a Bernoulli random variable with mean $\boldsymbol{p}_j$. This Binomial random variable has mean $k\boldsymbol{p}_j$, and so we can re-interpret the quantity under the square root as its variance:

$$
\mathop{\mathbb{E}}_{\boldsymbol{y}_1, \ldots, \boldsymbol{y}_k \sim \boldsymbol{p}, i.i.d.} \left[ \left( k\boldsymbol{p}_j - \sum_{i=1}^{k} \mathbf{1}[\boldsymbol{y}_i = j] \right)^2 \right] = \mathrm{Var}(\mathrm{Bin}(k, \boldsymbol{p}_j)) = k\boldsymbol{p}_j(1 - \boldsymbol{p}_j) \leq k/4.
$$

Plugging this upper bound back into the expectation, we get

$$
W_1(\pi, \mathrm{proj}_k \pi) \leq \mathop{\mathbb{E}}_{\boldsymbol{p} \sim \pi} \left[ \frac{1}{k} \sum_{j=1}^{|\mathcal{Y}|} \sqrt{k/4} \right] = \frac{1}{k} \sum_{j=1}^{|\mathcal{Y}|} \sqrt{k/4} = \frac{|\mathcal{Y}|}{2\sqrt{k}}
$$

This gives us the desired upper bound on $W_1(\pi, \mathrm{proj}_k \pi)$, completing the proof. $\qquad \square$

Using the result of Lemma D.3, we get the following corollary:

**Corollary D.4** (Corollary to Lemma D.3). *Consider any $k$-snapshot predictor $g : \mathcal{X} \to \Delta \mathcal{Y}^{(k)}$ and higher-order predictor $f : \mathcal{X} \to \Delta\Delta\mathcal{Y}$. For any $x$, we have*

$$
|W_1(g(x), \mathrm{proj}_k f(x)) - W_1(g(x), f(x))| \leq \frac{|\mathcal{Y}|}{2\sqrt{k}}.
$$

*Proof.* By the triangle inequality,

$$
\begin{aligned}
W_1(g(x), \mathrm{proj}_k f(x)) &\leq W_1(g(x), f(x)) + W_1(f(x), \mathrm{proj}_k f(x)) \\
&\leq W_1(g(x), f(x)) + \frac{|\mathcal{Y}|}{2\sqrt{k}} && \text{(Lemma D.3)}
\end{aligned}
$$

The other direction follows similarly:

$$
\begin{aligned}
W_1(g(x), f(x)) &\leq W_1(g(x), \mathrm{proj}_k f(x)) + W_1(\mathrm{proj}_k f(x), f(x)) && \text{(triangle inequality)} \\
&\leq W_1(g(x), \mathrm{proj}_k f(x)) + \frac{|\mathcal{Y}|}{2\sqrt{k}} && \text{(Lemma D.3)}
\end{aligned}
$$

$$\square$$

Thus far, we've considered cases where we have a fixed $k$-snapshot predictor and compare it to a higher-order predictor or its projection. Lemma D.3 also allows us to prove Lemma D.5, a sandwich bound for the distance between two higher-order predictions compared to their $k^{\text{th}}$-order projections:

**Lemma D.5.** *Consider any higher-order predictors $f, h : \mathcal{X} \to \Delta\Delta\mathcal{Y}$. For any $k > 0$ and $x \in \mathcal{X}$, we have*

$$W_1(\text{proj}_k f(x), \text{proj}_k h(x)) \leq W_1(f(x), h(x)) \leq W_1(\text{proj}_k f(x), \text{proj}_k h(x)) + \frac{|\mathcal{Y}|}{\sqrt{k}}.$$

*Proof.* Fix any $x \in \mathcal{X}$ and $k > 0$. We begin with the upper bound, as the proof is very short (it follows almost immediately from Lemma D.3), and then continue to the lower bound.

By Lemma D.3, we are guaranteed that $W_1(f(x), \text{proj}_k f(x))$ and $W_1(\text{proj}_k h(x), h(x))$ are both upper-bounded by $|\mathcal{Y}|/(2\sqrt{k})$. Applying the triangle inequality, it follows that

$$W_1(f(x), h(x)) \leq W_1(f(x), \text{proj}_k f(x)) + W_1(\text{proj}_k f(x), \text{proj}_k h(x)) + W_1(\text{proj}_k h(x), h(x))$$

$$\leq W_1(\text{proj}_k f(x), \text{proj}_k h(x)) + \frac{|\mathcal{Y}|}{\sqrt{k}} \qquad\qquad \text{(Lemma D.3)}$$

This completes the upper bound. We proceed to the lower bound, and show that $W_1(\text{proj}_k f(x), \text{proj}_k h(x)) \leq W_1(f(x), h(x))$.

Let $\mu = \arg\min_{\mu \in \Gamma(f(x), h(x))} \mathbb{E}_{(\boldsymbol{p}, \boldsymbol{p}') \sim \mu}[\|\boldsymbol{p} - \boldsymbol{p}'\|_1]$, i.e. the optimal coupling of $f(x)$ and $h(x)$ in the context of Wasserstein distance. We use $\mu$ to construct a coupling $\mu_k$ of $\text{proj}_k f(x)$ and $\text{proj}_k h(x)$ as follows. We first draw $(\boldsymbol{p}, \boldsymbol{p}') \sim \mu$, and then sample $k$ pairs of $y$-values, i.i.d., $(\boldsymbol{y}_1, \boldsymbol{y}'_1), ..., (\boldsymbol{y}_k, \boldsymbol{y}'_k) \sim \gamma(\boldsymbol{p}, \boldsymbol{p}')$, where $\gamma(\boldsymbol{p}, \boldsymbol{p}')$ is the optimal coupling of $\boldsymbol{p}$ and $\boldsymbol{p}'$ defined as

$$\gamma(\boldsymbol{p}, \boldsymbol{p}') = \arg\min_{\gamma \in \Gamma(\boldsymbol{p}, \boldsymbol{p}')} \Pr_{(\boldsymbol{y}, \boldsymbol{y}') \sim \gamma}[\boldsymbol{y} \neq \boldsymbol{y}'].$$

We then aggregate these $k$ y's as $k$-snapshots, and output the pair of distributions $\boldsymbol{q} = \text{Unif}(\boldsymbol{y}_1, \ldots, \boldsymbol{y}_k)$ and $\boldsymbol{q}' = \text{Unif}(\boldsymbol{y}'_1, \ldots, \boldsymbol{y}'_k)$. This defines our coupling $\mu_k$. Since $\boldsymbol{q} \sim \text{proj}_k f(x)$ and $\boldsymbol{q}' \sim \text{proj}_k h(x)$, this is a valid coupling of $\text{proj}_k f(x)$ and $\text{proj}_k h(x)$, so it upper bounds the Wasserstein distance between these two mixtures:

$$W_1(\text{proj}_k f(x), \text{proj}_k h(x)) \leq \mathbb{E}_{(\boldsymbol{q}, \boldsymbol{q}') \sim \mu_k}[\|\boldsymbol{q} - \boldsymbol{q}'\|_1]$$

We will now show that

$$\mathbb{E}_{(\boldsymbol{q}, \boldsymbol{q}') \sim \mu_k}[\|\boldsymbol{q} - \boldsymbol{q}'\|_1] \leq \mathbb{E}_{(\boldsymbol{p}, \boldsymbol{p}') \sim \mu}[\|\boldsymbol{p} - \boldsymbol{p}'\|_1]$$

i.e. the expected distance under pairs of distributions drawn from $\mu_k$ is smaller than the expected distance under pairs drawn from $\mu$ and thus lower-bounds $W_1(f(x), h(x))$ by definition.

Expanding the definition of $\mu_k$, we have

$$\mathbb{E}_{(\boldsymbol{q}, \boldsymbol{q}') \sim \mu_k}[\|\boldsymbol{q} - \boldsymbol{q}'\|_1] = \mathbb{E}_{\substack{(\boldsymbol{p}, \boldsymbol{p}') \sim \mu \\ (\boldsymbol{y}_1, \boldsymbol{y}'_1), ..., (\boldsymbol{y}_k, \boldsymbol{y}'_k) \sim \gamma(\boldsymbol{p}, \boldsymbol{p}'), i.i.d.}}[\|\text{Unif}(\boldsymbol{y}_1, ..., \boldsymbol{y}_k) - \text{Unif}(\boldsymbol{y}'_1, ...\boldsymbol{y}'_k)\|_1]$$

$$= \mathbb{E}_{\substack{(\boldsymbol{p}, \boldsymbol{p}') \sim \mu \\ (\boldsymbol{y}_1, \boldsymbol{y}'_1), ..., (\boldsymbol{y}_k, \boldsymbol{y}'_k) \sim \gamma(\boldsymbol{p}, \boldsymbol{p}'), i.i.d.}}[2 \inf_{\gamma \in \Gamma(\text{Unif}(\boldsymbol{y}_1, ..., \boldsymbol{y}_k), \text{Unif}(\boldsymbol{y}'_1, ...\boldsymbol{y}'_k))} \Pr_{(\boldsymbol{y}, \boldsymbol{y}') \sim \gamma}[\boldsymbol{y} \neq \boldsymbol{y}']]$$

Where the second step follows from Lemma D.2 (note that $d_{TV}(p_1, p_2) = \|p_1 - p_2\|_1/2$).

Among the possible couplings of $\text{Unif}(\boldsymbol{y}_1, ..., \boldsymbol{y}_k)$ and $\text{Unif}(\boldsymbol{y}'_1, ...\boldsymbol{y}'_k)$, one such coupling is to draw an index $i \sim [k]$ uniformly, and then output the two $\boldsymbol{y}_i$ and $\boldsymbol{y}'_i$ corresponding to this index. This gives an upper bound of

$$\mathbb{E}_{(\boldsymbol{q},\boldsymbol{q}')\sim\mu_k}[\|\boldsymbol{q}-\boldsymbol{q}'\|_1] = \mathbb{E}_{\substack{(\boldsymbol{p},\boldsymbol{p}')\sim\mu \\ (\boldsymbol{y}_1,\boldsymbol{y}_1'),\ldots,(\boldsymbol{y}_k,\boldsymbol{y}_k')\sim\gamma(\boldsymbol{p},\boldsymbol{p}'),i.i.d.}}\left[2\inf_{\gamma\in\Gamma(\mathrm{Unif}(\boldsymbol{y}_1,\ldots,\boldsymbol{y}_k),\mathrm{Unif}(\boldsymbol{y}_1',\ldots\boldsymbol{y}_k'))}\Pr_{(\boldsymbol{y},\boldsymbol{y}')\sim\gamma}[\boldsymbol{y}\neq\boldsymbol{y}']\right]$$

$$\leq \mathbb{E}_{\substack{(\boldsymbol{p},\boldsymbol{p}')\sim\mu \\ (\boldsymbol{y}_1,\boldsymbol{y}_1'),\ldots,(\boldsymbol{y}_k,\boldsymbol{y}_k')\sim\gamma(\boldsymbol{p},\boldsymbol{p}'),i.i.d.}}\left[2\Pr_{i\sim\mathrm{Unif}([k])}[\boldsymbol{y}_i\neq\boldsymbol{y}_i']\right]$$

$$= \mathbb{E}_{\substack{(\boldsymbol{p},\boldsymbol{p}')\sim\mu \\ (\boldsymbol{y}_1,\boldsymbol{y}_1'),\ldots,(\boldsymbol{y}_k,\boldsymbol{y}_k')\sim\gamma(\boldsymbol{p},\boldsymbol{p}'),i.i.d.}}\left[\frac{2}{k}\sum_{i=1}^{k}\Pr[\boldsymbol{y}_i\neq\boldsymbol{y}_i']\right]$$

$$= \mathbb{E}_{(\boldsymbol{p},\boldsymbol{p}')\sim\mu}\left[\frac{2}{k}\sum_{i=1}^{k}\Pr_{(\boldsymbol{y},\boldsymbol{y}')\sim\gamma(\boldsymbol{p},\boldsymbol{p}')}[\boldsymbol{y}\neq\boldsymbol{y}']\right]$$

where the last step uses the definition of $\mu_k$ as obtaining $\boldsymbol{y}_i, \boldsymbol{y}_i'$ as independent draws from the optimal coupling of $\boldsymbol{p}$ and $\boldsymbol{p}'$. Further simplifying, we get

$$\mathbb{E}_{(\boldsymbol{q},\boldsymbol{q}')\sim\mu_k}[\|\boldsymbol{q}-\boldsymbol{q}'\|_1] \leq \mathbb{E}_{(\boldsymbol{p},\boldsymbol{p}')\sim\mu}\left[\frac{2}{k}\sum_{i=1}^{k}\Pr_{(\boldsymbol{y},\boldsymbol{y}')\sim\gamma(\boldsymbol{p},\boldsymbol{p}')}[\boldsymbol{y}\neq\boldsymbol{y}']\right]$$

$$= \mathbb{E}_{(\boldsymbol{p},\boldsymbol{p}')\sim\mu}\left[2\Pr_{(\boldsymbol{y},\boldsymbol{y}')\sim\gamma(\boldsymbol{p},\boldsymbol{p}')}[\boldsymbol{y}\neq\boldsymbol{y}']\right]$$

$$= \mathbb{E}_{(\boldsymbol{p},\boldsymbol{p}')\sim\mu}[\|\boldsymbol{p}-\boldsymbol{p}'\|_1] \qquad\text{(Lemma D.2)}$$

$$= W_1(f(x), h(x))$$

(Recall in the application of Lemma D.2 that $\gamma$ was defined as the optimal coupling of $\boldsymbol{p}$ and $\boldsymbol{p}'$ in terms of the cost $d(y, y') = \mathbf{1}[y \neq y']$). The last step applies the definition of $\mu$ as the optimal coupling for $\ell_1$-Wasserstein distance. We conclude that $W_1(\mathrm{proj}_k f(x), \mathrm{proj}_k h(x)) \leq W_1(f(x), h(x))$, completing the lower bound. $\qquad\square$

## D.3  PROOF OF THEOREM 2.4

We restate the theorem for readability:

**Theorem D.6** (Restatement of Theorem 2.4, Corollary to Lemma D.3: $k^{\text{th}}$-order calibration implies higher-order calibration.)**.** *Let $g : \mathcal{X} \to \Delta\mathcal{Y}^{(k)}$ be $\epsilon$-$k^{th}$-order calibrated with respect to a partition $[\cdot]$. Then, we can guarantee that $g$ is also $(\epsilon + \frac{|\mathcal{Y}|}{2\sqrt{k}})$-higher-order calibrated with respect to the same partition $[\cdot]$.*

*Proof.* The theorem follows immediately from Corollary D.4, which tells us that for any $x \in \mathcal{X}$,

$$W_1(g(x), f^*([x])) \leq W_1(g(x), \mathrm{proj}_k f^*([x])) + \frac{|\mathcal{Y}|}{2\sqrt{k}}.$$

$k^{\text{th}}$-order calibration tells us that $W_1(g(x), \mathrm{proj}_k f^*([x])) \leq \epsilon$, and thus the higher-order calibration error $W_1(g(x), f^*([x]))$ is bounded by $\epsilon + \frac{|\mathcal{Y}|}{2\sqrt{k}}$.

$\qquad\square$

## D.4  PROOF OF LEMMA 2.5

We restate the lemma for readability:

**Lemma D.7** (Restatement of Lemma 2.5)**.** *Let $\mathcal{Y} = \{0, 1\}$, and let $g : \mathcal{X} \to \Delta\mathcal{Y}^{(k)}$ be $\epsilon$-$k^{th}$-order calibrated with respect to a partition $[\cdot]$. Then for each $x \in \mathcal{X}$, we can use $g(x)$ to generate a vector*

*of moment estimates* $(m_1, \ldots, m_k) \in \mathbb{R}^n_{\geq 0}$ *such that for each* $i \in [k]$,

$$\left| m_i - \mathop{\mathbb{E}}_{\boldsymbol{x} \sim [x]} [f^*(\boldsymbol{x})^i] \right| \leq i\epsilon/2.$$

The proof will make use of an intermediate lemma, which tells us that the first $k$-moments of a mixture $\pi \in \Delta\Delta\mathcal{Y}$ can be exactly reconstructed using only its $k^{\text{th}}$-order projection:

**Lemma D.8.** *Consider any mixture* $\pi \in \Delta[0, 1]$ *and its* $k^{th}$*-order projection* $\mathrm{proj}_k \pi \in \Delta\{0, 1/k, ..., 1\}$. *For* $m \in [k]$, *let* $M_{k,m} : \{0, 1/k, 2/k, ..., 1\} \to [0, 1]$ *be a function defined as follows:*

$$M_{k,m}(p) = \begin{cases} \frac{\binom{pk}{m}}{\binom{k}{m}} & p \geq m/k \\ 0 & otherwise \end{cases}.$$

*Then,*

$$\mathop{\mathbb{E}}_{\boldsymbol{p} \sim \mathrm{proj}_k \pi} [M_{k,m}(\boldsymbol{p})] = \mathop{\mathbb{E}}_{\boldsymbol{p} \sim \pi} [\boldsymbol{p}^m].$$

*Proof.* Consider the expectation of $M_{k,m}$ under $\mathrm{proj}_k \pi$.

$$\mathop{\mathbb{E}}_{\widehat{\boldsymbol{p}} \sim \mathrm{proj}_k \pi} [M_{k,m}(\widehat{\boldsymbol{p}})] = \mathop{\mathbb{E}}_{\substack{\boldsymbol{p} \sim \pi \\ \boldsymbol{y}_1, \ldots, \boldsymbol{y}_k \sim \boldsymbol{p}, \text{i.i.d.}}} \left[ M_{k,m} \left( \sum_{i=1}^{k} \frac{\boldsymbol{y}_i}{k} \right) \right]$$

One way to view $M_{k,m}(p)$ is to view $p = i/k$ as a snapshot with $i$ 1s and $k - i$ 0s. $M_{k,m}(p)$ gives the probability that a random permutation of this snapshot starts with $m$ consecutive 1s. Because the distribution over possible permutations of snapshots is symmetric, we can equivalently express the above expectation as just the probability that a random $k$-snapshot starts with $m$ 1s:

$$\mathop{\mathbb{E}}_{\widehat{\boldsymbol{p}} \sim \mathrm{proj}_k \pi} [M_{k,m}(\widehat{\boldsymbol{p}})] = \mathop{\mathbb{E}}_{\substack{\boldsymbol{p} \sim \pi \\ \boldsymbol{y}_1, \ldots, \boldsymbol{y}_k \sim \boldsymbol{p}, \text{i.i.d.}}} [\mathbf{1}[\boldsymbol{y}_1, ..., \boldsymbol{y}_m = 1]]$$

$$= \mathop{\mathbb{E}}_{\substack{\boldsymbol{p} \sim \pi \\ \boldsymbol{y}_1, \ldots, \boldsymbol{y}_k \sim \boldsymbol{p}, \text{i.i.d.}}} [\prod_{i=1}^{m} \boldsymbol{y}_i]$$

$$= \mathop{\mathbb{E}}_{\boldsymbol{p} \sim \pi} [\prod_{i=1}^{m} \mathop{\mathbb{E}}_{\boldsymbol{y} \sim \boldsymbol{p}} [\boldsymbol{y}]] \qquad \text{(independence of } \boldsymbol{y}_i \text{s)}$$

$$= \mathop{\mathbb{E}}_{\boldsymbol{p} \sim \pi} [\prod_{i=1}^{m} \boldsymbol{p}]$$

$$= \mathop{\mathbb{E}}_{\boldsymbol{p} \sim \pi} [\boldsymbol{p}^m].$$

Thus, the expectation of $M_{k,m}$ over the mixture $\mathrm{proj}_k \pi$ is guaranteed to be exactly the $m$-th moment of the higher-order mixture $\pi$. $\qquad \square$

Additionally, we show that each $M_{j,k}$ satisfies a Lipschitz condition, which will imply that evaluating the expectation of $M_{j,k}$ on a distribution close in Wasserstein distance to $\mathrm{proj}_k f^*([x])$ will closely approximate the true moment.

**Lemma D.9.** *Let* $M_{k,m}$ *be defined as in Lemma D.8. Then, for any* $p_1, p_2 \in \{0, 1/k, ..., 1\}$, *we are guaranteed that*

$$|M_{k,m}(p_1) - M_{k,m}(p_2)| \leq m|p_1 - p_2|.$$

*Proof.* We note that it suffices to show that for any consecutive $p_1$ and $p_2 = p_1 - 1/k$, we have

$$|M_{k,m}(p_1) - M_{k,m}(p_2)| \leq m/k,$$

as then it follows by a telescoping sum that for any $p_1, p_2 \in \{0, 1/k, ..., 1\}$, assuming without loss of generality that $p_1 \geq p_2$, we have

$$|M_{k,m}(p_1) - M_{k,m}(p_2)| \leq m|p_1 - p_2|,$$

giving the desired bound.

There are three potential cases: either (1) $M_{k,m}(p_1), M_{k,m}(p_2) > 0$, (2) $M_{k,m}(p_1) = 0, M_{k,m}(p_2) > 0$ (wlog), or (3) $M_{k,m}(p_1) = M_{k,m}(p_2) = 0$.

In the last case, we trivially have $|M_{k,m}(p_1) - M_{k,m}(p_2)| = 0 - 0 = 0$. We focus on the other two cases.

**Case 1:** $M_{k,m}(p_1), M_{k,m}(p_2) > 0$. By assumption, there is some $i > m$ such that $p_1 k = i$, $p_2 k = i - 1$. We use this to simplify the difference between $M_{k,m}$ values:

$$|M_{k,m}(p_1) - M_{k,m}(p_2)| = \frac{\binom{i}{m}}{\binom{k}{m}} - \frac{\binom{i-1}{m}}{\binom{k}{m}} = \frac{\binom{i-1}{m-1}}{\binom{k}{m}}$$

We observe that for a fixed $m$, and $k$, this quantity is increasing in $i$ for $i \geq m$ because the denominator is constant, and $\binom{i-1}{m-1}$ is increasing in $i \geq m$. Thus, we can upper bound the difference by the difference achieved by the largest possible value of $i$, which is $i = k$:

$$|M_{k,m}(p_1) - M_{k,m}(p_2)| \leq \frac{\binom{k-1}{m-1}}{\binom{k}{m}} = \frac{m}{k}.$$

Finally, we consider the last case:

**Case 2:** $M_{k,m}(p_1) > 0$, $M_{k,m}(p_2) = 0$. Because $p_1 = p_2 + 1/k$, and $M_{k,m}(p)$ is always positive for any $p \geq m/k$, this case can only happen when $p_1 = m/k$, and $p_2 = (m - 1)/k$. We calculate the difference for these values:

$$|M_{k,m}(p_1) - M_{k,m}(p_2)| = \frac{\binom{m}{m}}{\binom{k}{m}} - 0 = \frac{1}{\binom{k}{m}} = \frac{m!(k-m)!}{k!} = \frac{m}{k}\frac{1}{\binom{k-1}{m-1}} \leq \frac{m}{k}.$$

Thus, all consecutive differences are bounded by $m/k$, completing the proof. $\square$

We are now ready to prove Lemma 2.5 in full.

*Proof of Lemma 2.5.* At a high-level, to calculate an approximation of $\mathbb{E}_{x \sim [x]}[f^*(x)^i]$ using $g(x)$, we will calculate $\mathbb{E}_{p \sim g(x)}[M_{k,i}(p)]$ and show that it is a $i\epsilon/2$ approximation of the $i^{\text{th}}$ moment, $\mathbb{E}_{x \sim [x]}[f^*(x)^i]$.

Consider any $x \in \mathcal{X}$ and $i \in [k]$. Using the triangle inequality, we upper bound the difference between our estimate and the true moment value by:

$$\left| \mathbb{E}_{\widehat{p} \sim g(x)}[M_{k,i}(\widehat{p})] - \mathbb{E}_{p_* \sim f^*([x])}[p_*^i] \right|$$

$$\leq \left| \mathbb{E}_{\widehat{p} \sim g(x)}[M_{k,i}(\widehat{p})] - \mathbb{E}_{p \sim \text{proj}_k f^*([x])}[M_{k,i}(p)] \right| + \left| \mathbb{E}_{p \sim \text{proj}_k f^*(x)}[M_{k,i}(p)] - \mathbb{E}_{p_* \sim f^*([x])}[p_*^i] \right|$$

$$= \left| \mathbb{E}_{\widehat{p} \sim g(x)}[M_{k,i}(\widehat{p})] - \mathbb{E}_{p \sim \text{proj}_k f^*([x])}[M_{k,i}(p)] \right| + 0 \qquad \text{(Lemma D.8)}$$

We will now use Lemma D.9 to show that the remaining term is bounded by $m\epsilon/2$.

Let $\Gamma(g(x), \mathrm{proj}_k f^*([x]))$ be the space of couplings of $g(x)$ and $\mathrm{proj}_k f^*([x])$. We can re-express the difference in expectations as

$$\left| \mathop{\mathbb{E}}_{\widehat{\boldsymbol{p}} \sim g(x)}[M_{k,i}(\widehat{\boldsymbol{p}})] - \mathop{\mathbb{E}}_{\boldsymbol{p}_* \sim f^*([x])}[\boldsymbol{p}_*^i] \right| = \left| \mathop{\mathbb{E}}_{\widehat{\boldsymbol{p}} \sim g(x)}[M_{k,i}(\widehat{\boldsymbol{p}})] - \mathop{\mathbb{E}}_{\boldsymbol{p} \sim \mathrm{proj}_k f^*([x])}[M_{k,i}(\boldsymbol{p})] \right|$$

$$= \min_{\gamma \in \Gamma(g(x), \mathrm{proj}_k f^*([x]))} \left| \mathop{\mathbb{E}}_{(\widehat{\boldsymbol{p}}, \boldsymbol{p}) \sim \gamma}[M_{k,i}(\widehat{\boldsymbol{p}}) - M_{k,i}(\boldsymbol{p})] \right|$$

$$\leq \min_{\gamma \in \Gamma(g(x), \mathrm{proj}_k f^*([x]))} \mathop{\mathbb{E}}_{(\widehat{\boldsymbol{p}}, \boldsymbol{p}) \sim \gamma}[i|\widehat{\boldsymbol{p}} - \boldsymbol{p}|] \qquad \text{(Lemma D.9)}$$

$$= i W_1(g(x), \mathrm{proj}_k f^*([x]))/2$$

$$\leq i\epsilon/2 \qquad\qquad (k^{\text{th}}\text{-order calibration})$$

we note that the factor of $1/2$ comes in because $|\widehat{\boldsymbol{p}} - \boldsymbol{p}|$ is $1/2$ of the $\ell_1$ distance when $\widehat{\boldsymbol{p}}$ and $\boldsymbol{p}$ are viewed as distributions over $\{0, 1\}$.

Thus, we've shown an additive error of $i\epsilon/2$ for estimating any moment $i \in [k]$, and thus can construct a vector of such moments for each $i \in [k]$ satisfying the desired guarantee. $\qquad\square$

## D.5 Proof of Theorem 2.6

We restate the theorem for readability:

**Theorem D.10** (Restatement of Theorem 2.6, Empirical estimate of $k^{\text{th}}$-order projection guarantee). *Consider any $x \in \mathcal{X}$, and a sample $\boldsymbol{p}_1, \cdots, \boldsymbol{p}_N \in \mathcal{Y}^{(k)}$ where each $\boldsymbol{p}_i$ is drawn i.i.d. from $\mathrm{proj}_k f^*([x])$. Given $\epsilon > 0$ and $0 \leq \delta \leq 1$, if $N \geq (2(|\mathcal{Y}^{(k)}| \log(2) + \log(1/\delta)))/\epsilon^2$, then we can guarantee that with probability at least $1 - \delta$ over the randomness of the sample we will have*

$$W_1(\mathrm{proj}_k f^*([x]), \mathrm{Unif}(\boldsymbol{p}_1, ..., \boldsymbol{p}_N)) \leq \epsilon.$$

As some discussion before the proof, we note that this sample complexity guarantee is obtained via bounding the total variation distance between $\mathrm{proj}_k f^*([x])$ and $\mathrm{Unif}(\boldsymbol{p}_1, ... \boldsymbol{p}_N)$, which provides an upper bound on the Wasserstein distance. This approach is reasonable for small values of $k$, which we consider to be the most realistic setting where $k$-snapshots might be obtained. When $k$ is large compared to $\ell$, we can alternatively obtain sample complexity bounds depending only on the dimensionality $\ell$ (roughly of the form $(1/\epsilon)^{O(\ell)}$) by directly exploiting known results on the convergence of empirical measures to true measures in Wasserstein distance (Yukich, 1989).

*Proof.* For ease of notation, we will denote $\widehat{\pi} := \mathrm{Unif}(\boldsymbol{p}_1, ..., \boldsymbol{p}_N)$ and $\pi^* := \mathrm{proj}_k f^*([x])$.

We first observe that by Lemma D.1, we are guaranteed that

$$W_1(\widehat{\pi}, \pi^*) \leq 2 d_{TV}(\widehat{\pi}, \pi_*).$$

Thus, we conclude that whenever $d_{TV}(\pi^*, \widehat{\pi}) \leq \epsilon/2$, then we are also guaranteed that

$$W_1(\pi^*, \widehat{\pi}) \leq \epsilon.$$

This reduces our problem to bounding the total variation distance of the empirical estimate of the discrete distribution $\pi^* = \mathrm{proj}_k f^*([x])$, which we observe is supported on $|\mathcal{Y}^{(k)}|$ points.

We now use a standard argument for bounding the total variation distance (see Canonne (2020) for a discussion of various approaches to this argument).

Note that the total variation distance can be alternatively expressed as

$$d_{TV}(\pi^*, \widehat{\pi}) = \sup_{S \subseteq \mathcal{Y}^{(k)}} \left| \Pr_{\boldsymbol{p} \sim \pi^*}[\boldsymbol{p} \in S] - \Pr_{\boldsymbol{p} \sim \widehat{\pi}}[\boldsymbol{p} \in S] \right|.$$

Thus, $d_{TV}(\pi^*, \widehat{\pi}) > \epsilon/2$ iff there exists some $S \subseteq \mathcal{Y}^{(k)}$ such that $\Pr_{\boldsymbol{p} \sim \widehat{\pi}}[\boldsymbol{p} \in S] > \Pr_{\boldsymbol{p} \sim \pi^*}[\boldsymbol{p} \in S] + \epsilon/2$.

We bound the probability that this happens for any $S$ via a union bound. Select any $S \subseteq \mathcal{Y}^{(k)}$, and note that $\Pr_{\boldsymbol{p} \sim \widehat{\pi}}[\boldsymbol{p} \in S]$ can be written as a sum of $N$ i.i.d. Bernoulli random variables $\boldsymbol{X}_1, ..., \boldsymbol{X}_N$, all with mean $\Pr_{\boldsymbol{p} \sim \pi^*}[\boldsymbol{p} \in S]$. This allows us to apply Hoeffding's Inequality to conclude that

$$\Pr\left[\Pr_{\boldsymbol{p} \sim \widehat{\pi}}[\boldsymbol{p} \in S] > \Pr_{\boldsymbol{p} \sim \pi^*}[\boldsymbol{p} \in S] + \epsilon/2\right] = \Pr\left[\frac{1}{N}\sum_{i=1}^{N}\boldsymbol{X}_i > \mathbb{E}[\frac{1}{N}\sum_{i=1}^{N}\boldsymbol{X}_i] + \epsilon/2\right]$$

$$\leq \exp(-N\epsilon^2/2) \qquad \text{(Hoeffding's Inequality)}$$

Thus, for any $N \geq 2(|\mathcal{Y}^{(k)}|\log(2) + \log(1/\delta))/\epsilon^2$, we are guaranteed that

$$\Pr\left[\Pr_{\boldsymbol{p} \sim \widehat{\pi}}[\boldsymbol{p} \in S] \geq \Pr_{\boldsymbol{p} \sim \pi^*}[\boldsymbol{p} \in S] + \epsilon/2\right] \leq \delta/2^{|\mathcal{Y}^{(k)}|}.$$

Union bounding over all $2^{|\mathcal{Y}^{(k)}|}$ possible $S$, we get that

$$\Pr[d_{TV}(\pi^*, \widehat{\pi}) > \epsilon/2] = \Pr\left[\exists S \subseteq \mathcal{Y}^{(k)} \text{s.t.} \Pr_{\boldsymbol{p} \sim \widehat{\pi}}[\boldsymbol{p} \in S] > \Pr_{\boldsymbol{p} \sim \pi^*}[\boldsymbol{p} \in S] + \epsilon/2\right]$$

$$\leq 2^{|\mathcal{Y}^{(k)}|}\frac{\delta}{2^{\mathcal{Y}^{(k)}}} = \delta.$$

Thus, we have shown thatn $N \geq \frac{2(|\mathcal{Y}^{(k)}|\log(2) + \log(1/\delta))}{\epsilon^2}$ samples are sufficient to guarantee that with probability at least $1 - \delta$,

$$W_1(\text{proj}_k f^*([x]), \text{Unif}(\boldsymbol{p}_1, ..., \boldsymbol{p}_N)) \leq \epsilon$$

$\square$

# E    PROOFS AND DISCUSSION FROM SECTION 3

## E.1    PROOF OF LEMMA 3.2

We restate the lemma for readability.

**Lemma E.1.** *(Restatement of Lemma 3.2) If $f : \mathcal{X} \to \Delta\Delta\mathcal{Y}$ is perfectly higher-order calibrated, then for all $x \in \mathcal{X}$,*
$$\text{AU}_G(f : x) = \mathbb{E}_{\boldsymbol{x} \sim [x]}[\text{AU}_G^*(\boldsymbol{x})].$$

*Proof.* Since $f(x) = f^*([x])$ as a mixture,
$$\text{AU}_G(f : x) = \mathbb{E}_{\boldsymbol{p} \sim f(x)}[G(\boldsymbol{p})] = \mathbb{E}_{\boldsymbol{p}_* \sim f^*([x])}[G(\boldsymbol{p}_*)] = \mathbb{E}_{\boldsymbol{x} \sim [x]}[G(f^*(\boldsymbol{x}))] = \mathbb{E}_{\boldsymbol{x} \sim [x]}[\text{AU}_G^*(\boldsymbol{x})].$$

$\square$

## E.2    EQUIVALENCE BETWEEN PERFECT HIGHER-ORDER CALIBRATION AND CALIBRATED UNCERTAINTY ESTIMATES

At first sight, the ability to produce calibrated estimates of aleatoric uncertainty may appear to be merely one small consequence of higher-order calibration. But somewhat remarkably, it turns out that having accurate estimates of AU wrt all concave generalized entropy functions is *equivalent* to higher-order calibration.

Such a statement requires a definition of what entails a "concave generalized entropy function." We take the most general view possible, building on the work of Gneiting & Raftery (2007), who show that every proper loss has an associated generalized concave entropy function, and in particular every concave function $G : \Delta\mathcal{Y} \to \mathbb{R}$ is associated with some proper loss (see Section C.1 for a more in-depth discussion of proper losses and their associated entropy functions). From this point of view, the class of generalized concave entropy functions is exactly the set of all concave functions. Under this definition, we get an equivalence between calibrated estimates of concave entropy functions and higher-order calibration.

**Theorem E.2.** *A higher-order predictor $f : \mathcal{X} \to \Delta\Delta\mathcal{Y}$ is perfectly higher-order calibrated to $f^*$ wrt a partition $[\cdot]$ if and only if $\mathrm{AU}_G(f : x) = \mathbb{E}_{\boldsymbol{x}\sim[x]}[\mathrm{AU}_G^*(\boldsymbol{x})]$ wrt all concave functions $G : \Delta\mathcal{Y} \to \mathbb{R}$.*

*Proof.* The higher-order calibration implies $\mathrm{AU}_G(f : x) = \mathbb{E}_{\boldsymbol{x}\sim[x]}[\mathrm{AU}_G^*(\boldsymbol{x})]$ direction is proved in Lemma 3.2. We focus on the reverse direction.

We will leverage the notion of *moment generating functions*. The moment generating function of a random variable $\boldsymbol{X} \in \mathbb{R}^\ell$ is a function $M_{\boldsymbol{X}} : \mathbb{R}^\ell \to \mathbb{R}$ defined by $M_{\boldsymbol{X}}(t) = \mathbb{E}_{\boldsymbol{X}}[e^{t^T\boldsymbol{X}}]$. An important property of the moment generating function is that if $M_{\boldsymbol{X}}(t)$ is finite for all $t \in [-c, c]^\ell$ for some $c > 0$, then its values *uniquely determine* the distribution of $\boldsymbol{X}$ (DasGupta (2010), Theorem 11.8). In other words, if we have two random variables $\boldsymbol{X}$ and $\boldsymbol{Y}$ such that $M_{\boldsymbol{X}}(t) = M_{\boldsymbol{Y}}(t) < \infty$ for all $t \in [-c, c]^\ell$, then $\boldsymbol{X}$ and $\boldsymbol{Y}$ must be the same distribution.

We will use this line of reasoning to show that the predicted and true mixture for some partition $[x]$ must be the same if their estimates of all concave functions are identical.

Fix some $x \in X$, and consider the predicted mixture $\boldsymbol{p} \sim f(x)$ as well as the true mixture $\boldsymbol{p}^* \sim f^*([x])$.

We consider the set of functions $\mathcal{G} = \{G_t\}_{t\in[-1,1]^\ell}$ where each $G_t : \Delta\mathcal{Y} \to \mathbb{R}$ is defined as $G_t(p) := -e^{t^T p}$.[16]

Note that by definition, for any $t \in [-1, 1]^\ell$,

$$\mathrm{AU}_{G_t}(f : x) = \underset{\boldsymbol{p}\sim f(x)}{\mathbb{E}}[G_t(\boldsymbol{p})] = \underset{\boldsymbol{p}\sim f(x)}{\mathbb{E}}[-e^{t^T\boldsymbol{P}}] = -M_{\boldsymbol{p}}(t) \tag{E.1}$$

and similarly

$$\underset{\boldsymbol{x}\sim[x]}{\mathbb{E}}[\mathrm{AU}_{G_t}^*(\boldsymbol{x})] = \underset{\boldsymbol{p}^*\sim f^*([x])}{\mathbb{E}}[G_t(\boldsymbol{p}^*)] = \underset{\boldsymbol{p}^*\sim f^*([x])}{\mathbb{E}}[-e^{t^T\boldsymbol{p}^*}] = -M_{\boldsymbol{p}^*}(t) \tag{E.2}$$

We make two assumptions which we will prove later:

1. Every $G_t \in \mathcal{G}$ is concave.

2. $\mathrm{AU}_{G_t}(f : x)$ and $\mathbb{E}_{\boldsymbol{x}\sim[x]}[\mathrm{AU}_{G_t}^*(\boldsymbol{x})]$ are finite for all $G_t \in \mathcal{G}$.

With these two assumptions, the proof is quickly complete. In particular, if all $G_t$ are concave, then by the main assumption of our theorem, we have

$$\mathrm{AU}_{G_t}(f : x) = \underset{\boldsymbol{x}\sim[x]}{\mathbb{E}}[\mathrm{AU}_{G_t}^*(\boldsymbol{x})]$$

for all $G_t \in \mathcal{G}$. The equivalences shown in E.1 and E.2 imply that we also have

$$M_{\boldsymbol{p}}(t) = M_{\boldsymbol{p}*}(t)$$

for all $t \in [-1, 1]^\ell$. Our additional finiteness assumption implies that these moment generating functions are finite for all $t \in [-1, 1]^\ell$, and thus by the uniqueness of moment generating functions, we conclude that the distributions $f(x)$ and $f^*([x])$ are identical. Because this holds for any $x \in \mathcal{X}$, we conclude that $f(x)$ is perfectly higher-order calibrated.

It remains to prove our assumptions of concavity and finiteness.

We first show finiteness, which follows almost immediately because $\Delta\mathcal{Y}$ is bounded, and thus for any $p \in \Delta\mathcal{Y}$ and $t \in [-c, c]^\ell$, we have $t^T p \in [-c, c]$ and thus $-e^{t^T p} \in [e^{-c}, e^c]$. Thus the expectation of $e^{t^T\boldsymbol{p}}$ for any random variable $\boldsymbol{p}$ over $\Delta\mathcal{Y}$ is guaranteed to be finite.

---

[16]Note that scaling by an additive constant can ensure this function is non-negative on $\Delta\mathcal{Y}$ if we would like to require that entropy functions are non-negative, without changing the result of the proof.

We finally show concavity. Consider any $G_t \in \mathcal{G}$ and $p_1, p_2 \in \Delta \mathcal{Y}$, $\lambda \in [0, 1]$. We have

$$G_t(\lambda p_1 + (1 - \lambda)p_2) = -e^{t^T(\lambda p_1 + (1-\lambda)p_2)}$$
$$= -e^{\lambda t^T p_1 + (1-\lambda)t^T p_2}$$

because $-e^X$ is a concave function, we have that

$$-e^{\lambda t^T p_1 + (1-\lambda)t^T p_2} \geq -\lambda e^{t^T p_1} - (1 - \lambda)e^{t^T p_2}$$
$$= \lambda G_t(p_1) + (1 - \lambda)G_t(p_2)$$

Thus, we conclude that $G_t(\lambda p_1 + (1 - \lambda)p_2) \geq \lambda G_t(p_1) + (1 - \lambda)G_t(p_2)$, and so each $G_t \in \mathcal{G}$ is concave. $\qquad\square$

We add some additional discussion of more stringent definitions of what qualifies as a generalized entropy function. Above, our only requirement was concavity, though we note that the nature of the proof suggests that this class could be further restricted to the set of continuous, concave, non-negative functions on $\Delta \mathcal{Y}$.

Another potential requirement for entropy functions is symmetry—invariance to permutations of the class probabilities. We highlight that requiring symmetry would break the equivalence between calibrated entropy estimates and higher-order calibration. To illustrate this, consider a simple counterexample:

Assume $f^*(x)$ has no aleatoric uncertainty for each $x$, and thus assigns probability 1 to some class $i \in [\ell]$. Now, consider a higher-order predictor that, for each $x$, predicts a point mixture concentrated on a distribution that gives 100% probability to some class $j \in [\ell]$. Under any symmetric entropy function, this predictor would appear to have perfect aleatoric uncertainty estimates. Both $f^*$ and the predictor assign 100% probability to a single class for each $x$. However, this predictor is far from being higher-order calibrated, as it may consistently predict the wrong class for every $x \in \mathcal{X}$.

### E.3    Proofs associated with Theorem 3.3

In this section, we present proofs associated with the statement of Theorem 3.3. We first prove the last statement of the theorem, which follows from a general guarantee in terms of a concave entropy function's modulus of continuity.

**Definition E.3** (Uniform continuity and modulus of continuity). Consider a function $G : \Delta \mathcal{Y} \to \mathbb{R}$, where we view $\Delta \mathcal{Y}$ as a subset of $\mathbb{R}^\ell$ equipped with the $\ell_1$ distance metric. We say that $G$ is *uniformly continuous* if there exists a function $\omega_G : \mathbb{R}_{\geq 0} \to \mathbb{R}_{\geq 0}$ vanishing at 0 such that for all $p, p' \in \Delta \mathcal{Y}$, we have
$$|G(p) - G(p')| \leq \omega_G(\|p - p'\|_1).$$
The function $\omega_G$ is called the *modulus of continuity* of $G$.

*Remark* E.4. Uniform continuity can be usefully thought of as a relaxation of Lipschitzness. The latter is the special case where $\omega_G(\delta) \leq O(\delta)$. We will always be concerned with the behavior of $\omega_G$ for small values of $\delta \ll 1$, and in this regime we may assume without loss of generality that $\omega_G(\delta) \leq O(\delta^\alpha)$ for some $\alpha$. Moreover, we may assume that $\alpha \leq 1$, simply because if $\alpha > 1$ then $\delta^\alpha \leq \delta$ (for small $\delta$). In particular, $\omega_G$ can be assumed WLOG to be a concave, non-decreasing function for small $\delta$.

With this notation in hand, we are now ready to present the statement in full. The proof is provided in the following section (E.3.1).

**Theorem E.5** (Informally stated in last point of Theorem 3.3). *Consider a $k^{th}$-order predictor $g : \mathcal{X} \to \Delta \mathcal{Y}^{(k)}$ that satisfies $\epsilon$-$k^{th}$-order calibration with respect to a partition $[\cdot]$. Let $G$ be any concave entropy function that satisfies uniform continuity with modulus of continuity $\omega_G : \mathbb{R}_{\geq 0} \to \mathbb{R}_{\geq 0}$. Then, $g$'s estimate of aleatoric uncertainty with respect to $G$ satisfies the following guarantee for all $x \in \mathcal{X}$:*
$$\left| \mathrm{AU}_G(g : x) - \mathop{\mathbb{E}}_{\boldsymbol{x} \sim [x]}[\mathrm{AU}_G^*(\boldsymbol{x})] \right| \leq \omega_G(\epsilon + \frac{|\mathcal{Y}|}{2\sqrt{k}}).$$

We now move on to the first two statements of Theorem 3.3, both of which follow from an alternative means of estimating the true aleatoric uncertainty under $k^{\text{th}}$-order calibration using polynomial approximation.

In the rest of this section we specialize to binary labels $\mathcal{Y} = \{0, 1\}$, where $\Delta \mathcal{Y}$ may be identified with $[0, 1]$, and $G : [0, 1] \to \mathbb{R}$ becomes a simple real-valued function on the unit interval. We now state Jackson's well-known theorem from approximation theory on approximating such functions using polynomials.

**Definition E.6.** We say that $G : [0, 1] \to \mathbb{R}$ admits a $(d, \alpha, B)$-polynomial approximation if there exists a degree $d$ polynomial $p(t) = \sum_{i=0}^{d} \beta_i t^i$ such that $|\beta_i| \le B$ for all $i$ and

$$\sup_{t \in [0,1]} |p(t) - G(t)| \le \alpha.$$

**Theorem E.7** (Jackson's theorem; see e.g. (Rivlin, 1981, Thm 1.4)). *Let $G : [0, 1] \to \mathbb{R}$ be a uniformly continuous function with modulus of continuity $\omega_G$. Then for any $d > 0$, $G$ admits a $(d, O(\omega_G(\frac{1}{d})), \exp(\Theta(d)))$-polynomial approximation.*

The bound on the coefficients can be shown using standard bounds on the coefficients of the Chebyshev polynomials, but also holds more generally for any polynomial $F$ that is bounded on the interval (see e.g. (Natanson, 1964, Cor 2, p56)).

We now give an alternative to Theorem E.5 that can provide better error guarantees for particular choices of $G$ in regimes where $k$ is quite small, and thus the $\frac{|\mathcal{Y}|}{2\sqrt{k}}$ term arising in that theorem could dominate the overall error. Recall that we are working with the binary case ($\mathcal{Y} = \{0, 1\}$) for simplicity. In this case our mixture gives us a random variable on $[0, 1]$, and we have the following theorem:

**Theorem E.8.** *Let $\mathcal{Y} = \{0, 1\}$ and let $G : [0, 1] \to \mathbb{R}$ be a concave generalized entropy function that has a $(d, \alpha, B)$-polynomial approximation. Then, if $g : \mathcal{X} \to \Delta \mathcal{Y}^{(k)}$ is an $\epsilon$-$k^{\text{th}}$-order calibrated predictor for any $k \ge d$, then for each $x \in \mathcal{X}$, we can use $g$ to estimate $\mathbb{E}_{\boldsymbol{x} \sim [x]}[\text{AU}_G^*(\boldsymbol{x})]$ to within an additive error $\delta = \alpha + d^2 \epsilon B / 2$.*

The full proof can be found in Section E.3.2. To apply this theorem, we need to show that commonly used entropy functions have good polynomial approximations. We show that this is indeed the case for the examples considered earlier. The Brier entropy or Gini impurity $G_{\text{Brier}}(p) = 4p(1 - p)$ is trivial since it is itself a quadratic. Hence we get the following corollary (informally stated as the first bullet point in Theorem 3.3):

**Corollary E.9** (Corollary to Theorems E.8 and 2.6, Estimating Brier Entropy). *Let $\epsilon > 0$. Let $g : \mathcal{X} \to \Delta \mathcal{Y}^{(2)}$ be an $(\epsilon/8)$-second-order calibrated predictor. Let $G_{Brier}(p) = 4p(1 - p)$ denote the Brier entropy. Then we can use $g$ to obtain an estimate $\widehat{\text{AU}}_{Brier}$ such that with high probability*

$$\left| \widehat{\text{AU}}_{Brier} - \mathbb{E}_{\boldsymbol{x} \sim [x]}[\text{AU}_{G_{Brier}}^*(\boldsymbol{x})] \right| \le \epsilon.$$

*In particular, we require only $N \ge 128(4 \log(2) + \log(1/\delta))/\epsilon^2$ 2-snapshot examples from $[x]$ for this guarantee to hold with probability at least $1 - \delta$.*

*Proof.* The Brier entropy presents a particularly easy case of Theorem E.8 because $G_{\text{Brier}}(p) = 4p(1 - p)$ is itself a polynomial of degree 2 with coefficients bounded in absolute value by 4.

Theorem E.8 thus guarantees that an $(\epsilon/8)$-second-order calibrated predictor can estimate $\mathbb{E}_{\boldsymbol{x} \sim [x]}[\text{AU}_{G_{\text{Brier}}}^*(f^*(\boldsymbol{x}))]$ to within an additive error of $\epsilon$.

Theorem 2.6 tells us that $N \ge 128(4 \log(2) + \log(1/\delta))/\epsilon^2$ samples from $[x]$ are sufficient to guarantee an $\epsilon/8$-second-order-calibrated prediction for $[x]$ with probability at least $1 - \delta$, thus giving an additive error of $\epsilon$ when estimating the aleatoric uncertainty. $\square$

In the case of the Shannon entropy, we get the following corollary, captured in the second statement of Theorem 3.3:

**Corollary E.10** (Corollary to Theorems E.8 and 2.5, Estimating Shannon Entropy). *Let $\epsilon > 0$. Let $g : \mathcal{X} \to \Delta\mathcal{Y}^{(k)}$ be an $\epsilon'$-$k^{th}$-order calibrated predictor where $k \geq \Theta((\frac{1}{\epsilon})^{\ln 4})$, and $\epsilon' \leq \frac{\epsilon}{\exp(\Theta(k))}$. Let $G_{Shannon}(p) = -p \log p - (1-p) \log(1-p)$ denote the Shannon entropy. Then we can use $g$ to obtain an estimate $\widehat{\mathrm{AU}}_{Shannon}$ such that with high probability,*

$$\left| \widehat{\mathrm{AU}}_{Shannon} - \mathop{\mathbb{E}}_{\boldsymbol{x} \sim [x]}[\mathrm{AU}^*_{G_{Shannon}}(\boldsymbol{x})] \right| \leq \epsilon.$$

*In particular, we require only $N \geq O(\log(1/\delta)\exp(O((1/\epsilon)^{\ln 4})))$ $k$-snapshot examples from $[x]$ for this guarantee to hold with probability at least $1 - \delta$.*

*Proof.* The Shannon entropy $G_{\mathrm{Shannon}}(p) = -p \log p - (1-p) \log(1-p)$ can be dealt with using Jackson's theorem (Theorem E.7), which tells us that for any $d \geq 1$, there exists a degree-$d$ polynomial $F$ and constant $C_1$ such that

$$\sup_{x \in [0,1]} |F(x) - G_{\mathrm{Shannon}}(x)| \leq C_1 \omega_G(1/d),$$

and the coefficients of $F$ are bounded by $\exp(C_2 d)$ for some constant $C_2$. Here $\omega_G$ is the modulus of continuity of $G_{\mathrm{Shannon}}$ (Definition E.3), defined by

$$\omega_G(x) = \sup\{|G_{\mathrm{Shannon}}(p) - G_{\mathrm{Shannon}}(p')| \mid p, p' \in [0,1], |p - p'| \leq x\}.$$

In the case of the binary entropy function, it is easy to see that the supremum is achieved at the endpoints, e.g. at $p = 0, p' = x$, and so $\omega_G(x) = G_{\mathrm{Shannon}}(x)$. Now, it turns out that the binary entropy function satisfies the following useful bound (Topsøe, 2001):

$$G_{\mathrm{Shannon}}(x) \leq (4x(1-x))^{1/\ln 4} \leq (4x)^{1/\ln 4}.$$

Thus, to ensure $C_1 \omega_G(1/d) \leq \epsilon/2$, it suffices to take $k \geq d \geq \frac{1}{4}\left(\frac{2C_1}{\epsilon}\right)^{\ln 4}$. We choose the minimal snapshot size that can achieve this, and take $k = \frac{1}{4}\left(\frac{2C_1}{\epsilon}\right)^{\ln 4}$.

By Theorem E.8, for this value of $k$ we are guaranteed that an $\epsilon'$-$k^{th}$-order calibrated predictor will give estimates of the aleatoric uncertainty with additive error at most $\epsilon/2 + k^2 \epsilon' e^{C_2 k}/2$.

Thus, there exists a constant $C_3$ such that it suffices to have $\epsilon' \leq \frac{\epsilon}{e^{C_3 k}}$ to guarantee an error of at most $\epsilon$. Plugging this bound into Theorem 2.6 tells us that there exists a constant $C_4$ such that $N \geq \exp(C_4 (1/\epsilon)^{\ln 4}) \log(1/\delta)$ samples from $[x]$ are enough to guarantee that the $k^{th}$-order calibration error in that partition is at most $\frac{\epsilon}{e^{C_3 k}}$ with probability at least $1 - \delta$, thus guaranteeing that the overall additive approximation error of estimating the Shannon entropy on $[x]$ will be at most $\epsilon$ with probability at least $1 - \delta$.

We complete the proof by noting that $\exp(C_4 (1/\epsilon)^{\ln 4}) \log(1/\delta) = O(\log(1/\delta)\exp(O((1/\epsilon)^{\ln 4})))$.
$\square$

### E.3.1 PROOF OF THEOREM E.5

We restate the theorem for readability:

**Theorem E.11** (Restatement of Theorem E.5). *Consider a $k^{th}$-order predictor $g : \mathcal{X} \to \Delta\mathcal{Y}^{(k)}$ that satisfies $\epsilon$-$k^{th}$-order calibration with respect to a partition $[\cdot]$. Let $G$ be any concave entropy function that satisfies uniform continuity with modulus of continuity $\omega_G : \mathbb{R}_{\geq 0} \to \mathbb{R}_{\geq 0}$. Then, $g$'s estimate of aleatoric uncertainty with respect to $G$ satisfies the following guarantee for all $x \in \mathcal{X}$:*

$$\left| \mathrm{AU}_G(g : x) - \mathop{\mathbb{E}}_{\boldsymbol{x} \sim [x]}[\mathrm{AU}^*_G(\boldsymbol{x})] \right| \leq \omega_G(\epsilon + \frac{|\mathcal{Y}|}{2\sqrt{k}}).$$

*Proof of Theorem E.5.* Consider any $x \in \mathcal{X}$. Using the definition of aleatoric uncertainty with respect to $G$, we can rewrite the distance between the true and estimated AU as

$$\left| \mathrm{AU}_G(g : x) - \mathop{\mathbb{E}}_{\boldsymbol{x} \sim [x]}[\mathrm{AU}^*_G(\boldsymbol{x})] \right| = \left| \mathop{\mathbb{E}}_{\boldsymbol{p} \sim g(x)}[G(\boldsymbol{p})] - \mathop{\mathbb{E}}_{\boldsymbol{p}^* \sim f^*([x])}[G(\boldsymbol{p}^*)] \right|$$

Let $\mu$ be the optimal coupling of $g(x)$ and $f^*([x])$, i.e.,

$$\mu := \underset{\mu' \in \Gamma(g(x), f^*([x]))}{\arg\min} \underset{(\boldsymbol{p}, \boldsymbol{p}^*) \sim \mu'}{\mathbb{E}} [\|\boldsymbol{p} - \boldsymbol{p}^*\|_1].$$

We can rewrite the above expression in terms of $\mu$ as

$$
\begin{aligned}
& \left| \mathrm{AU}_G(g : x) - \underset{\boldsymbol{x} \sim [x]}{\mathbb{E}}[\mathrm{AU}_G^*(\boldsymbol{x})] \right| \\
&= \left| \underset{(\boldsymbol{p}, \boldsymbol{p}^*) \sim \mu}{\mathbb{E}} [G(\boldsymbol{p}) - G(\boldsymbol{p}^*)] \right| \\
&\leq \underset{(\boldsymbol{p}, \boldsymbol{p}^*) \sim \mu}{\mathbb{E}} [|G(\boldsymbol{p}) - G(\boldsymbol{p}^*)|] \\
&\leq \underset{(\boldsymbol{p}, \boldsymbol{p}^*) \sim \mu}{\mathbb{E}} [\omega_G(\|\boldsymbol{p} - \boldsymbol{p}^*\|_1)] && \text{(uniform continuity of } G) \\
&\leq \omega_G\left( \underset{(\boldsymbol{p}, \boldsymbol{p}^*) \sim \mu}{\mathbb{E}} [\|\boldsymbol{p} - \boldsymbol{p}^*\|_1] \right) && \text{(Jensen's Inequality + Concavity of } \omega_G)
\end{aligned}
$$

See Remark E.4 for a discussion of why we can assume that $\omega_G$ is concave.

Because $g$ is $\epsilon$-$k^{\text{th}}$-order calibrated, by Theorem 2.4, we can guarantee that it is also $(\epsilon + \frac{|\mathcal{Y}|}{2\sqrt{k}})$-higher-order calibrated, and thus we are guaranteed that $W_1(g(x), f^*([x])) \leq \epsilon + \frac{|\mathcal{Y}|}{2\sqrt{k}}$.

Expanding out the definition of $\ell_1$-Wasserstein distance, because $\mu$ was defined as the optimal coupling of $g(x)$ and $f^*([x])$, we have

$$W_1(g(x), f^*([x])) = \underset{\mu' \in \Gamma(g(x), f^*([x]))}{\arg\min} \underset{(\boldsymbol{p}, \boldsymbol{p}^*) \sim \mu'}{\mathbb{E}} [\|\boldsymbol{p} - \boldsymbol{p}^*\|_1] = \underset{(\boldsymbol{p}, \boldsymbol{p}^*) \sim \mu}{\mathbb{E}} [\|\boldsymbol{p} - \boldsymbol{p}^*\|_1] \leq \epsilon + \frac{|\mathcal{Y}|}{2\sqrt{k}}.$$

We can use this upper bound to further simplify our above inequality as

$$
\begin{aligned}
\left| \mathrm{AU}_G(g : x) - \underset{\boldsymbol{x} \sim [x]}{\mathbb{E}}[\mathrm{AU}_G^*(\boldsymbol{x})] \right| &\leq \omega_G\left( \underset{(\boldsymbol{p}, \boldsymbol{p}^*) \sim \mu}{\mathbb{E}} [\|\boldsymbol{p} - \boldsymbol{p}^*\|_1] \right) \\
&\leq \omega_G\left( \epsilon + \frac{|\mathcal{Y}|}{2\sqrt{k}} \right),
\end{aligned}
$$

giving us the desired upper bound on the error of our aleatoric uncertainty estimate. Note that the final step assumes $\omega_G$ is non-decreasing. We refer to Remark E.4 for a discussion of why this is a reasonable assumption. $\qquad\square$

### E.3.2 PROOF OF THEOREM E.8

**Theorem E.12** (Restatement of Theorem E.8). *Let $\mathcal{Y} = \{0, 1\}$ and let $G : [0, 1] \to \mathbb{R}$ be a concave generalized entropy function that has $(d, \alpha, B)$-polynomial approximations. Then, if $g : \mathcal{X} \to \Delta\mathcal{Y}^{(k)}$ is an $\epsilon$-$k^{\text{th}}$-order calibrated predictor for any $k \geq d$, then for each $x \in \mathcal{X}$, we can use $g$ to estimate $\mathbb{E}_{\boldsymbol{x} \sim [x]}[\mathrm{AU}_G^*(\boldsymbol{x})]$ to within an additive error $\delta = \alpha + d^2 \epsilon B / 2$.*

At a high-level, the proof will leverage Lemma 2.5, which tells us that $k^{\text{th}}$-order calibrated predictors can be used to obtain good estimates for the first $k$ moments of the true bayes mixture. Because a degree-d polynomial of a random variable $\boldsymbol{p}$ can be computed using only the first $d$ moments of $\boldsymbol{p}$, we can use our moment estimates to compute any degree $d \leq k$ polynomial, which will provide a good estimate of aleatoric uncertainty when that polynomial closely approximates $G$.

*Proof of Theorem E.8.* Fix some $x \in \mathcal{X}$. By assumption of the theorem, $G$ has $(d, \alpha, B)$-polynomial approximations. This means that there exists a degree-$d$ polynomial $c(t) = \sum_{i=0}^d \beta_i t^i$ such that for all $i \in \{0, ..., d\}$, $|\beta_i| \leq B$, and for all $t \in [0, 1]$, $|c(t) - G(t)| \leq \alpha$.

The polynomial $c$ can thus be used to approximate $\mathbb{E}_{\boldsymbol{x}\sim[x]}[\mathrm{AU}_G^*(\boldsymbol{x})]$ to within an additive error of $\alpha$:

$$
\begin{aligned}
\left|\mathop{\mathbb{E}}_{x\sim[x]}[c(f^*(\boldsymbol{x}))] - \mathop{\mathbb{E}}_{\boldsymbol{x}\sim[x]}[\mathrm{AU}_G^*(\boldsymbol{x})]\right| &= \left|\mathop{\mathbb{E}}_{x\sim[x]}[c(f^*(\boldsymbol{x}))] - \mathop{\mathbb{E}}_{\boldsymbol{x}\sim[x]}[G(f^*(\boldsymbol{x}))]\right| \\
&\leq \mathop{\mathbb{E}}_{x\sim[x]}[|c(f^*(\boldsymbol{x})) - G(f^*(\boldsymbol{x}))|] \\
&\leq \mathop{\mathbb{E}}_{x\sim[x]}[\alpha] \\
&= \alpha
\end{aligned}
$$

We now want to show that we can obtain a good estimate of $\mathbb{E}_{x\sim[x]}[c(f^*(\boldsymbol{x}))]$ only with access to $g$. We expand out the definition of $c$ to have

$$
\begin{aligned}
\mathop{\mathbb{E}}_{x\sim[x]}[c(f^*(\boldsymbol{x}))] &= \mathop{\mathbb{E}}_{x\sim[x]}\left[\sum_{i=0}^{d}\beta_i(f^*(\boldsymbol{x}))^i\right] \\
&= \sum_{i=0}^{d}\beta_i\mathop{\mathbb{E}}_{x\sim[x]}[f^*(\boldsymbol{x})^i]
\end{aligned}
$$

Let $m_1, ..., m_d \in \mathbb{R}_{\geq 0}$ be the moment estimates of $\mathbb{E}_{x\sim[x]}[f^*(\boldsymbol{x})^1], ..., \mathbb{E}_{x\sim[x]}[f^*(\boldsymbol{x})^d]$, respectively that are obtained from $g$ via Lemma 2.5. Because $d \leq k$, for each $i \in [d]$, the lemma guarantees that

$$
\left|m_i - \mathop{\mathbb{E}}_{x\sim[x]}[f^*(\boldsymbol{x})^i]\right| \leq i\epsilon/2 \leq d\epsilon/2. \tag{E.3}
$$

Substituting in each of these moment estimates for $\mathbb{E}_{x\sim[x]}[f^*(\boldsymbol{x})^i]$ in the computation of $c$ will give a $d^2\epsilon B/2$-additive approximation of $\mathbb{E}_{x\sim[x]}[c(f^*(\boldsymbol{x}))]$:

$$
\begin{aligned}
\left|\mathop{\mathbb{E}}_{x\sim[x]}[c(f^*(\boldsymbol{x}))] - \left(\beta_0 + \sum_{i=1}^{d}\beta_i m_i\right)\right| &= \left|\sum_{i=0}^{d}\beta_i\mathop{\mathbb{E}}_{x\sim[x]}[f^*(\boldsymbol{x})^i] - \left(\beta_0 + \sum_{i=1}^{d}\beta_i m_i\right)\right| \\
&\leq \sum_{i=1}^{d}|\beta_i|\left|\mathop{\mathbb{E}}_{x\sim[x]}[f^*(\boldsymbol{x})^i] - m_i\right| \\
&\leq B\sum_{i=1}^{d}\left|\mathop{\mathbb{E}}_{x\sim[x]}[f^*(\boldsymbol{x})^i] - m_i\right| \quad (|\beta_i| \leq B \text{ for all } i \in [d]) \\
&\leq B\sum_{i=1}^{d}d\epsilon/2 \tag{E.3} \\
&= Bd^2\epsilon/2.
\end{aligned}
$$

Putting all our pieces together, we show that $\beta_0 + \sum_{i=1}^{d}\beta_i m_i$, which is computed only using $g$, gives an additive $\alpha + d^2 B\epsilon/2$-approximation to $\mathbb{E}_{\boldsymbol{x}\sim[x]}[\mathrm{AU}_G^*(\boldsymbol{x})]$. By the triangle inequality,

$$
\begin{aligned}
&\left|\mathop{\mathbb{E}}_{\boldsymbol{x}\sim[x]}[\mathrm{AU}_G^*(\boldsymbol{x})] - \left(\beta_0 + \sum_{i=1}^{d}\beta_i m_i\right)\right| \\
&\leq \left|\mathop{\mathbb{E}}_{\boldsymbol{x}\sim[x]}[\mathrm{AU}_G^*(\boldsymbol{x})] - \mathop{\mathbb{E}}_{x\sim[x]}[c(f^*(\boldsymbol{x}))]\right| + \left|\mathop{\mathbb{E}}_{x\sim[x]}[c(f^*(\boldsymbol{x}))] - \left(\beta_0 + \sum_{i=1}^{d}\beta_i m_i\right)\right| \\
&\leq \alpha + Bd^2\epsilon/2
\end{aligned}
$$

giving the desired additive approximation.

$\square$

## F  HIGHER-ORDER PREDICTION SETS

In this section we describe how to obtain prediction intervals, or more generally prediction sets, from higher-order calibration. These prediction sets have the property that they capture the true $f^*(\boldsymbol{x})$ with a certain prescribed probability when $\boldsymbol{x}$ is drawn conditionally at random from an equivalence class. We stress that these are not prediction sets for *outcomes* at a particular instance, but rather higher-order prediction sets for entire *ground truth probability vectors* ranging over a group of instances; they are subsets of the simplex $\Delta\mathcal{Y}$ and not of $\mathcal{Y}$. In this way they provide an operational way of expressing and using our epistemic (rather than merely predictive) uncertainty.[17]

**Definition F.1** (Higher-order prediction set). A set $S \subseteq \Delta\mathcal{Y}$ is said to be a *higher-order prediction set* for the Bayes mixture $f^*([x])$ with *coverage* $1 - \alpha$ if the following holds:

$$\mathbb{P}_{\boldsymbol{x} \sim [x]}[f^*(\boldsymbol{x}) \in S] = 1 - \alpha.$$

The main idea is simple. If $f$ is perfectly higher-order calibrated for a certain partition $[\cdot]$, then for every $x$, $f(x)$ matches $f^*([x])$ exactly as a mixture. Thus any set $S$ that captures $1 - \alpha$ mass under $f(x)$ also captures $1 - \alpha$ mass under $f^*([x])$:

$$\mathbb{P}_{\boldsymbol{x} \sim [x]}[f^*(\boldsymbol{x}) \in S] = \mathbb{P}_{\boldsymbol{p}^* \sim f^*([x])}[\boldsymbol{p}^* \in S] = \mathbb{P}_{\boldsymbol{p} \sim f(x)}[\boldsymbol{p} \in S] = 1 - \alpha. \tag{F.1}$$

In this way we can take our prediction $f(x)$ (which in principle we have a complete description of), take any set $S$ that captures $1 - \alpha$ of its mass (there are many natural ways of doing so), and use it directly as a $1 - \alpha$ prediction set for $f^*(\boldsymbol{x})$ when $\boldsymbol{x} \sim [x]$.

This argument becomes slightly more technically involved when we have only approximate higher-order or $k^{\text{th}}$-order calibration. Essentially, if we only have Wasserstein closeness between $f(x)$ and $f^*([x])$, then we need to enlarge the set $S$ slightly in order to account for the fact that a typical draw from $f(x)$ is slightly far from a corresponding (coupled) draw from $f^*([x])$. For intuition, consider the binary labels setting, where $f(x)$ and $f^*([x])$ both reduce to distributions on $[0, 1]$. Suppose that the density of $f(x)$ is exactly that of $f^*([x])$ but just shifted by $\epsilon$ (this is a particularly simple case of $\epsilon$-Wasserstein closeness). In this case we simply need to consider an $\epsilon$-neighborhood of a $1 - \alpha$ set under $f(x)$ to obtain $1 - \alpha$ coverage under $f^*([x])$.

We now formalize this idea. As in the rest of the paper, we specialize to the $\ell_1$-Wasserstein distance.

**Definition F.2** (Neighborhood of a set). Let $S \subseteq \Delta\mathcal{Y}$ be a subset of the simplex, regarded as probability vectors in $\mathbb{R}^\ell$. Then for any $\delta > 0$, the $\delta$-neighborhood of $S$ is all vectors with $\ell_1$-distance at most $\delta$ from $S$:

$$S_\delta = \{p \in \Delta\mathcal{Y} \mid \exists p' \in S : \|p - p'\|_1 \leq \delta\}.$$

**Theorem F.3** (Higher-order prediction sets from higher-order calibration). *Let $f : \mathcal{X} \to \Delta\Delta\mathcal{Y}$ be $\epsilon$-higher-order calibrated wrt a partition $[\cdot]$. Fix any $x \in \mathcal{X}$ and let $\pi = f(x), \pi^* = f^*([x])$. Then for any set $S \subseteq \Delta\mathcal{Y}$ and any $\delta > 0$, we have*

$$\mathbb{P}_{\boldsymbol{p} \sim \pi}[\boldsymbol{p} \in S] - \frac{\epsilon}{\delta} \leq \mathbb{P}_{\boldsymbol{p}^* \sim \pi^*}[\boldsymbol{p}^* \in S_\delta] \leq \mathbb{P}_{\boldsymbol{p} \sim \pi}[\boldsymbol{p} \in S_{2\delta}] + \frac{\epsilon}{\delta}. \tag{F.2}$$

*In particular, if $S$ contains $1 - \alpha$ of the mass under $f(x)$, then $S_\delta$ is a prediction set for $f^*([x])$ with coverage at least $1 - \alpha - \frac{\epsilon}{\delta}$.*

*Proof.* Since $f$ is $\epsilon$-higher-order calibrated, we know that $W_1(\pi, \pi^*) \leq \epsilon$. Let $\mu$ be the corresponding optimal coupling of $\boldsymbol{p}, \boldsymbol{p}^*$, with marginals being $\pi, \pi^*$ respectively, and guaranteeing that

$$\mathbb{E}_{(\boldsymbol{p}, \boldsymbol{p}^*) \sim \mu}[\|\boldsymbol{p} - \boldsymbol{p}^*\|_1] \leq \epsilon.$$

Letting $\boldsymbol{\eta} = \boldsymbol{p} - \boldsymbol{p}^*$ under this coupling, we see that $\mathbb{E}[\|\boldsymbol{\eta}\|_1] \leq \epsilon$. By Markov's inequality, we have $\mathbb{P}[\|\boldsymbol{\eta}\|_1 > \delta] \leq \epsilon/\delta$.

---

[17]It is worth noting that this type of coverage guarantee can only be directly evaluated when we have access to the true $f^*(\boldsymbol{x})$ values, or at least approximately as $k$-snapshots. Of course, even when we do not, the guarantee still holds mathematically.

We now prove the left-hand inequality. We have

$$
\begin{aligned}
\mathbb{P}[\boldsymbol{p} \in S] &= \mathbb{P}[\boldsymbol{p}^* + \boldsymbol{\eta} \in S] \\
&= \mathbb{P}[(\boldsymbol{p}^* + \boldsymbol{\eta} \in S) \wedge (\|\boldsymbol{\eta}\|_1 \leq \delta)] + \mathbb{P}[(\boldsymbol{p}^* + \boldsymbol{\eta} \in S) \wedge (\|\boldsymbol{\eta}\|_1 > \delta)] \\
&\leq \mathbb{P}[\boldsymbol{p}^* \in S_\delta] + \frac{\epsilon}{\delta}.
\end{aligned}
$$

Here the second term is bounded by Markov's inequality, and the first term is bounded because the event $(\boldsymbol{p}^* + \boldsymbol{\eta} \in S) \wedge (\|\boldsymbol{\eta}\|_1 \leq \delta)$ is a subset of the event $\boldsymbol{p}^* \in S + \delta$. Rearranging gives the left-hand inequality.

The right-hand inequality is very similar:

$$
\begin{aligned}
\mathbb{P}[\boldsymbol{p}^* \in S_\delta] &= \mathbb{P}[\boldsymbol{p} - \boldsymbol{\eta} \in S_\delta] \\
&= \mathbb{P}[(\boldsymbol{p} - \boldsymbol{\eta} \in S_\delta) \wedge (\|\boldsymbol{\eta}\|_1 \leq \delta)] + \mathbb{P}[(\boldsymbol{p} - \boldsymbol{\eta} \in S_\delta) \wedge (\|\boldsymbol{\eta}\|_1 > \delta)] \\
&\leq \mathbb{P}[\boldsymbol{p} \in S_{2\delta}] + \frac{\epsilon}{\delta}.
\end{aligned}
$$

$\square$

In practical situations, we expect that the $\epsilon$ parameter is the one that is fixed first. In this case a reasonable choice is to take $\alpha = \delta = \sqrt{\epsilon}$ in the theorem above and obtain prediction sets with coverage at least $1 - 2\sqrt{\epsilon}$.

Note that this theorem is also applicable if we only have $k^{\text{th}}$-order calibration, as we can leverage Theorem 2.4 to obtain approximate higher-order calibration from approximate $k^{\text{th}}$-order calibration.

### F.1 MOMENT-BASED PREDICTION SETS FROM $k^{\text{TH}}$-ORDER CALIBRATION

We now describe a slightly different way of obtaining prediction sets directly using $k^{\text{th}}$-order calibration, instead of passing through higher-order calibration and using Theorem F.3 as a black box. The idea is to use Theorem 2.5 to approximate the moments of the Bayes mixture, and directly apply a moment-based concentration inequality to the true Bayes mixture. Throughout this subsection we specialize to the binary case.

We will need to slightly adapt Theorem 2.5 to approximate the *central* moments $\mathbb{E}[(\boldsymbol{p} - \mathbb{E}[\boldsymbol{p}])^k]$ of the Bayes mixture as opposed to the moments about $0$. In fact, for the purposes of generating prediction sets it will be more convenient to directly bound the moments around our *estimate* $m_1$ of the mean.

**Corollary F.4.** *Let $\mathcal{Y} = \{0, 1\}$, and let $g : \mathcal{X} \to \Delta \mathcal{Y}^{(k)}$ be $\epsilon$-$k^{th}$-order calibrated with respect to a partition $[\cdot]$. Fix any $x \in \mathcal{X}$. Let $(m_1, \ldots, m_k) \in \mathbb{R}^n_{\geq 0}$ be a vector of moment estimates obtained from Theorem 2.5 such that for each $i \in [k]$,*

$$
\left| m_i - \mathbb{E}_{\boldsymbol{x} \sim [x]}[f^*(\boldsymbol{x})^i] \right| \leq i\epsilon/2.
$$

*Then we can obtain a $k^{th}$ central moment estimate $c_k$ such that*

$$
\left| c_k - \mathbb{E}_{\boldsymbol{x} \sim [x]}[(f^*(\boldsymbol{x}) - m_1)^k] \right| \leq k\epsilon(1 + m_1)^k/2.
$$

*Proof.* For brevity, let $\boldsymbol{p} \sim f^*([x])$ denote a random draw from the Bayes mixture. Observe that

$$
\mathbb{E}[(\boldsymbol{p} - m_1)^k] = \sum_{i=0}^{k} \binom{k}{i} \mathbb{E}[\boldsymbol{p}^i] m_1^{k-i}.
$$

Our estimate $c_k$ is naturally formed by replacing each $\mathbb{E}[\boldsymbol{p}^i]$ term by $m_i$ (taking $m_0 = 1$):

$$
c_k := \sum_{i=0}^{k} \binom{k}{i} m_i m_1^{k-i}.
$$

The error in this estimate can be bounded as follows:

$$\left| c_k - \mathbb{E}[(\boldsymbol{p} - m_1)^k] \right| \leq \sum_{i=0}^{k} \binom{k}{i} \left| m_i - \mathbb{E}[\boldsymbol{p}^i] \right| m_1^{k-i}$$

$$\leq \sum_{i=0}^{k} \binom{k}{i} \frac{i\epsilon}{2} m_1^{k-i}$$

$$\leq \frac{k\epsilon}{2} \sum_{i=0}^{k} \binom{k}{i} m_1^{k-i}$$

$$= \frac{k\epsilon}{2} (1 + m_1)^k.$$

$\square$

We can now obtain prediction sets using a simple moment-based concentration inequality.

**Theorem F.5.** *Let $\mathcal{Y} = \{0, 1\}$, and let $g : \mathcal{X} \to \Delta\mathcal{Y}^{(k)}$ be $\epsilon$-$k^{th}$-order calibrated with respect to a partition $[\cdot]$. Fix any $x \in \mathcal{X}$. Let $\epsilon, \alpha > 0$ be given. Let $(m_1, \ldots, m_k)$ and $c_k$ be as in the previous lemma (Corollary F.4). Let $\epsilon' = k\epsilon(1 + m_1)^k/2$. Suppose that $\delta > 0$ is chosen such that*

$$\delta \geq \left( \frac{c_k + \epsilon'}{\alpha} \right)^{1/k}.$$

*Then the interval $[m_1 - \delta, m_1 + \delta]$ is a prediction set for $f^*([x])$ with coverage at least $1 - \alpha$.*

*Proof.* Again for brevity let $\boldsymbol{p} \sim f^*([x])$ denote a random draw from the Bayes mixture. Let $\epsilon' = \frac{k\epsilon}{2}(1 + m_1)^k$. Assume for simplicity $k$ is even (otherwise take $k - 1$). By Markov's inequality and the previous lemma, we have

$$\mathbb{P}[|\boldsymbol{p} - m_1| \geq \delta] \leq \frac{\mathbb{E}[(\boldsymbol{p} - m_1)^k]}{\delta^k}$$

$$\leq \frac{c_k + \epsilon'}{\delta^k}$$

$$\leq \alpha,$$

by our choice of $\delta$. $\square$

For context, it is helpful to consider the ideal case when $\epsilon = 0$ and we have exact values for all the moments up to degree $k$. In this case $c_k$ is exactly $\mathbb{E}[(\boldsymbol{p} - \mathbb{E}[\boldsymbol{p}])^k]$ and $\delta$ can be set to

$$\frac{\mathbb{E}[(\boldsymbol{p} - \mathbb{E}[\boldsymbol{p}])^k]^{1/k}}{\alpha^{1/k}},$$

which is the best one can hope for using only moment-based concentration.

## G  ADDITIONAL EXPERIMENTS

### G.1  EXPERIMENTAL DETAILS

We configure our wide ResNets on CIFAR-10 using the "Naive NN" formula from Table 2 of Johnson et al. (2024) and the Python implementation of the network from the `uncertainty_baselines` package[18] (Nado et al., 2021). We use a learning rate of $3.799\mathrm{e}{-3}$ and relatively large weight decay of $3.656\mathrm{e}{-1}$. We train the model for 50 epochs with the AdamW optimizer (Kingma & Ba, 2014; Loshchilov & Hutter, 2017). For the first epoch, we warm up the learning rate and apply cosine decay thereafter. All models are trained using Aug-Mix data augmentation (Hendrycks et al., 2020) with the standard hyperparameters used in the `uncertainty_baselines` package.

To calibrate the model for each value of $k$, we create $k$-snapshots by sampling $k$ labels at random once for each image in the calibration set. We do not sample multiple snapshots per image.

---

[18]https://github.com/google/uncertainty-baselines

| METHOD | ALEATORIC ERROR | ACC |
|---|---|---|
| ENSEMBLE | $0.308 \pm 0.018$ | $0.932 \pm 0.004$ |
| EPINET | $0.307 \pm 0.006$ | $0.900 \pm 0.008$ |
| 1-SNAPSHOT MODEL | $0.501 \pm 0.006$ | $0.912 \pm 0.004$ |
| 2-SNAPSHOT MODEL | $0.307 \pm 0.010$ | $0.912 \pm 0.004$ |
| 5-SNAPSHOT MODEL | $0.158 \pm 0.010$ | $0.912 \pm 0.006$ |
| 10-SNAPSHOT MODEL | $0.088 \pm 0.010$ | $0.912 \pm 0.004$ |
| 50-SNAPSHOT MODEL | $\mathbf{0.026} \pm 0.006$ | $0.912 \pm 0.004$ |

Table 1: Expected pointwise aleatoric error (Eq. 4.1) of various methods for uncertainty estimation. 95% confidence intervals are computed across 50 random resamplings of the test and calibration sets, where applicable.

## G.2 EVALUATING OTHER UNCERTAINTY ESTIMATION METHODS IN TERMS OF HIGHER-ORDER CALIBRATION

While the snapshot algorithms we present in Section 2.2 *provably* tend towards higher-order calibration, they are not a requirement for higher-order calibration. It can in principle be achieved by any mixture model or higher-order predictor, even one trained without explicit access to snapshots at all. In this section, we measure how well-calibrated other methods for uncertainty estimation are on the CIFAR-10 image classification task.

In addition to higher-order calibrated snapshot models from our work, we evaluate the following methods:

**Naive decomposition**: As a sanity check, we train a 1-snapshot predictor and treat it as a naive mixture model (i.e., one that places all of its probability mass on the lone distribution output by the predictor).

**Ensemble**: One simple way to obtain a mixture model is to train $N$ independent 1-snapshot predictors (let's call them $f_1, ..., f_N : \mathcal{X} \to \Delta\mathcal{Y}$) and output the uniform mixture over their predicted label distributions: $g(x) = \mathrm{Unif}(f_1(x), ..., f_N(x))$. Concretely, we use the same network architecture we calibrated in Section 4 and, as in Johnson et al. (2024), we use $N = 8$.

**Epinet**: A popular method for uncertainty estimation is the Epinet (Osband et al., 2023). In addition to regular inputs, Epinets are conditioned on an additional random vector $z$. By integrating over possible values of $z$, it is possible to elicit joint predictions from the network (or, in the terminology of this paper, snapshot distributions). Epinets are usually instantiated with a pair of MLPs affixed to the end of a network, trained in tandem with or after the main trunk. One MLP is frozen at initialization; the other is not. Both are conditioned on $z$ and the penultimate hidden state of the main network. Loosely speaking, the Epinet learns to mitigate noise introduced by the frozen MLP with the trainable one, and it learns to do this best in regions of the input space encountered most often during training (that is to say, regions of the input space where there is least epistemic uncertainty).

We attach such MLPs to a regular 1-snapshot CIFAR-10 model and train both in tandem for about 20 additional epochs, using the default hyperparameters in the `enn` Python library[19] (Osband et al., 2023) (notably, 50-dimensional hidden layers, 20-dimensional index vectors, and 5 index vectors per training input) and stopping early based on loss on our validation set. Index vectors are sampled from $\mathcal{N}(0, I_d)$, where $d = 20$ is the dimension of the index vectors, and then normalized. At evaluation time, we use 50 index vectors per input and treat the resulting cloud of predictions as the predicted mixture at that point. If $f(x, z)$ is our finished epinet, then our final model is $g(x) = \mathrm{Unif}(f(x, z_1), ..., f(x, z_{50}))$ for some $z_1, \ldots, z_{50} \sim \mathcal{N}(0, I_d)$.

Results are given in Table 1. We see that calibration with large snapshots is a simple and effective path to strong higher-order calibration.

## G.3 COMPARING OUR HIGHER-ORDER CALIBRATION ALGORITHMS

In this section we compare our two algorithms for achieving higher-order calibration (namely *learning directly from snapshots* and *post-hoc* calibration; see Section 2.2) on the FER+ (Barsoum et al.,

---

[19]https://github.com/google-deepmind/enn

| METHOD | NO. SNAPSHOTS | ALEATORIC ERROR | ACC |
|---|---|---|---|
| LEARNING FROM SNAPSHOTS | 1 | 0.615 | 0.802 |
| POST-HOC | 1 | 0.615 | 0.802 |
| LEARNING FROM SNAPSHOTS | 2 | 0.418 | 0.818 |
| POST-HOC | 2 | 0.349 | 0.802 |
| LEARNING FROM SNAPSHOTS | 3 | 0.369 | 0.823 |
| POST-HOC | 3 | 0.217 | 0.802 |
| LEARNING FROM SNAPSHOTS | 4 | 0.355 | 0.819 |
| POST-HOC | 4 | 0.139 | 0.802 |
| POST-HOC | 10 | 0.041 | 0.802 |

Table 2: Expected pointwise aleatoric error (Eq. 4.1) of two higher-order calibration algorithms (see Section 2.2 for details) evaluated on FER+.

2016) dataset. We use this rather than CIFAR-10H for a fairer comparison since the learning from snapshots algorithm requires more multi-label data to be effective, and FER+ is a larger dataset than CIFAR-10H (36,000 vs. 10,000).

FER+ is a facial recognition dataset of approx. 36,000 $48 \times 48$ grayscale images with 10 possible classes (emotions such as happy, neutral, sad, etc.), and with 10 independent human label annotations per image. Thus $\mathcal{Y}$ is the space of 10 possible emotions, and for each image $x \in \mathcal{X}$, $f^*(x)$ is the uniform distribution over its 10 independent annotations. There is substantial disagreement among annotators; the annotations (when converted to a distribution over $\mathcal{Y}$) have a mean entropy of $0.6$ natural units (STD 0.5). In this paper's terminology, this means that the mean aleatoric uncertainty $\mathbb{E}[\text{AU}_H^*(\boldsymbol{x})] = \mathbb{E}[H(f^*(\boldsymbol{x}))] = 0.6$, where $H$ is the Shannon entropy. Note that for training and evaluation we use a further 80/10/10 train/val/test split of the dataset. For the post-hoc calibration algorithm, the calibration set is composed of half of the test set.

Results are given in Table 2. At least at this model scale, it appears that the post-hoc algorithm achieves strictly better calibration on FER+ for a given number of snapshots than learning from snapshots directly. While the latter has shown promise in Johnson et al. (2024) and achieves non-trivial calibration, it has disadvantages compared to the former: unlike the post-hoc calibration scheme, it requires modifying the parameters of the predictor and also training an output layer of size that grows exponentially with the number of snapshots. However, note that this is not a perfect apples-to-apples comparison as the partitions induced by the two methods are not necessarily the same.

## G.4 BINARY REGRESSION

As a sanity check, we verify that it is possible to create higher-order calibrated models on the synthetic 1D binary regression task from Johnson et al. (2024), which fixes a specific distribution of coins $p(x)$ and a bias distribution $p(y|x)$ with both high- and low-frequency variation. Specifically, we train the neural network defined in their Algorithm 2 to predict 1-snapshots from a training set sampled from the synthetic distribution. We then sample a separate calibration set of 5000 snapshots, apply the algorithm from Section 2.2 to the predictor, and finally measure $\epsilon$ (as specified in Definition 2.3) on a third test set.

We find that, indeed, a predictor calibrated in this fashion achieves calibration error $\epsilon$ lower than naive predictors that output uniform distributions over snapshots or the snapshot distribution that results from fair coins. A 1-snapshot predictor pretrained for 10,000 batches and calibrated with 5,000 10-snapshots reaches $\epsilon = 0.400 \pm 0.23$; both naive baselines achieve $0.959 \pm 0.014$, close to the maximum value of 1. If we relax the condition in Definition 2.3 somewhat and exclude the noisiest 5% of the equivalence classes, the 95% percentile of error, the same model reaches $0.212 \pm 0.089$. All 95% confidence intervals are taken over 50 resamplings of the calibration and test sets.

For a more easily interpretable measure of higher-order calibration, we also compute the mean aleatoric estimation error, as defined in Equation 4.1. See Figure 6 for results. As previously, increasing the size of the snapshots used for calibration does indeed improve estimates of aleatoric uncertainty.

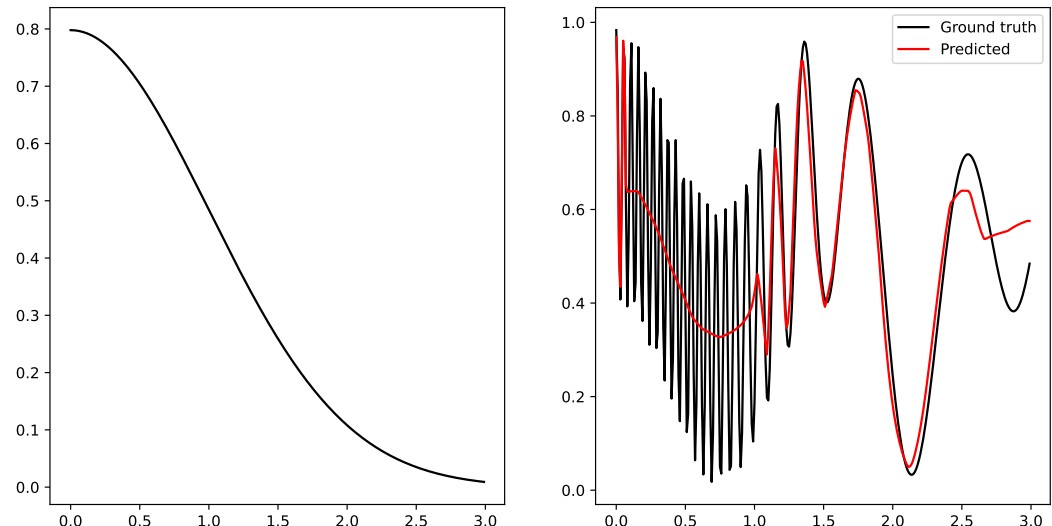

Figure 5: The simple binary regression task from Johnson et al. (2024). (Positive) inputs are drawn from a normal distribution (left), and outputs are determined by a fixed function $p(y|x)$ (right) with low- and high-frequency components, the latter of which our simple predictor ($k = 1$) fails to learn completely.

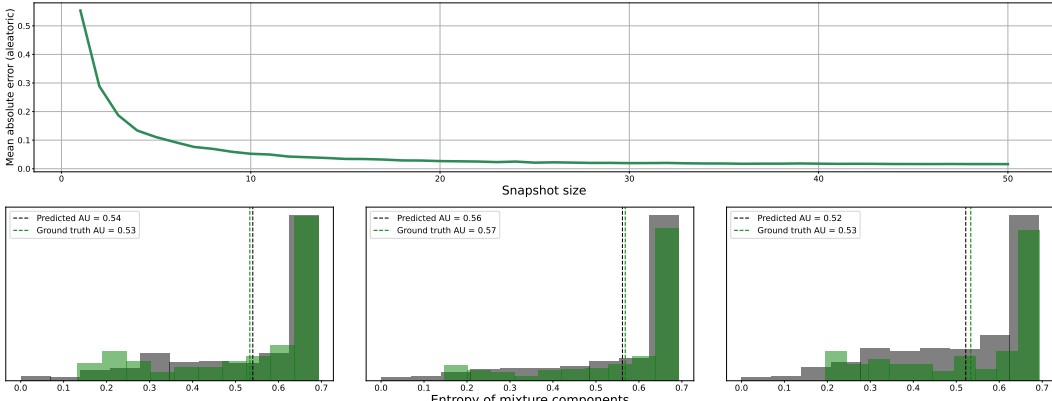

Figure 6: *Top:* Average aleatoric uncertainty estimation error (Eq. (4.1)) of binary regression models calibrated using snapshots of increasing size. *Bottom:* For three of the highest-entropy equivalence classes, we depict the distribution of entropies ranging over components of the predicted mixture (gray) and the Bayes mixture (green). We see that the distributions and in particular the means are similar.

