# OpenReview forum: "Provable Uncertainty Decomposition via Higher-Order Calibration"
_ICLR.cc/2025/Conference — ICLR 2025 Spotlight_

### Official Review · Reviewer_aa9Z · 2024-10-31

**Soundness:** 3
**Presentation:** 2
**Contribution:** 3
**Rating:** 6
**Confidence:** 2

**Summary:**

This work provides a methodology for decomposing predictive uncertainty through calibration on the conditional distribution of the label, using ``higher-order calibration" enabled by multiple samples from conditional distributions.

**Strengths:**

This work addresses an important problem within the area of conditional predictive inference, to my understanding, and includes rigorous theoretical analysis.

**Weaknesses:**

I believe it would be beneficial to improve the clarity regarding the problem setting and objective in a more formal manner.

**Questions:**

1. Could the authors provide some background or explanation on the importance of decomposing predictive uncertainty? Specifically, how do the results of this work guide practitioners in improving prediction or predictive inference, given some data and a trained model in practice? Is the primary aim to enhance understanding of the behavior of a given model or prediction method, rather than to directly improve things?

2. I was wondering if there might be any connection between this work and ``Distribution-free Inference with Hierarchical Data" by Lee and Barber. Specifically, their work considers a similar setting with multiple label samples for each feature observation, focusing on a form of conditional predictive inference. They propose a methodology for constructing a prediction set that achieves a `second-moment coverage guarantee' and note that a 'k-th moment coverage guarantee' can be achieved similarly when the number of repeated samples exceeds k across many groups. Although the inferential targets differ in form between the two works, I feel that there may be a connection in that both works measure conditional uncertainty through higher-order moments using multiple label observations. Could the authors perhaps share their thoughts on this?

---

> ### Author Response · Authors · 2024-11-21
>
> We thank the reviewer for their time and appreciate that they agree on the importance of the problem. We provide responses to their individual questions below:
>
> **Re. background and explanation**
>
> We think good uncertainty estimation could directly guide practitioners as they seek to improve their classifiers.
>
> Suppose we’ve trained a classifier on some dataset. Now suppose that the model is uncertain on a particular datapoint: it assigns high probability to more than one class—its predictive uncertainty is high. What do we do?
>
> Without good uncertainty estimation, this is not obvious. The problem might just be that our data is ambiguous and that the model has accordingly learned to output ambiguous predictions. Alternatively, it might just be that we haven’t trained our model long enough, or that our model is too small. In the former case, the solution is simply to acquire better data; tinkering with our model or its training procedure will not help us at all here. In the latter, we can completely resolve the issue by just modifying our training procedure, without touching the data: we might just want to train longer, or use a larger/better tuned model.
>
> A good uncertainty decomposition reveals precisely which of these two scenarios we’re in (or, if we have some mixture of *both* problems, it tells us what fraction of our predictive uncertainty is because of our data and what fraction is because of our model). Accordingly, it can tell practitioners precisely what they need to do to improve their classifiers.
>
> In response to a request from reviewer y8E9, we have added a detailed walkthrough of our running X-ray example to the appendix (edits in orange), which we hope illustrates the value of high-quality uncertainty decomposition. We have also added a conclusion to the main paper, which likewise summarizes the main points. We hope these help communicate the importance of the problem, but we’d be happy to make further changes if necessary. Let us know what you think!
>
> **Re. Lee et al.**
>
> Thanks for bringing this to our attention! You’re right—like [1], this paper reinforces how two-label data can be used to make much stronger, conditional statements about the distribution of labels for a particular input. There are important differences between their work and ours—*e.g.* they treat specifically with hierarchical data, only indirectly touch on calibration, and rule out the practicality of even higher-order data (they speculate that this would produce unstable intervals, contrary to our work)—but it’s certainly relevant. We’ve added a citation in the most recent version of our manuscript.
>
> Please let us know if we’ve addressed all of the reviewer’s concerns and whether they’d be willing to raise the score.
>
> [1] Johnson, Daniel D., et al. "Experts Don't Cheat: Learning What You Don't Know By Predicting Pairs." arXiv preprint arXiv:2402.08733 (2024).

---

> > ### Comment · Reviewer_aa9Z · 2024-11-26
> >
> > I appreciate the authors' detailed response. All my questions have been resolved.

---

> > > ### Author Response · Authors · 2024-11-26
> > >
> > > Thanks! Please let us know if there'a anything we can do to make the paper more clear and earn a higher score.

---

### Official Review · Reviewer_y8E9 · 2024-11-02

**Soundness:** 4
**Presentation:** 4
**Contribution:** 3
**Rating:** 8
**Confidence:** 3

**Summary:**

The authors explore the implications of higher-order calibration on decomposing the uncertainty of a predictor into aleatoric and epistemic components. In this regard they propose to approximately achieve higher-order calibration through calibrating over a finite k number of snapshots and show the convergence as k goes to infinity. This work extends the second-order calibration using pairs of labels to arbitrarily higher orders with k-snapshots. Authors validate the theoretical claims through experiments.

**Strengths:**

1. Presents important and interesting theoretical formalization on higher-order calibration and convergence of approximate k-order calibration to approximate higher-order calibration, preserving the mathematical rigor.
2. The approximate k-snapshot calibration is certainly important in applying the theory in practice.
2. The paper is well written with a comprehensive study on related works which can be very useful for a novice reader.
3. The experiments are well motivated and demonstrate the theoretical claims made in the paper.

**Weaknesses:**

1. In the experiments, it is useful to mention how the estimation of these quantities of interest were done, for example the how the entropy values were computed empirically in order to get an idea on how the estimation errors of those might affect the overall uncertainty estimation.
2. The running example can be discussed in detail in the main text or in the appendix as an aid for understanding the notation and the definitions.

**Questions:**

Please see Weaknesses.

---

> ### Author Response · Authors · 2024-11-21
>
> We’re glad the reviewer found our approach valuable and that they enjoyed the manuscript. We address their individual comments below:
>
> **Re. estimation of quantities of interest**
>
> We’ve reworded parts of the experimental section to make it clearer how entropy values were computed (highlighted in orange).
>
> **Re. the running example**
>
> Certainly. We’ve updated Section A of the appendix to include a fully elaborated walkthrough of the running example (also highlighted in orange).
>
> We thank the reviewer for their time and hope we have addressed all of their concerns.

---

> > ### Comment · Reviewer_y8E9 · 2024-11-22
> > **Updated version looks good**
> >
> > All my concerns have been addressed well by the authors. I appreciate the effort put into compiling the updated version.

---

### Official Review · Reviewer_HdGS · 2024-11-03

**Soundness:** 4
**Presentation:** 3
**Contribution:** 4
**Rating:** 8
**Confidence:** 5

**Summary:**

In this paper, the authors propose a generalization of classical (first-order) calibration by introducing notions of higher-order and $k$-th order calibration. They formulate and prove theoretical guarantees that a higher-order calibrated predictor provides accurate estimates of aleatoric uncertainty (AU), and they show similar results for $k$-th order calibrated predictors. The authors demonstrate that their theoretical results hold in experiments for image classification.

**Strengths:**

The main strengths of the paper are its theoretical contributions.

Specifically, it generalizes the notion of calibration by extending it to "higher-order" calibration. The authors show that a higher-order calibrated predictor provides correct estimates of the true AU. Moreover, they show that being higher-order calibrated is the necessary and sufficient condition for producing accurate estimates of AU, which I believe is an important theoretical result.

Since in practice we can only have finitely calibrated predictors (i.e., $k$-th order calibrated predictors), the authors additionally prove results on the accuracy of AU estimates for small $k$ for different instantiations of generalized entropies.

Also, compared to the prior work of Johnson et al. (2024), one does not need to change the output layer of the model, but can use the model's architecture as it is.

**Weaknesses:**

I see several weaknesses in the paper, which I list below. Additionally, there are aspects I did not fully understand (which may not necessarily be weaknesses), and I will list them in the Questions section.

### Structure and Text:

I find the structure of the paper somewhat confusing and believe it could be improved. Specifically:

- Figure 1 on page 2 is never referenced in the text. The authors might consider referencing it in the paragraph on lines 37-43, where they essentially describe it in words.

- In lines 101-102, there is a section(?) titled "Summary of main results," which is almost two pages long and is not actually a summary. Rather, it introduces relevant definitions / some theorems, and is too lengthy for a summary. I would suggest that the main results are presented in Sections 2 and 3, not in this section.
The authors might consider renaming this section to something like "Background" or "Preliminaries."

- The paper is missing a Conclusion section. Given the substantial theoretical material presented, a conclusion would help readers to get the key takeaways. For example, the authors could outline the proved results for the correctness of the AU estimate, discuss limitations, and suggest possible further research directions.

- [Minor] Some parts of the Appendix are not referenced, e.g. the proof of Theorem 2.6 in Appendix D.4.


### Missing Related Work:

I believe the paper is missing references to prior work:

[1] Kotelevskii, N., & Panov, M. (2024). Predictive Uncertainty Quantification via Risk Decompositions for Strictly Proper Scoring Rules. arXiv preprint arXiv:2402.10727.

[2] Durasov, N., Dorndorf, N., Le, H., & Fua, P. (2022). Zigzag: Universal sampling-free uncertainty estimation through two-step inference. arXiv preprint arXiv:2211.11435.

The first reference is cited in Hofman et al. (2024) and also provides a way to decompose predictive uncertainties via proper scoring rules (which is one of the key things in the submitted paper).

The second is cited in Johnson et al. (2024) and appears to be the first work in the context of predicting pairs of labels.


### Baseline in experiments:

The current work, as mentioned by the authors, is inspired by the initial results in Johnson et al. (2024). In their approach, they propose a Cheat NN that aims to be second-order calibrated. Using their Cheat NN requires modifying the output layer to output an $l \times l$ matrix (where $l$ is the number of classes), which could be a limitation when $l$ is large. Nonetheless, it represents a third method [in addition to two proposed in line 318] to achieve higher (k-th) order calibration. It would be interesting to see how the proposed strategies compare to the one in Johnson et al. (2024). Including the Cheat NN as a baseline in the experiments would strengthen the paper by providing a direct comparison to existing methods.


----

Overall, I am positive about the theoretical contribution of the paper, and I am willing to significantly increase my evaluation score, given answered questions and incorporated changes.

**Questions:**

1. In lines 196-197, the authors state: "For any given instance we now have not just a single predicted distribution over outcomes, but rather a full 'posterior predictive distribution' over such distributions. In this way every Bayesian model is a higher-order predictor." I want to clarify why the posterior predictive distribution is considered a higher-order predictor. Isn't it an averaged prediction (over the posterior of the weights), hence first order? A sample from the posterior predictive distribution is a label, not a categorical vector?

2. In line 318, the authors propose two methods to achieve $k$-th order calibration: "minimizing a proper loss, or post-processing the level sets of a learned predictor." However, it seems that in the experimental section only one strategy (the post-hoc calibration algorithm) is used. Did the authors employ the first strategy? How do these two strategies perform compared to each other?

3. In line 433, the authors mention: "our epistemic uncertainty is precisely the true “variance” in $f^*(x)$ as $x \sim [x].$." I understand this informal claim, but how can we see that epistemic uncertainty indeed corresponds to the variance of the Bayesian mixture?

4. In the experimental section, the examples of "mostly epistemic" images still resemble in-distribution samples, similar to "soft-OOD" scenarios (e.g., in [1]). I think that higher-order calibration helps identify these objects. However, in practice, it is common to use estimates of EU to detect "hard-OOD" samples, where images do not resemble in-distribution ones (e.g., training images from CIFAR-10 and testing on ImageNet samples). In this case, will higher-order calibration improve out-of-distribution detection?

5. In lines 378-379, the authors state: "Let any concave generalized entropy function $G$ be given..." In prior works (Kotelevskii & Panov (2024) and Hofman et al. (2024)), several choices of generalized entropy are considered, including Shannon entropy and Brier score, but also the Zero-one score. The Zero-one score leads to a convex (but not strictly convex) generalized entropy. Will the necessary and sufficient conditions of being higher-order calibrated to produce calibrated estimates of AU still hold in this case?

[1] Kotelevskii, N., & Panov, M. (2024). Predictive Uncertainty Quantification via Risk Decompositions for Strictly Proper Scoring Rules. arXiv preprint arXiv:2402.10727.

---

> ### Author Response · Authors · 2024-11-21
>
> Thank you for your thorough review! We appreciate the feedback, and have included below answers to your questions as well as how we plan to make changes to address your comments.
>
> **Re. structure/text weaknesses**
>
> You make a great point that it would be useful to include a reference to Figure 1 in the text. We have added a reference in the intro near the location you recommended (all edits in orange). Regarding the "Summary of Main Results" section --- note that the section is intended to end with the paragraph after Theorem 1.2. However, we have broken up the section into “Main result” and “Techniques” to improve clarity. We have added a conclusion section to summarize the key theoretical results, discuss limitations, and suggest potential future work. We’ve ensured all appendices are properly referenced in the main text.
>
> **Re. missing related work**
>
> Thanks for bringing these to our attention. We’ve added citations to both works to the most recent version of the manuscript.
>
> **Re. experiment baselines**
>
> [3]’s “Cheat NN” coincides with the first strategy, as it trains a predictor to predict 2-snapshot outcomes (where the matrix corresponds to pairs of classes). Our suggested first strategy generalizes this approach to k > 2. See our response to Q2 for more details and experiment updates!
>
> **Re. Q1 (posterior predictive distribution)**
>
> The posterior predictive distribution can be seen as a mixture as follows: suppose $f$ is a Bayesian model and $f(x; \theta) \in \Delta Y$ is the output at $x$ with some fixed parameters $\theta$. When $\mathbf{\theta}$ is itself drawn from the posterior over parameters, $f(x; \mathbf{\theta})$ becomes a random distribution. The posterior predictive distribution at $x$ is thus a mixture whose mixture components are given by $f(x; \mathbf{\theta})$. Importantly, we do not average the mixture components --- we refer to that in our terminology as the mixture centroid or in this context the Bayesian Model Average (BMA).
>
> **Re. Q2 (higher-order calibration strategy)**
>
> The first strategy is that employed by [3] for the case k = 2, but it is both less practical than the second strategy (it requires modifying the weights of the predictor and greatly increasing the dimension of the output layer) and also not as strongly supported by our theory. Nevertheless, we’re currently running experiments using the first strategy on a new dataset with enough multi-label datapoints to train a model (CIFAR-10H is just a test/calibration set) and will include them with the final version of the manuscript.
>
> **Re. Q3 (epistemic uncertainty and variance)**
>
> We use ``variance’’ in quotes to refer to an intuitive notion of variation between values, rather than the actual variance of the Bayes mixture. We show in Theorem 1.2 and more generally in Lemma C.1 that this intuitive variance can be formalized as the divergence between the mixture components and the mixture centroid. We’ve edited the sentence to improve clarity.
>
> **Re. Q4 (HOC and OOD Detection)**
>
> While it is indeed common to use EU for OOD detection, our theoretical guarantees only provide formal guarantees with respect to in-distribution data. For this reason, we do not make the claim that HOC would improve hard-OOD detection, but this remains a direction for future study.
>
> **Re. Q5 (concave generalized entropy function)**
>
> Our results with respect to the consequences of higher-order calibration hold very generally for any (strict or non-strict) concave entropy function, and thus for any of the entropy functions associated with proper scoring rules that are discussed in the sources you reference. Our results on estimating uncertainty with kth-order calibration (Theorem 3.3) are more or less efficient depending on the smoothness (in terms of either their approximation by low-degree polynomials, or uniform continuity) of these entropy functions, i.e. less smooth concave entropy functions will require larger values of k to approximate well.
>
> Please let us know if you have any other feedback or questions.
>
> [3] Johnson, Daniel D., et al. "Experts Don't Cheat: Learning What You Don't Know By Predicting Pairs." arXiv preprint arXiv:2402.08733 (2024).

---

> > ### Comment · Reviewer_HdGS · 2024-11-21
> > **Response to authors**
> >
> > Dear Authors,
> >
> > Thank you for your thorough responses to my comments. I appreciate the effort you've put into addressing my concerns, and I believe the manuscript has improved significantly, especially with the addition of the explicit "Summary" section and the restructuring of the "Main Results" section.
> >
> > I'm looking forward to seeing the experimental results on the new dataset for the first strategy, as I think this will enhance the breadth of the paper.
> >
> > I have one final kind request: could you please properly cite [1]? This work is very close to Hofman et al. (2024) and is even cited there as concurrent research that provides similar outcomes in terms of uncertainty decompositions. I believe it's worth mentioning explicitly, as you have done with Hofman et al. (2024) (in the revised manuscript on lines 83, 197, 369, etc.), because currently it is not clearly evident how this work relates to yours.
> >
> > ----
> >
> > [1] Kotelevskii, N., & Panov, M. (2024). Predictive Uncertainty Quantification via Risk Decompositions for Strictly Proper Scoring Rules. arXiv preprint arXiv:2402.10727.

---

> ### Author Response · Authors · 2024-11-25
>
> We apologize for the oversight; you’re right that Kotelevskii et al. should be mentioned alongside Hofman et al. We’ve edited the manuscript to reflect this.
>
> We've also added experiments with the first strategy, finding that it slightly underperforms the post-hoc calibration strategy, at least at the model scale we explore in the paper. See Appendix G3 for details.
>
> Thanks again for your detailed reading of our manuscript! We feel it is much improved from its original version. Please let us know if there are any remaining issues with the paper and whether you’d be willing to raise your score.

---

> ### Comment · Reviewer_HdGS · 2024-11-26
> **Response to authors**
>
> Dear Authors,
>
> I deeply appreciate your engagement and the efforts you've made to address my concerns. From my perspective, the paper now appears more solid and comprehensive.
>
> Given the changes and the new results you've added, I am now more confident that the paper should be accepted. Therefore, I have increased my evaluation score and the confidence level of my assessment.

---

### Official Review · Reviewer_1pi3 · 2024-11-05

**Soundness:** 3
**Presentation:** 3
**Contribution:** 3
**Rating:** 8
**Confidence:** 4

**Summary:**

The authors present a new way of decomposing aleatoric and epistemic uncertainties based on second-order distributions. They claim they are the first one to provide estimates enjoying some calibration properties that they specify, and show empirically their findings.

**Strengths:**

The approach is innovative, and the authors look at calibration of second-order distribution, which is still a shallow-studied subfield. The exposition is mostly clear, and the results they derive are mathematically sound.

**Weaknesses:**

See Questions.

**Questions:**

In Definition 1.1, you write $f^*([x]):=\lbrace{f^*(\mathbf{x}) | \mathbf{x} \sim [x] }\rbrace$. Why is $\mathbf{x}$ suddenly in boldface? Is it just to separate notationally the equivalence class $[x]$ from a possible element $\mathbf{x}$ of such a class? In addition, what does $\mathbf{x} \sim [x]$ mean? From what you write before $[x]$ is just an equivalence class. In turn, the squiggle cannot mean "$\mathbf{x}$ is distributed according to $[x]$". Do you mean that $\mathbf{x}$ is a generic element of the equivalence class $[x]$? This becomes even more puzzling in Theorems 1.2 and 2.7, and Lemma 3.2, where expectations are taken w.r.t. $\mathbf{x} \sim [x]$.

Lines 145-146: "Then we cannot distinguish scenario 1 from scenario 2 if all we receive are ordinary labeled examples with a single label per image". It seems that you rediscovered the classical Fisher-Laplace paradox (see e.g. Appendix A.(i) in [1]).

The author(s) seem to be unaware of an extensive literature that studies the pitfalls of second-order distributions in quantifying and disentangling between AU and EU (see e.g. the last paragraph of [1, Section 5]). In light of this, does their method solve such shortcomings? If so, how? If not, or if only partially, it should be (i) discussed why it is still of interest to take a higher-order route, as the authors do, and (ii) compare the author's method $-$or at least introduce a discussion$-$ with others that do not suffer from these problems, such as credal-set-based uncertainty quantification (see works by Hüllermeier, Sale, Caprio, Desterke to name a few).

How does the author's notion of calibration relates to type-2 validity [2] and credal set calibration of e.g. [3]?

Why in line 270 the authors write $g:\mathcal{X} \rightarrow \Delta \mathcal{Y}^{(k)}$? That is, why is $k$ in parentheses?

The conceptual simplification in line 273 is not immediately clear to me. Predictor $g:\mathcal{X} \rightarrow \Delta\mathcal{Y}^k$ outputs a collection of $k$ probabilities over the labels in $\mathcal{Y}$. Does your interpretation $\Delta\mathcal{Y}^k \subseteq \Delta\Delta\mathcal{Y}$ mean that we can see $g$ as outputting an empirical second-order distribution that gives probability $q/k$ to the elements of $\Delta\mathcal{Y}$, where $q$ is the number of times a specific distribution over the outputs was returned by the predictor $g$? In other words, suppose $\mathcal{Y}=\lbrace{1,2,3}\rbrace$, and that $k=3$. Suppose also that $g(x)=((0.9,0.05,0.05), (0.9,0.05,0.05), (0.8,0.1,0.1))$. Then, this can be seen as an empirical second-order distribution giving probability $2/3$ to $(0.9,0.05,0.05)$, probability $1/3$ to $(0.8,0.1,0.1)$, and probability $0$ to all other elements of $\Delta\mathcal{Y}$. Is this what the authors mean?

The 1-Wasserstein metric is well-defined on $\Delta\Delta\mathcal{Y}$ only when $\mathcal{Y}$ is discrete. This should be mentioned.

Careful in using $G$ as any concave generalized entropy function, since in the literature it usually refers to the generalized Hartley measure (see e.g. eq (27) in [4]).

Why is there not a Conclusion?

---

[1] https://openreview.net/forum?id=4NHF9AC5ui

[2] https://www.sciencedirect.com/science/article/pii/S0888613X21001195

[3] https://arxiv.org/abs/2410.12921

[4] https://link.springer.com/article/10.1007/s10994-021-05946-3

---

> ### Author Response · Authors · 2024-11-21
>
> We thank the reviewer for their positive and informed review and detailed reading. We address their specific questions and comments below:
>
> **Re. questions about notation**
>
> > “In Definition 1.1, you write… w.r.t. x ~ [x]”
>
> We use a convention of referring to random variables using boldface. _x_ ∼ [x] refers to a draw from the marginal distribution D_X restricted to the equivalence class [x]. This notation was indeed not very clearly explained in the main text, only in Appendix A. In the most recent version of the manuscript, we clarify the notation earlier in the manuscript. (All changes in orange)
>
> > “Why in line 270… *k* in parentheses?”
>
> Y^(k) specifically denotes the space of uniform distributions over snapshots. Specifically, an element of Y^(k) is not a tuple (y_1, …, y_k) but rather the distribution Unif{y_1, …, y_k} (equivalently, it is the space of symmetrized tuples where only relative frequency matters). This is an important point --- we currently define it right before Definition 2.3 in the manuscript and emphasize it more carefully in the revised version. This is exactly the way in which Y^(k) is directly a subset of \Delta Y and hence \Delta Y^(k) a subset of \Delta \Delta Y. We hope this also addresses the question about the conceptual simplification in Line 273.
>
> > “The 1-Wasserstein… should be mentioned”
>
> Indeed; we do state that Y must be discrete when we introduce it. We have added a note emphasizing this fact to the paragraph where we introduce Wasserstein distance.
>
> > “Careful in using… (see e.g. eq (27) in [4]).”
>
> In the context of the current paper, we feel that “G” does not risk much confusion. Please also note that the authors of [4] use “GH” to refer to the generalized Hartley measure, not “G”.
>
> **Re. the Fisher-Laplace paradox**
>
> This is indeed a common motivating problem in the literature—for example, [5], as well as various papers on Bayesian deep learning and credal sets make similar arguments. To be clear, we do not claim to have discovered this issue with single-distribution predictors. We simply propose 1) a new way to evaluate any solution to the problem and 2) rigorous and practical solutions of our own.
>
> **Re. pitfalls of higher-order distributions**
>
> We did make an effort to cite as many relevant papers as possible (including several works by Hüllermeier and Sale and another by Caprio), but we thank the reviewer for drawing attention to a few that we missed and the need for further discussion. We also note that there is additional discussion of related work in Appendix B, including on mixture-based uncertainty decompositions.
>
> Regarding the pitfalls of higher-order distributions, Bengs et al. ([6], cited in [1, Section 5]) arguably captures the key concern: given only level-0 samples (i.e. ordinary labeled examples), it does not make sense to learn a level-2 predictor (i.e. a mixture predictor). This same concern also underlies the critique in [7] and [8] of one popular way of obtaining higher-order distributions, namely evidential deep learning.
>
> We agree with this concern completely (it is also related to the Fisher-Laplace paradox just discussed). It is exactly why we instead learn a Level-2 predictor from k-snapshots, ie multilabeled examples (which are a kind of Level-1 sample) (see the last paragraph of Bengs et al, Section 2.3 for a similar point). Without k-snapshots, we fully agree that existing approaches to using higher-order distributions fall short in various ways. Moreover, our work identifies exactly *how* these approaches fall short, and proposes (a) learning from k-snapshots as a principled way of learning mixtures, and (b) higher-order calibration as a principled way of ensuring that these mixtures have clear frequentist semantics. In fact, higher-order calibration also yields an evaluation metric for all such methods producing higher-order distributions.
>
> Other methods of representing higher-order uncertainty such as credal sets tackle these concerns in different ways. We view mixtures as especially expressive and natural, since they occur already in common methods such as Bayesian and ensemble methods. Note also that there is a considerable literature on mixture-based uncertainty decompositions (by Hüllermeier and Sale and others) that also takes mixtures as a starting point; see Appendix B for discussion. Ultimately, however, all such representations are formalisms for a subtle real-world concept, with various pros and cons. Our core reason for using mixtures is that we can prove a strong formal result such as Thm 1.2, which we do not see how to do for other representations. We have included some of this discussion in the extended related work (Appendix B, in orange) in the updated manuscript.

---

> ### Author Response · Authors · 2024-11-21
>
> **Re. type-2 validity**
>
> Thank you for bringing this to our attention. Type-2 validity [2] is a certain guarantee on the predicted probability distribution for a new unseen point drawn marginally from the underlying distribution. Importantly, it is not merely a coverage guarantee about the _value_ of the new point, but rather a guarantee about its _distribution_. Higher-order calibration provides a guarantee that is similar in spirit but different in its particulars. A particular consequence is _higher-order prediction sets_ with coverage guarantees for entire ground truth probability distributions --- please see Appendix F for details. Moreover, these guarantees hold conditionally over an equivalence class rather than merely marginally over the entire distribution.
>
> For further discussion of the differences between conformal prediction and prediction intervals derived from second-order calibration (a special case of our notion of higher-order calibration), please see a nice section of the appendix of [5] titled “Comparison to other distribution-free statistical guarantees”.
>
> **Re. credal sets**
>
> Credal sets are not second-order probability distributions, and in particular allow expressing total ignorance via the empty credal set. However, for many purposes a nonempty credal set behaves similarly to the uniform mixture over that set, and in this sense mixtures are still a useful and general language with significant pros of their own. Ultimately we believe these are incomparable approaches. A key benefit of mixtures for us is that we can prove formal guarantees such as Thm 1.2 for mixtures. We are not aware of similar guarantees for credal sets --- note that we were not able to find a definition of credal set calibration in [3].
>
> **Re. conclusion**
>
> We have added a conclusion to the most recent draft of the manuscript.
>
>
> Please let us know if we missed anything or if there are concerns that still remain. We would be happy to engage further.
>
> [5] Johnson, Daniel D., et al. "Experts Don't Cheat: Learning What You Don't Know By Predicting Pairs." arXiv preprint arXiv:2402.08733 (2024).
>
> [6] Bengs, Viktor, Eyke Hüllermeier, and Willem Waegeman. "Pitfalls of epistemic uncertainty quantification through loss minimisation." Advances in Neural Information Processing Systems 35 (2022): 29205-29216.
>
> [7] Jürgens, Mira, et al. "Is Epistemic Uncertainty Faithfully Represented by Evidential Deep Learning Methods?." arXiv preprint arXiv:2402.09056 (2024).
>
> [8] Pandey, Deep Shankar, and Qi Yu. "Learn to accumulate evidence from all training samples: theory and practice." International Conference on Machine Learning. PMLR, 2023.

---

> > ### Comment · Reviewer_1pi3 · 2024-11-21
> > **Thank you**
> >
> > I'm satisfied with the authors' answers, and I'm happy to keep my score. Just one last comment: I would say complete ignorance in the case of credal sets is captured by the fully vacuous set (i.e. the set of all countably additive probability measures), and not by the empty set, to which I have difficulties to attach an interpretation. In addition, for credal sets, calibration properties have been recently proposed in https://arxiv.org/pdf/2411.04852, Propositions 5 and 6. Finally, what I meant by citing [3] (especially Section 3.1) is that you can test whether a precise (true) distribution belongs to a credal set of interest.

---

> > > ### Author Response · Authors · 2024-11-22
> > >
> > > Thank you for the prompt response and the clarification regarding credal sets. We’re happy to cite [3] and this new preprint as concurrent work. We’ve added both to our new related works section on credal sets (Appendix B).
> > >
> > > Please let us know if the manuscript has any remaining weaknesses standing in the way of a stronger endorsement.

---

> > > > ### Comment · Reviewer_1pi3 · 2024-11-26
> > > > **Thank you**
> > > >
> > > > I'm happy with the author response, and I'm happy to increase my score

---

### Meta-Review · Area_Chair_Z68f · 2024-12-18

**Metareview:**

This study proposes a novel approach for the uncertainty quantification about two types of uncertainty—aleatoric and epistemic uncertainty—by introducing a new concept of calibration. By extending the traditional notion of calibration, the authors propose the concept of higher-order calibration based on the mixture of predictive probabilities, which provides guarantees for the estimation of aleatoric uncertainty. Furthermore, the study introduces k-th order calibration as a more practical formulation of higher-order calibration, demonstrating that the proposed method is not only of theoretical interest but also practically effective.
One identified weakness was the lack of discussion regarding the relationship with existing studies, but this was addressed during the discussion phase. All reviewers strongly appreciated the novelty and contributions of this study, and I recommend it for acceptance.

**Additional Comments On Reviewer Discussion:**

Concerns regarding the relationship with existing studies were raised by Reviewer 1pi3, Reviewer aa9Z, and Reviewer HdGS. However, these concerns were adequately addressed during the discussion. Reviewer y8E9 requested a more detailed explanation of the example presented in the main text, which was resolved by adding supplementary explanations in the Appendix.
Reviewer HdGS also pointed out a lack of baseline experiments, but this concern was addressed through the inclusion of additional experiments.
Other issues, such as insufficient discussion, were also raised, but all concerns have been resolved. Therefore, there are no remaining issues preventing this work from being published.

---

### Decision · Program_Chairs · 2025-01-22

Accept (Spotlight)